# Spatially resolved transcriptomics reveals the architecture of the tumor-microenvironment interface

Miranda V. Hunter 1,4, Reuben Moncada2,4, Joshua M. Weiss1,3, Itai Yanai2,5✉ & Richard M. White 1,5✉

During tumor progression, cancer cells come into contact with various non-tumor cell types, but it is unclear how tumors adapt to these new environments. Here, we integrate spatially resolved transcriptomics, single-cell RNA-seq, and single-nucleus RNA-seq to characterize tumor-microenvironment interactions at the tumor boundary. Using a zebrafish model of melanoma, we identify a distinct "interface" cell state where the tumor contacts neighboring tissues. This interface is composed of specialized tumor and microenvironment cells that upregulate a common set of cilia genes, and cilia proteins are enriched only where the tumor contacts the microenvironment. Cilia gene expression is regulated by ETS-family transcription factors, which normally act to suppress cilia genes outside of the interface. A cilia-enriched interface is conserved in human patient samples, suggesting it is a conserved feature of human melanoma. Our results demonstrate the power of spatially resolved transcriptomics in uncovering mechanisms that allow tumors to adapt to new environments.

[1] Cancer Biology and Genetics, Memorial Sloan Kettering Cancer Center, New York, NY, USA. [2] Institute for Computational Medicine, NYU Langone Health, New York, NY, USA. [3] Weill Cornell/Rockefeller/Sloan Kettering Tri-Institutional MD-PhD Program, Memorial Sloan Kettering Cancer Center, New York, NY, USA. [4] These authors contributed equally: Miranda V. Hunter, Reuben Moncada. [5] These authors jointly supervised this work: Itai Yanai, Richard M. White. ✉email: itai.yanai@nyulangone.org; whiter@mskcc.org

As tumors grow and invade into new tissues, they come into contact with various new cell types, but it is poorly understood how these cell–cell interactions allow for successful invasion and tumor progression. In melanoma, these interactions can occur between the tumor cells and a diverse number of cell types. In many cases, the tumor cells interact directly with stromal cells such as fibroblasts[1] or immune cells[2]. However, increasing evidence suggests that the repertoire of such interactions is considerably broader, and can include cell types including adipocytes[3] and keratinocytes[4]. Many of these cell interactions can influence tumor cell behavior[1–4].

There are likely at least two levels of cell–cell interactions that are relevant to cancer: "microenvironmental" interactions in which the tumor cell directly interacts with adjacent non-tumor cells, and "macroenvironmental" interactions, in which the tumor cell indirectly interacts with more distant cells. The micro-environment is increasingly appreciated to play a major role in cancer phenotypes, including proliferation, invasion, metastasis, and drug resistance[5,6]. However, it is debatable whether every cell type that a tumor interacts with is truly part of the micro-environment, since the mechanisms by which these cells influence tumor cell behavior are often unclear. This uncertainty is compounded by the fact that tumor cells themselves are highly heterogeneous[7], making it challenging to determine which subset of tumor cells are directly interacting with surrounding non-tumor cells. The macroenvironment may also influence tumor progression, since the tumor cells can interact with other cells in the body at a distance, as recently demonstrated for metabolic coupling between melanoma cells and distant cells in the liver[8].

A better understanding of the nature of these cell–cell interactions requires high resolution imaging and analyses of genes expressed by tumor cells as they interact with different cell types. While bulk and single-cell RNA-sequencing approaches have improved our ability to understand cell–cell interactions, these techniques require dissociation of the tissue of interest, resulting in a loss of spatial information. Thus, a comprehensive understanding of how tumor and surrounding cells interact in situ is lacking, at least in part due to the limitations of current RNA-sequencing technologies.

Spatially resolved transcriptomics (SRT) has recently emerged as a way to address the limitations of both bulk and single-cell RNA-seq by preserving tissue architecture, while still profiling the genes expressed by the cell or tissue at high resolution. Current SRT techniques typically either use spatially-barcoded probes to capture and sequence mRNA from tissue sections[9,10], or multiple rounds of in situ hybridization, sequencing, and imaging to computationally reconstruct the transcriptional landscape of the cell[11,12]. In situ hybridization-based SRT techniques allow the user to profile the transcriptional landscape of the cell at cellular or even subcellular resolution, whereas the resolution of techniques that capture and sequence mRNA from sections is limited by the diameter of each capture spot on the SRT array (for example, 55 μm with the current 10× Genomics Visium SRT technology, with a 45 μm gap between spots). However, to overcome the limited spatial resolution of SRT arrays, a number of computational methods to infer single cell resolved gene expression profiles have recently been developed, including SPOTlight[13] and Stereoscope[14]. We recently developed a technique to integrate capture probe-based SRT and scRNA-seq to map the transcriptomic and cellular architecture of tumors[15]. This provides a unique opportunity to understand the mechanisms that are driving the cell–cell interactions that occur between the tumor and its immediately adjacent microenvironment.

Here, we integrate SRT, single-cell RNA-seq, and single-nucleus RNA-seq to characterize the transcriptional landscape of melanoma cells as they interact with the immediately adjacent microenvironment. Using a zebrafish model of melanoma, we construct a spatially-resolved gene expression atlas of transcriptomic heterogeneity within tumors and surrounding tissues. We discover a histologically invisible but transcriptionally distinct "interface" region where tumors contact neighboring tissues, composed of cells in specialized tumor-like and microenvironment-like states. We uncover enrichment of cilia genes and proteins at the tumor boundary, and find that ETS-family transcription factors regulate cilia gene expression specifically at the interface. We further demonstrate that this distinct "interface" transcriptional state may be conserved in human melanoma, suggesting a conserved mechanism that presents opportunities for halting melanoma invasion and progression.

## Results

**Spatially resolved transcriptomics reveals the architecture of the melanoma–microenvironment interface.** To investigate the transcriptional landscape of tumors and neighboring tissues in situ with spatial resolution, we processed frozen sections from three adult zebrafish with large, invasive $BRAF^{V600E}$-driven melanomas[16] for capture probe-based spatially resolved transcriptomics (SRT), using the 10× Genomics Visium platform (Fig. 1a–c). Although the size of the tissue section used is limited by the size of the Visium array (6.5 mm$^2$), zebrafish allow us the unique advantage that a transverse section through an adult fish (~5 mm diameter) fits in its entirety on the array (Fig. 1c). Zebrafish are thus one of the only vertebrate animals that can be used to study both the tumor and all surrounding tissues in their intact forms, without any need for dissection. Our SRT dataset contained transcriptomes for 7281 barcoded array spots across three samples, encompassing 17,317 unique genes (Fig. 1d, e and Supplementary Fig. 1). We detected approximately 1000–15,000 transcripts (unique molecular identifiers, UMIs) and 500–3000 unique genes per spot, with somewhat fewer UMIs/genes detected in sample C (Supplementary Fig. 1). Visium array spots within the tumor region typically contained more UMIs than spots in the rest of the tissue (Supplementary Fig. 1d, f), likely at least in part due to higher density of cells within the tumor region (Supplementary Fig. 1g).

We first combined our expression matrices using an anchoring framework to identify common cell states across different datasets[17] (Supplementary Fig. 2a). After community-detection based clustering on our integrated dataset, we inferred the identities of 13 distinct clusters. When we projected the cluster assignments back onto the tissue coordinates (Fig. 1d) and onto the UMAP embeddings for each spot (Fig. 1e), we found complex spatial patterns in the data that strongly recapitulated tissue histology. Our Visium data captured multiple microenvironment cell types (muscle, liver, brain, skin, pancreas, heart, intestine, and gills) in addition to the $BRAF^{V600E}$-driven melanomas (Fig. 1d–f). We validated our cluster assignments by plotting onto the Visium array the expression of marker genes that should be expressed exclusively in the tumor ($BRAF^{V600E}$), muscle (*pvalb4*), heart (*kcn6a*), and nervous system (*mbpa*), and observed that expression of these marker genes was restricted to the expected regions of the tissue (Fig. 1f).

To further characterize the transcriptional architecture of the microenvironment, we asked whether we could leverage publicly available, annotated gene sets to uncover spatially-organized patterns of biological activity across the tissue. To this end, we computed the mean expression of genes associated with all zebrafish Gene Ontology (GO) terms, and measured the distance between spots that highly express these genes, reasoning that shorter distances between spots may represent underlying spatial organization of these biological pathways across the tissue. We

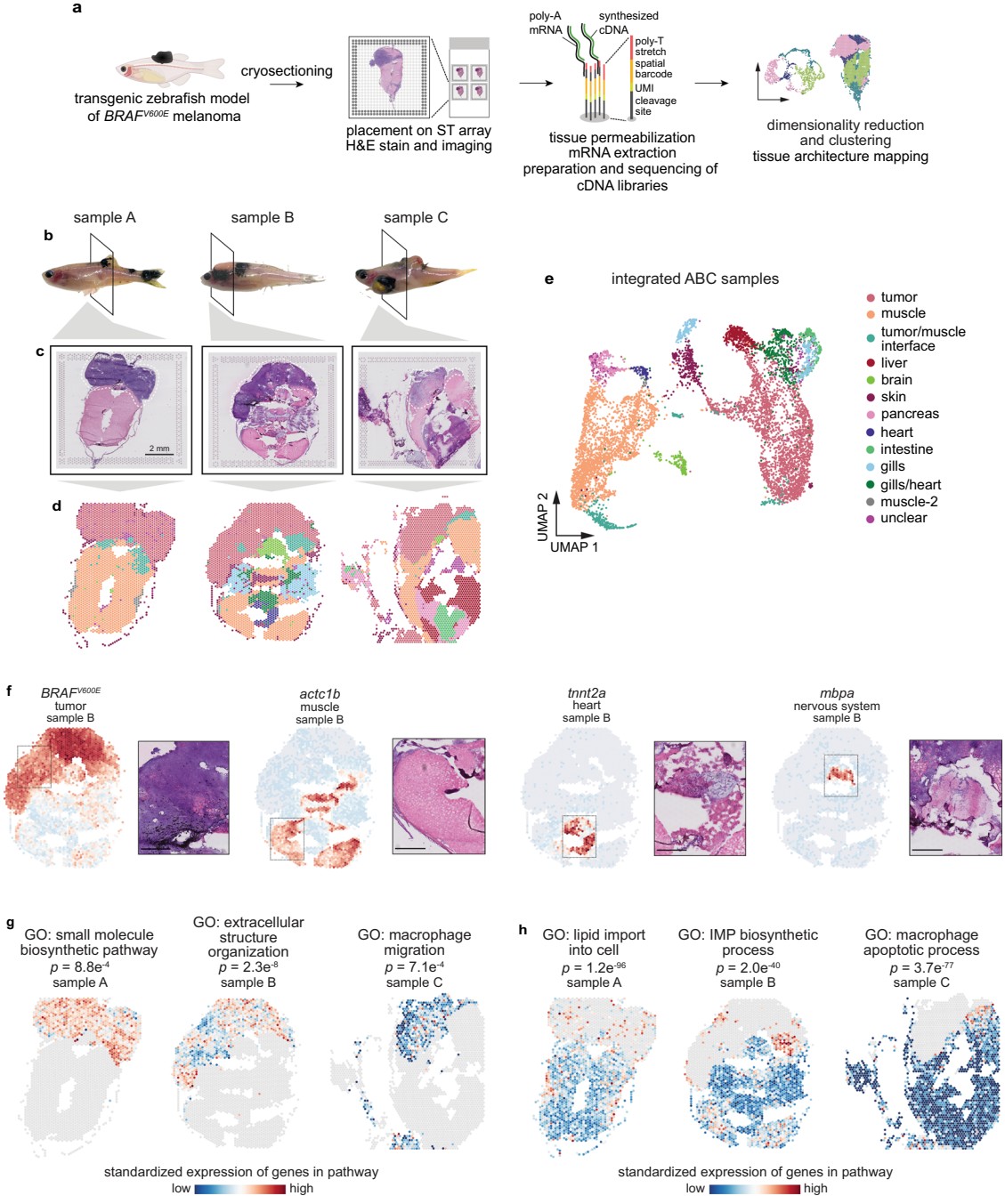

**Fig. 1 Spatially resolved transcriptomics reveals the transcriptional architecture of melanoma and its surrounding microenvironment. a** Schematic showing the spatially resolved transcriptomics (SRT) experiment workflow. **b** Images of zebrafish with $BRAF^{V600E}$-driven melanomas used for SRT. The region where the fish were sectioned is highlighted. Scale bar, 2 mm. **c** H&E staining of cryosections used for SRT ($n = 3$ sections). **d** Visium array spots colored by clustering assignments of the integrated dataset (see "Methods" section). **e** UMAP embedding of SRT spots from all three samples colored by cluster assignments of the integrated dataset (see "Methods" section). **f** The expression of select marker genes ($BRAF^{V600E}$, tumor; *pvalb4*, muscle; *kcn6a*, heart; *mbpa*, nervous system) from the SRT data projected over tissue space (left), with images of the corresponding histology from the indicated region of the SRT array (right). Scale bar, 500 μm. **g, h** Average, standardized expression of annotated genes for gene ontology (GO) terms displaying spatially-coherent expression patterns in the tumor (**g**) and microenvironment (**h**) regions in each sample. *p*-values represent the comparison between the distance between spots expressing that GO term genes and a null-distribution of distances between random spots (Wilcoxon's Rank Sum test, two-sided).

then compared this distribution to that of a null distribution of distances between random spots, allowing us to identify GO terms with spatially coherent, non-random expression patterns. Applying this to the tumor region of our samples, we identified several GO terms displaying interesting spatial expression patterns related to tissue structure (GO: extracellular structure

organization; $p = 2.3 \times 10^{-8}$) and the immune system (GO: macrophage migration, $p = 7.1 \times 10^{-4}$), among others (Fig. 1g and Supplementary Fig. 3a and Supplementary Movie 1). We performed the same analysis on the microenvironment, and found several notable spatially-organized pathways that function in tumor growth and invasion (GO: lipid import into cell,

$p = 1.2 \times 10^{-96}$; GO: IMP biosynthetic process, $p = 2.0 \times 10^{-40}$) (Fig. 1h and Supplementary Fig. 3b and Supplementary Movie 2). Together, these data validate our spatially resolved transcriptomics workflow and demonstrate the existence of discrete tumor and microenvironment regions within our SRT dataset.

**The tumor–microenvironment interface is transcriptionally distinct from the surrounding tissues**. We noticed in all of the samples a transcriptionally distinct cluster of array spots that localized to the border between the tumor and the adjacent microenvironment (Fig. 1c–e), in which specific biological pathways were upregulated (Fig. 1h). This "interface" cluster was present in all three samples (Supplementary Fig. 2). Interestingly, the tissue in this interface region appeared largely indistinguishable from the surrounding microenvironment (muscle) (Fig. 2a), despite it being transcriptionally distinct (Fig. 1e). We thus hypothesized that this interface cluster represented the region in which the tumor was contacting neighboring tissues. To get a better sense of the transcriptional profile of the interface cluster, we computed the correlation between the averaged transcriptomes of each SRT cluster across all three samples. We found that the transcriptional profile of the interface cluster was more correlated with the tumor ($R = 0.33$) than with muscle ($R = 0.06$) (Fig. 2b), despite the fact that the tissue in this region histologically resembles muscle with few tumor cells visible (Fig. 2a).

We next sought to identify genes that may differentiate the interface from muscle (to which it is most similar histologically) and from tumor (to which it is most similar transcriptionally) (see "Methods" section). We found a number of genes that were upregulated specifically in the interface cluster relative to both tumor and muscle, including, interestingly, a number of uncharacterized genes, genes related to increased transcriptional/translational activity (*atf3*, *eif3ea*, and ribosomal genes), and genes related to the microtubule cytoskeleton (*tuba1a* and *tuba1c*) (Fig. 2c and Supplementary Data). The upregulation of most of these genes was subtle (though statistically significant; Supplementary Data), which may be due to the somewhat lower cellular resolution of the Visium technology and number of UMIs detected per spot (note: to address this, we further compare the magnitude of changes for these genes in our single cell datasets below). To identify gene expression programs that are enriched specifically at the interface and provide further evidence for the interface as a transcriptionally distinct tissue region, we performed non-negative matrix factorization (NMF) on all microenvironment spots (including both interface and muscle clusters) across all samples (see "Methods" section). When we projected the NMF factor scores onto each spot, we found that some factors were enriched across all three samples (e.g., factor 2, Supplementary Fig. 4), whereas some were only enriched in one or two of the samples (e.g., factors 4, 11, Supplementary Fig. 4). These differences may be due to different tissue types present across the three samples. Notably, we also found that multiple factors were specifically enriched at the interface between the tumor and the microenvironment (factors 7 and 8; Fig. 2d and Supplementary Fig. 4). To investigate the biology underlying the genes contributing to each factor, we looked for significantly enriched GO terms among the top 150 genes contributing to each factor (Fig. 2e and Supplementary Fig. 4). This revealed several factors enriched in muscle-specific genes, as expected (Supplementary Fig. 4), and that the interface factors were enriched in genes functioning in biological processes including membrane-bound organelles, protein targeting to organelles and the membrane, and DNA replication (Fig. 2e and Supplementary Fig. 4). This result suggests a high degree of biological activity within the interface region, with a potential role for membrane-bound organelles in signaling within this region (the role of such organelles is discussed below). Together, these data uncover a unique "interface" region bordering the tumor, which histologically resembles the microenvironment, transcriptionally resembles tumor, but expresses distinct gene modules that may contribute to tumor–microenvironment cell interactions.

**The tumor–microenvironment interface is composed of specialized cell states**. Our SRT results so far detail a transcriptionally distinct "interface" region where tumors contact the microenvironment. However, spatially resolved transcriptomics data is limited in resolution by the diameter of each spot on the Visium array (55 μm with current technology). Thus, each array spot probably captures transcripts from multiple cells. As the interface region is, by nature, likely a mixture of tumor and microenvironment cells, we performed single-cell RNA-seq (scRNA-seq) on tumor and non-tumor cells from three adult zebrafish with large melanomas (Fig. 3a) in order to better define the cell states present in the interface. We detected approximately 10,000–75,000 transcripts and 1000–5000 unique genes per cell (Supplementary Fig. 5a–c). As expected, our scRNA-seq data contained tumor cells as well as various non-tumor cell types, including erythrocytes, keratinocytes, and several types of immune cells (Fig. 3b). We did not identify a muscle cell cluster in our scRNA-seq dataset, likely because adult skeletal muscle is composed of multinucleated muscle fibers that cannot be isolated and encapsulated for droplet-based scRNA-seq.

Consistent with our SRT results, clustering of our scRNA-seq data revealed a distinct "interface" cell cluster (Fig. 3b–d), which we identified based on the fact that cells in this cluster significantly upregulated the same genes that were upregulated in our SRT interface cluster ($p = 1.83 \times 10^{-26}$; Fig. 3c). The distinct clustering of the interface population was not due to the presence of a significant number of cell doublets within this cluster (Supplementary Fig. 5g). Strikingly, UMAP and principal component analysis of the interface cluster revealed two distinct cell populations (Fig. 3d–f), one expressing tumor markers such as $BRAF^{V600E}$ and the other expressing muscle genes such as *ckba*, with other genes such as the centromere gene *stra13* upregulated in both populations (Fig. 3e, f). This result suggests that the transcriptionally distinct "interface" region we identified in our SRT data is actually composed of at least two similar, but distinct cell states: a "tumor-like interface" and a "muscle-like interface". The interface region may not be limited to only tumor-like and muscle-like cell states; however, since zebrafish melanomas frequently invade into muscle, this likely contributes to the presence of muscle-like interface cells in our data.

Based on this, we separated the interface cluster into two subclusters, and confirmed that the two subclusters express anti-correlated levels of tumor markers such as $BRAF^{V600E}$, *mitfa*, and *pmela*, and muscle markers such as *ckba*, *neb*, and *ak1* (Fig. 3f). A common set of genes, including many genes related to the microtubule cytoskeleton and cell proliferation such as *stra13*, *stmn1a*, *plk1*, and *haus4*, were upregulated in both subclusters (Fig. 3d–f). Both the tumor-like and muscle-like interface cell states were present in both scRNA-seq samples (Supplementary Fig. 5c–f). The presence of putative "muscle" cells in the interface is particularly notable, in light of the fact that adult skeletal muscle is composed of multinucleated muscle fibers that we were unable to isolate in our scRNA-seq workflow due to their size, evidenced by the lack of a muscle cell cluster in our dataset (Fig. 3b). This could suggest the presence of mono-nucleated muscle cells, or a hybrid tumor-muscle cell state at the invasive front. Previous work suggests that tumor and immune cells can fuse to create a hybrid cell state that contributes to tumor

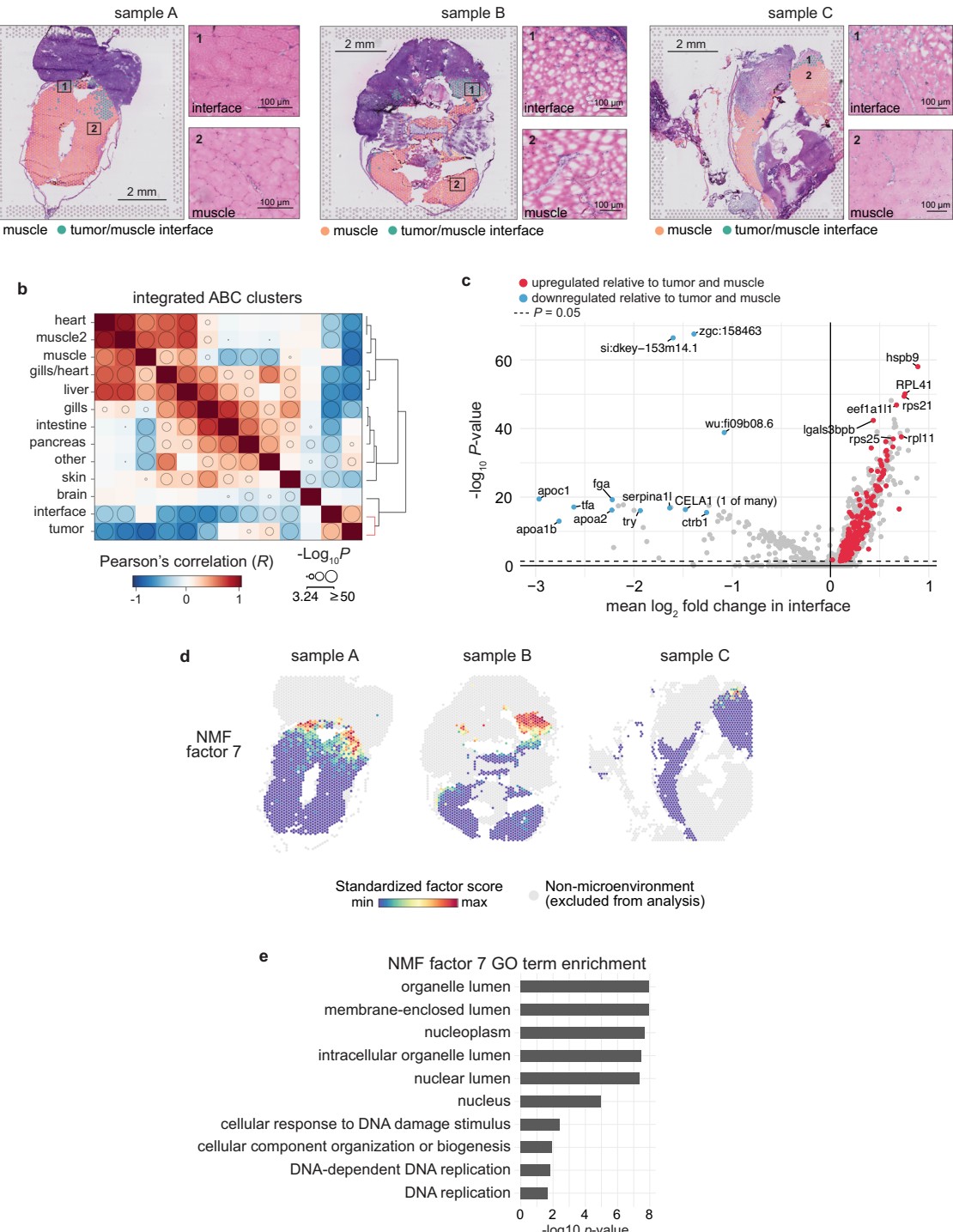

**Fig. 2 The tumor–microenvironment interface is transcriptionally distinct from the surrounding microenvironment. a** Interface and muscle-annotated cluster spots projected onto tissue image ($n = 3$ sections). Insets show the tissue underlying the interface spots (1) and muscle spots (2). **b** Correlation matrix between average expression profile of SRT clusters across all three datasets. Clusters are ordered by hierarchical clustering of the Pearson's correlation coefficients (see "Methods" section) and bubble sizes correspond to $p$-value ($-\log10$) of correlation (two-sided), with $p$-values $< 10^{-3}$ omitted. Clustering of tumor and interface together is highlighted in the dendrogram (red). **c** Volcano plot of differentially expressed genes between the interface cluster versus the muscle and tumor clusters. $p$-values were obtained from the Wilcoxon's rank sum test (two-sided). **d** Non-negative matrix factorization (NMF) of the microenvironment spots (muscle and interface clusters). Shown are the standardized factor scores for interface-specific NMF factor 7, projected onto microenvironment spots. Arrows denote areas with higher factor scores. **e** Enriched GO terms for the top 150 scoring genes in NMF factor 7.

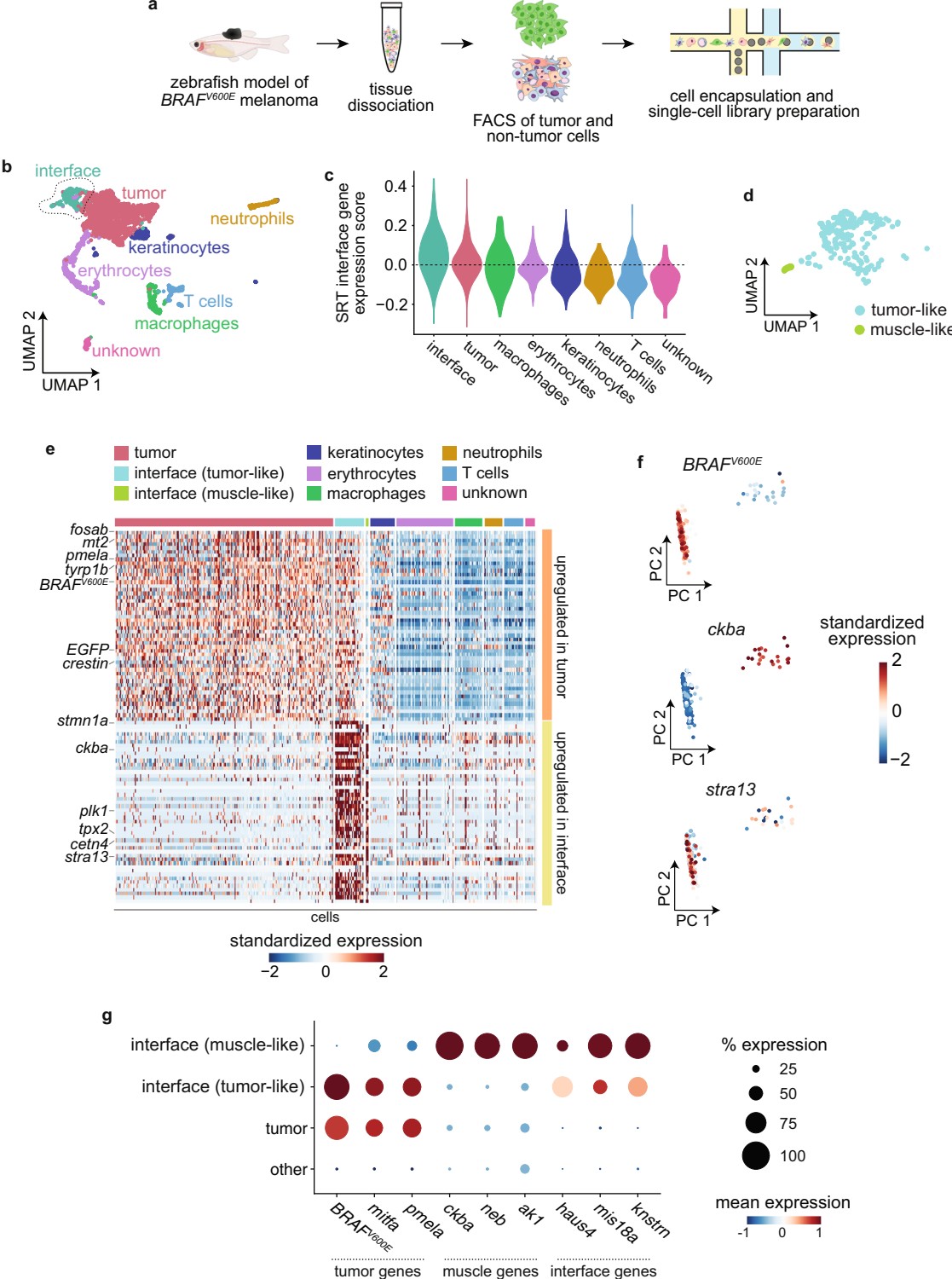

**Fig. 3 The tumor–microenvironment interface is composed of specialized tumor and muscle cells. a** Schematic showing scRNA-seq experiment workflow. **b** UMAP dimensionality reduction plot for 2889 cells sequenced as in **a**. Cluster/cell type assignments are labeled and colored. **c** Expression score per cell (scRNA-seq) for average expression of interface marker genes from the SRT interface cluster. **d** Inset of the outlined interface cluster in **b** showing the two interface subclusters. **e** Heatmap showing expression of the top 50 genes upregulated in the tumor cell cluster (top, orange) and interface cell cluster (bottom, yellow). Selected genes are labeled. **f** Principal component analysis of cells in the interface cluster, scored for expression of the tumor marker $BRAF^{V600E}$, the muscle marker *ckba*, and the centromere gene *stra13*. Scores for principal components 1 and 2 are plotted. Cells are labeled by standardized expression of the indicated genes. **g** Dot plot showing expression of tumor and muscle markers. The size of each dot corresponds to the percentage of cells in that cluster expressing the indicated gene, and the color of each dot indicates the expression level.

heterogeneity and metastasis[18], although tumor-muscle cell fusion has not yet been reported. Together, these data suggest that the interface region is composed of specialized tumor-like and microenvironment-like cell states.

**Interface cell states are distinct from neighboring tissues**. Our results so far indicate that we have uncovered an "interface" cell state localized to where the tumor contacts neighboring tissues. However, our scRNA-seq dataset does not contain a muscle cell cluster due to the fact that muscle fibers cannot be encapsulated for scRNA-seq. This makes it difficult to assess whether the specialized muscle cell state found in the interface is truly distinct from muscle that is not in proximity to the tumor. Thus, to effectively compare the interface cell state(s) to other micro-environment cell types/states that cannot be captured with scRNA-seq, we validated our scRNA-seq results by performing single-nucleus RNA-seq (snRNA-seq) on nuclei extracted from three adult zebrafish, all with large transgenic melanomas. Although snRNA-seq captures only nascent transcripts in the nucleus, which contains only 10–20% of the cell's mRNA[19], scRNA-seq, and snRNA-seq typically recover the same cell states/types, albeit sometimes in different proportions[20]. After quality control and filtering, our dataset encompassed transcriptomes for 10,527 individual nuclei (Fig. 4a). We detected an average of 3800 UMIs and 1350 unique genes per nucleus (Supplementary Fig. 6a, b). Overall gene expression was lower in our snRNA-seq dataset compared to our scRNA-seq dataset, likely due to the relatively low number of transcripts found in the nucleus in general (Supplementary Fig. 6c). Dimensionality reduction and clustering revealed 12 distinct clusters encompassing tumor and non-tumor cell types, including muscle, keratinocytes, liver, and various immune cells (Fig. 4a). We also identified an "interface" cluster in our snRNA-seq dataset (Fig. 4a). We identified the interface cluster based on the fact that nuclei in this cluster strongly upregulated genes that were strongly upregulated in the interface cluster in our scRNA-seq dataset, including *stmn1a* (Figs. 3d and 4b), *stra13* (Figs. 3e and 4b), *plk1* (Figs. 3e and 4b), and *haus4* (Figs. 3f and 4b), and that the interface cluster from our snRNA-seq dataset clustered with the interface cluster from our scRNA-seq dataset when the two datasets were integrated[17,21] (Fig. 4c).

To interrogate the types of nuclei present in the interface cluster in our snRNA-seq dataset, we performed dimensionality reduction and clustering on the nuclei from the interface cluster, which identified five discrete subclusters (Fig. 4d). Similar to our scRNA-seq dataset, within the interface cluster in our snRNA-seq dataset we identified subclusters of nuclei that upregulated tumor-specific or muscle-specific genes (Fig. 4d, e). The interface cluster in our snRNA-seq dataset also contained other subclusters that did not express tumor-specific or muscle-specific genes. Nuclei in these subclusters expressed genes related to other cell types in our snRNA-seq dataset, including immune cells (*ctss2.1*), liver (*fabp10a*), and digestive system (*ela2*) (Fig. 4d, e). This is in line with recent work showing that melanomas can reprogram microenvironmental cells such as liver cells even when not in physical contact[8]. However, similar to our scRNA-seq and SRT datasets, there were many genes that were specifically upregulated across the interface subclusters that were not upregulated in any other cell type in the snRNA-seq dataset, further suggesting that the "interface" cell state is a distinct transcriptional entity (Fig. 4f).

Although our snRNA-seq analysis workflow includes multiple processing steps to exclude doublets, including filtering steps based on the number of UMIs per nucleus and removing possible doublets identified by DoubletFinder[22] (see "Methods" section), to further interrogate whether these tumor-like and microenvironment-like interface nuclei could be attributed to doublets with the corresponding cell type, we quantified the number of UMIs/genes expressed by interface cells/nuclei relative to other cells/nuclei in the dataset. The results were inconclusive: in some cases we quantified significantly more UMIs/genes in interface cells, in some cases we quantified significantly less UMIs/genes in interface cells, but in other cases there was no significant difference (Supplementary Figs. 5b–e and 6b). Thus, to further investigate the presence of doublets in the interface, we calculated the degree of overlap between genes expressed by the tumor/microenvironment nuclei and genes expressed by the corresponding interface nuclei (Supplementary Fig. 7). Although the tumor-like and microenvironment-like interface clusters expressed some tumor-specific and microenvironment-specific genes, as expected (Fig. 4e and Supplementary Figs. 7–8), in most cases there was not a significant degree of overlap between all genes upregulated between both cell states (Supplementary Figs. 7–8), suggesting that these interface cell states are not caused by doublets. We did observe some overlap between all genes expressed by NK cells and macrophages relative to the immune-like interface cells, suggesting that some doublets could be present within the immune-like interface cluster. Notably, tumor-immune cell fusion has been reported in melanoma[18]. Determining whether these potential doublets result from technical or biological reasons will be an important area of future study.

Since our snRNA-seq dataset contained more cells/nuclei and a greater breadth of cell types than our scRNA-seq dataset, we integrated our snRNA-seq data with our Visium SRT data using our recently developed multimodal intersection analysis (MIA) method[15] to confirm the presence of tumor-like and microenvironment-like cell states within the interface region (see "Methods" section; Supplementary Fig. 9). Notably, our MIA results suggested that the interface regions in our SRT dataset were enriched in cell types including muscle, macrophages, and tumor, in line with our scRNA-seq and snRNA-seq results (Supplementary Fig. 9). The cluster that was most significantly enriched in the interface region was the muscle-like interface cell state (Supplementary Fig. 9), in accordance with the histology of our SRT samples that showed that the interface region closely resembles the surrounding muscle (Fig. 2a). Together, these results suggest that the interface is composed of tumor and microenvironment cells which upregulate a common gene program that may contribute to tumor–microenvironment cell interactions at the tumor boundary.

As our SRT results suggest that the interface cell state may be modulated by direct cell–cell interactions between tumor and microenvironment cells, we used NicheNet[23] to computationally infer interactions between interface cells and the rest of the cells in our snRNA-seq dataset by identifying potential ligands expressed by interface cells and receptors and target genes in the other cell types (Supplementary Fig. 10a–c). As the NicheNet model is currently designed to work with human genes, we performed this analysis with the human orthologs of the zebrafish genes in our dataset (see "Methods" section). The top ligand predicted to be active in interface nuclei was *HMGB2* (Supplementary Fig. 10a), of which there are two zebrafish orthologs: *hmgb2a* and *hmgb2b*. These genes were highly expressed in the interface clusters across our snRNA-seq, scRNA-seq, and SRT datasets (Supplementary Fig. 10d). Interestingly, *HMGB2* expression has been reported to be correlated with tumor aggressiveness[24,25]. The predicted receptors for *HMGB2* were *AR*, *ITPR1*, and *CDH1* (fish orthologs: *ar*, *itpr1a*, *itpr1b*, *cdh1*) (Supplementary Fig. 10b). Of these potential receptors, *cdh1* was the most highly expressed in general across the three datasets (Supplementary Fig. 10e). *cdh1* was expressed in various micro-environment cell types, including intestinal cells, keratinocytes, and

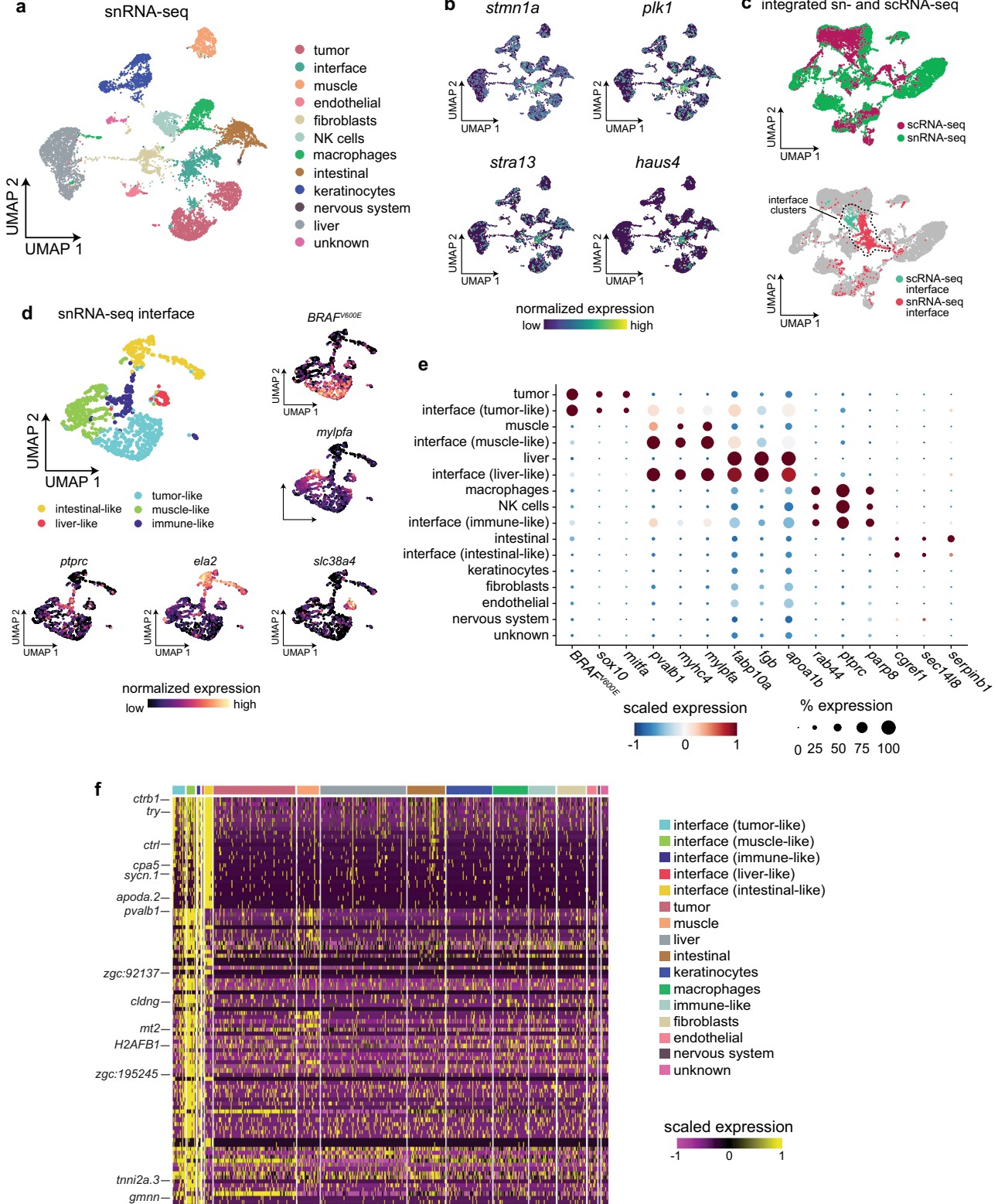

**Fig. 4 Single-nucleus RNA-seq demonstrates that the interface cell states are distinct from the rest of the microenvironment. a** snRNA-seq cluster assignments plotted in UMAP space. **b** Expression of marker genes from the scRNA-seq interface cluster in the snRNA-seq dataset. **c** Integrated UMAP of the snRNA-seq and scRNA-seq datasets (labeled, top plot) showing colocalization of the two interface clusters (bottom plot). **d** Subcluster assignments and expression of marker genes from the snRNA-seq interface cluster. **e** Dotplot showing expression of microenvironment cell-type specific genes within the interface subclusters. **f** Heatmap showing expression of the top 100 genes upregulated across all of the interface subclusters.

also in some interface cell states (Supplementary Fig. 10e). *cdh1* (E-cadherin) is a core component of adherens junctions along with α-catenin and β-catenin[26]. Interestingly, HMGB2 and β-catenin have been reported to cooperate to promote melanoma progression[27]. These data demonstrate one of likely many signaling interactions that occur between interface cells and other cells adjacent to the tumor. Taken together, our results suggest that we have identified a putative "interface" cell state in each of our SRT, scRNA-seq, and snRNA-seq datasets, composed of tumor and microenvironment cells which upregulate a common gene program that may contribute to tumor–microenvironment cell interactions at the tumor boundary.

**Cilia genes and pathways are upregulated at the interface**. To gain further insight into the biological processes underlying the specialized "interface" region identified in our SRT and scRNA-seq data, we performed pre-ranked gene set enrichment analysis (GSEA), using differentially expressed genes in the scRNA-seq interface cluster, to identify conserved pathways that may be active in interface cells. We noticed that many cilia-related pathways were enriched in the combined interface cluster (Fig. 5a). This enrichment of cilia-related pathways occurred in both the muscle-like and tumor-like interface cell states (Fig. 5b). Cilia-related GO terms were also enriched in the SRT interface (Fig. 5c), as were GO terms related to membrane-bound organelles in the genes contributing to NMF factor 7, which localized to the interface (Fig. 2d, e). When we calculated a list of common genes upregulated across the SRT, scRNA-seq, and snRNA-seq interface clusters, several cilia genes were present on this list including *ran*, *tubb4b*, *stmn1a*, and *tuba8l4* (Supplementary Fig. 11 and Supplementary Table 1). Several recent studies have implicated cilia in an important role in melanoma initiation and progression, although the mechanism by which cilia mediate melanoma progression is unclear[28–32]. To further investigate a role for cilia at the tumor–microenvironment interface, we scored each cell from our scRNA-seq dataset for relative enrichment of cilia genes, using the "gold standard" SYSCILIA gene list[33], and quantified a significant upregulation of cilia genes in both interface cell states in our scRNA-seq data, with a particularly strong upregulation in the muscle-like interface cluster (Fig. 5d). Although cilia genes generally were expressed at relatively low levels in our snRNA-seq dataset, in line with the overall lower expression of most genes in our snRNA-seq data relative to our scRNA-seq data (Supplementary Fig. 6c, d), the most highly upregulated cilia genes in the scRNA-seq interface cluster were also upregulated across the tumor-like and muscle-like cell states in the snRNA-seq interface cluster relative to the tumor and muscle clusters (Fig. 5e). Furthermore, we quantified a clear enrichment of cilia genes such as *ran*, *tubb4b*, *tuba4l*, and *gmnn* specifically in tumor-like and muscle-like interface cells in our snRNA-seq dataset, and, similar to our scRNA-seq results, all four genes were upregulated more highly in the muscle-like interface cluster than in any of the other interface clusters (Fig. 5f). Together, these results suggest a potential role for cilia at the tumor–microenvironment interface.

**The tumor–microenvironment interface is ciliated**. Interestingly, previous studies have shown that human and mouse melanomas are not ciliated, although they express cilia genes[28–32]. To reconcile these models, we stained sections through adult zebrafish with invasive *BRAF^(V600E)*-driven melanomas for acetylated tubulin, a common cilia marker[30]. Strikingly, we found that although the bulk of the tumor was not ciliated as expected (Fig. 5g, center and Supplementary Fig. 12), there was a specific enrichment of cilia at the invasive front of the tumor, where it contacts the muscle. We observed long, acetylated tubulin-positive projections that were often found in the extracellular space spanning tumor and adjacent muscle cells (Fig. 5g, h left and Supplementary Fig. 12). These projections were not found in the bulk of the tumor (Fig. 5g, center and Supplementary Fig. 12) or in muscle that was not adjacent to the tumor (Fig. 5g, right and Supplementary Fig. 12). These structures did not resemble typical cilia, which we occasionally see on cultured zebrafish melanoma cells expressing a transgenic cilia reporter (Supplementary Fig. 13), as the acetylated tubulin-positive structures we see in vivo are longer and structurally distinct from typical cilia. Determining the nature and function of these structures will be an exciting area of future study. We could not conclusively determine whether these cilia originated in tumor cells, muscle cells or both cell types, another interesting topic that awaits further study. These data suggest that although the bulk of primary melanomas is not ciliated, cilia are enriched at the tumor–microenvironment interface, where they may facilitate growth of the tumor into surrounding tissues.

**ETS-family transcription factors regulate cilia gene expression at the interface**. To identify potential regulators of gene expression within the interface, we performed HOMER motif analysis[34] to identify conserved transcription factor (TF) binding motifs enriched in genes differentially expressed in the interface. When we performed de novo motif enrichment analysis on genes differentially expressed in the SRT interface compared to normal muscle, the top-ranked motif was the highly conserved ETS DNA-binding domain, containing a core GGAA/T sequence ($p = 1 \times 10^{-22}$; Fig. 6a). The ETS domain was also the top-ranked motif enriched in genes differentially expressed in the SRT interface compared to all other SRT spots ($p = 1 \times 10^{-15}$), and was the second-ranked motif enriched in genes differentially expressed in the interface cluster identified in our scRNA-seq dataset ($p = 1 \times 10^{-13}$; Fig. 6a) and in genes differentially expressed in our snRNA-seq interface cluster ($p = 1 \times 10^{-13}$; Fig. 6a). Furthermore, ETS motifs were frequently enriched in both the tumor-like and muscle-like interface subclusters in our scRNA-seq dataset, along with, notably, motifs for RFX-family transcription factors which regulate ciliogenesis[35] (Fig. 6b). Although ETS-family transcription factors have not been widely studied in melanoma, they have been reported to function in melanoma invasion[36] and phenotype switching[37], and are aberrantly upregulated in many types of solid tumors[38]. Interestingly, zebrafish ETS-family transcription factors were downregulated in the interface in each of our scRNA-seq, snRNA-seq, and SRT datasets (Fig. 6c–e).

To identify potential biological processes that could be regulated by ETS transcription factors at the tumor–microenvironment interface, we investigated putative target genes containing an ETS motif in their promoter. We queried the zebrafish genome for genes with an ETS motif within 500 bp of the transcription start site, filtered these genes to include only those differentially expressed in the tissue/cell state of interest, and performed GSEA on the resulting target gene lists. Surprisingly, within the ETS-target genes in both the SRT and scRNA-seq interface clusters, we again found an enrichment of pathways related to cilia (Fig. 6g). As ETS TFs are downregulated specifically in the interface (Fig. 6d–f), this suggests that ETS-family TFs may act as a transcriptional repressor of cilia genes. ETS TFs can act as transcriptional activators and/or repressors depending on gene and context (Supplementary Table 2). In support of this model, when we scored each cell in the interface for relative expression of both ETS genes and ETS-target genes, the two were strongly anti-correlated ($R = -0.625$, $p = 9.02 \times 10^{-27}$, Fig. 6h). Collectively, these data suggest that ETS-family transcription factors act as transcriptional repressors of cilia genes in cells at the interface between tumors and

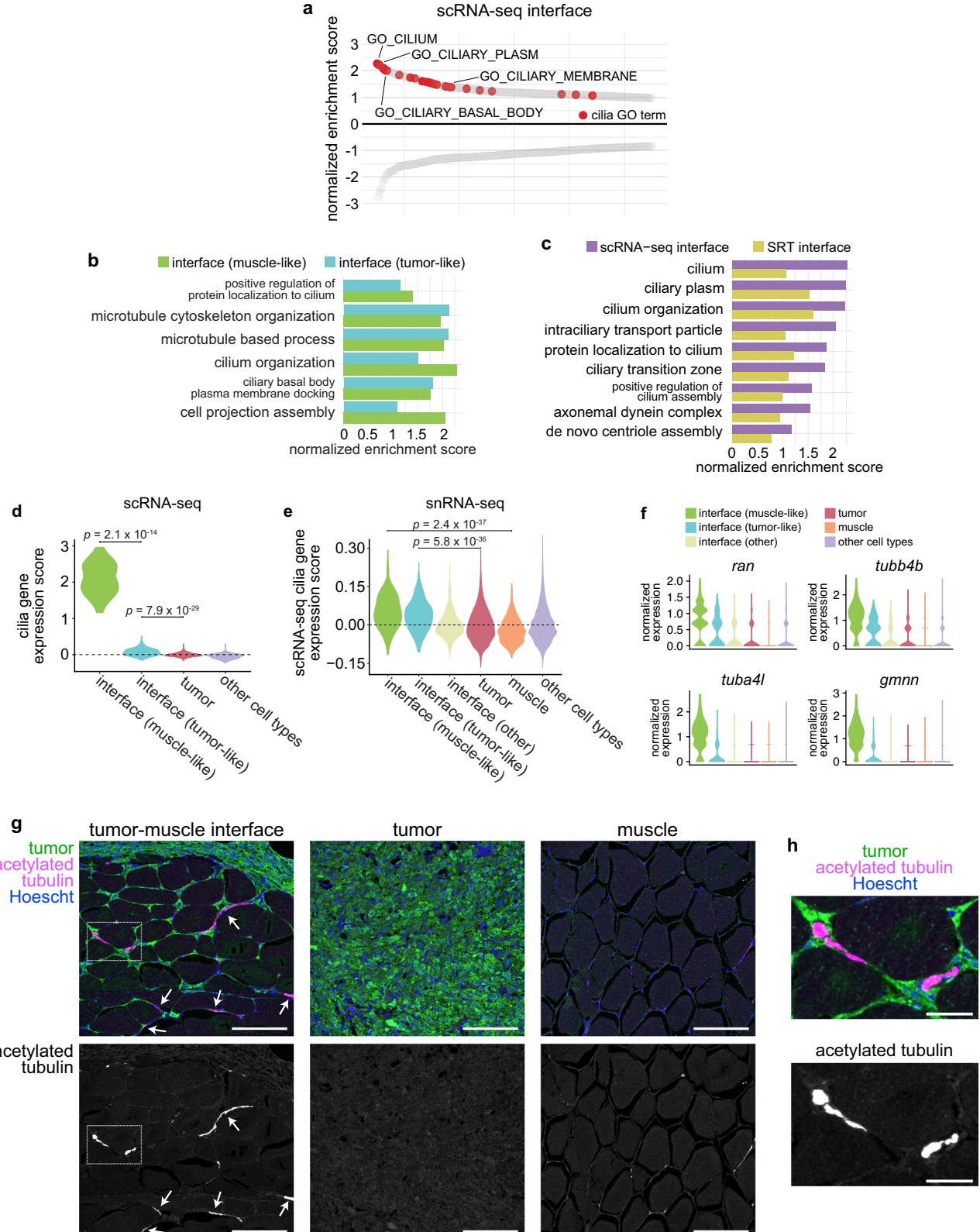

the microenvironment, where upregulation of cilia may contribute to tumor–microenvironment cell interactions.

**An interface signature may be present in human melanoma.** Finally, we investigated whether "interface"-like cells are also found in human melanoma. We chose to take advantage of a recently published scRNA-seq dataset of 29,247 cells isolated from 43 human patients with metastatic melanoma[39], as this dataset contains significantly more cells than other commonly used human metastatic melanoma scRNA-seq datasets[40]. This new dataset contains mostly tumor, myeloid, and immune cells[39] (Supplementary Fig. 14a). We scored each cell for expression of the human orthologs of the genes upregulated by more than 1.5-

**Fig. 5 Cilia genes and proteins are enriched at the tumor–microenvironment interface. a** Waterfall plot showing the top and bottom 250 GO cellular component terms by normalized enrichment score (NES) in the scRNA-seq interface cluster, with cilia GO terms labeled in red. **b** Cilia-related GO term enrichment scores within the tumor-like and muscle-like interface cell states from the scRNA-seq dataset. **c** Cilia-related GO term enrichment scores for the scRNA-seq and SRT interface clusters. **d** Relative expression of fish SYSCILIA genes in the scRNA-seq interface cluster. **e** Relative expression of the top 25 SYSCILIA genes upregulated in the scRNA-seq interface cluster across the snRNA-seq clusters. **d**, **e** p-values are noted (Wilcoxon rank sum test, two-sided, with Bonferroni's correction). **f** Normalized expression of selected cilia genes across the snRNA-seq interface clusters. **g** Immunofluorescent images of sections through adult zebrafish with invasive melanomas, stained for GFP (tumor cells), acetylated tubulin (cilia), and Hoescht (nuclei), showing the tumor-muscle interface (left), center of the tumor (middle), and distant muscle (right). Arrows denote cilia at the interface. Scale bars, 100 μm. Images are representative from at least three independent experiments. **h** Inset of region highlighted in **g** (left). Scale bars, 25 μm.

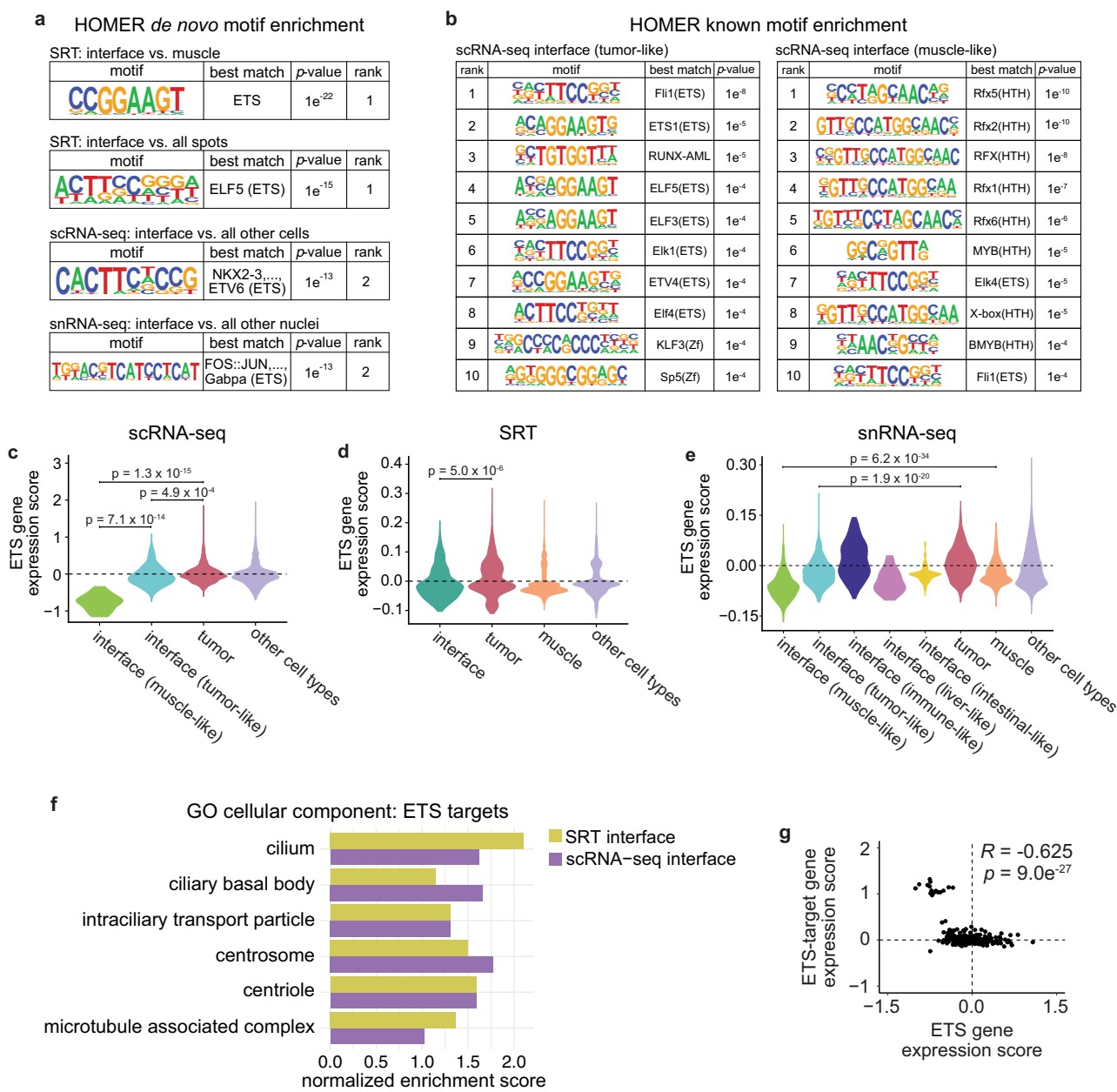

**Fig. 6 ETS transcription factors may regulate cilia gene expression at the interface. a** Results from HOMER de novo motif analysis of differentially expressed genes in the SRT, scRNA-seq, and snRNA-seq interface clusters. **b** Top ten enriched motifs from HOMER known motif analysis of the scRNA-seq tumor-like (left) and muscle-like (right) interface cell states. **a**, **b** p-values calculated using the hypergeometric test (one-tailed). **c–e** Relative expression of zebrafish ETS genes across the clusters in the scRNA-seq (**c**), SRT (**d**), and snRNA-seq (**e**) datasets. p-values are noted (Wilcoxon rank sum test, two-sided, with Bonferroni's correction). **f** Normalized enrichment score of cilia-related pathways enriched in ETS-target genes within the SRT and scRNA-seq interface clusters. **g** Scatter plot comparing ETS gene expression scores per cell and ETS-target gene expression scores per cell in the scRNA-seq interface cluster. Pearson's correlation coefficient (R) and p-value of correlation (two-sided) is indicated.

fold in our scRNA-seq interface cluster (Supplementary Fig. 14b), and classified cells that upregulated these genes as an "interface" population (Supplementary Fig. 14c, d). Similar to our snRNA-seq results, "interface" cells were found across all the major cell types in the human melanoma dataset (Supplementary Fig. 14b–d, g). For the purposes of statistical power, we focused on interface-like cells from the three largest clusters (tumor, myeloid cells, and T/NK cells). Human cells in an interface-like cell state upregulated many of the same genes upregulated in the interface in our zebrafish datasets, including *PLK1* (Figs. 3e and 4b and Supplementary Fig. 14e, g), *HMGB2* (Supplementary Figs. 10, and 14e, g), *TUBB4B* (Fig. 5f and Supplementary Fig. 14e, g) and *TPX2* (Fig. 3e and Supplementary Fig. 14e, g). Cilia genes were significantly upregulated across all of the interface cell states, relative to their corresponding tumor/TME cell types (Supplementary Fig. 14f). This suggests that a transcriptionally distinct "interface" gene signature may be found in human melanoma. Identifying which human melanoma subtypes (e.g., BRAF, NRAS, c-KIT, etc) in which an interface cell state is found awaits larger datasets of freshly isolated tumors subjected to scRNA-seq and/or SRT. Follow-up analyses determining the roles of specific types of immune and myeloid cells in the interface would also be an interesting area of future study. Together, our results suggest that cell-cell interactions at the tumor–microenvironment interface are accomplished by a subset of specialized tumor and muscle cells, which together upregulate a conserved common gene program characterized by upregulation of cilia genes and downregulation of ETS transcription factors.

## Discussion

Here, we combined spatially resolved and single-cell and single-nucleus transcriptomics approaches to characterize how tumor cells interact with new tissues in their surrounding environment, revealing key regulators of how this interface is formed. We analyzed a total of 49,944 transcriptomes encompassing expression of 20,589 unique genes from 7281 spatial array spots, 2889 zebrafish cells, 10,527 zebrafish nuclei, and 29,247 human cells. Our results identified a series of spatially-patterned gene modules, some of which specifically localize to the interface between tumors and surrounding tissue. We showed that the interface is composed of specialized tumor and muscle cell states, which are distinguished by upregulation of cilia genes and proteins. We further show that ETS transcription factors regulate expression of cilia genes at the interface, and that a distinct "interface" cell population is conserved in human melanoma patient samples. Together, our results reveal an "interface" transcriptional state that may mediate melanoma growth into surrounding tissues.

Our results identify a role for ETS-family transcription factors in mediating cilia gene expression at the interface. In recent years cilia have been implicated in multiple facets of melanoma biology, but their role in melanoma progression is still unclear. The bulk of melanomas are not ciliated[30,31] (Fig. 5g, h and Supplementary Fig. 12), and in fact, the "ciliation index" is gaining prominence as a diagnostic tool to distinguish melanomas from benign nevi[28,32]. Furthermore, cilia disassembly has recently been implicated in melanoma metastasis[29], in which deconstruction of cilia, regulated by EZH2, drives metastasis. The paradox is that while most melanoma cells are not ciliated, many melanomas still express cilia genes (Fig. 5d, e). Our data adds a layer onto this complexity, in that we find that not only are cilia genes upregulated specifically at the interface between tumor and microenvironment, but more importantly that only cells at that interface express high levels of cilia proteins. This raises the still not fully answered question of what role cilia play in various steps in melanoma

progression. In primary melanoma growth, it is clear that most cells are unciliated, and that EZH2 acts to suppress those genes. Loss of cilia via EZH2 increases metastasis in these models via enhanced Wnt/β-catenin signaling[29]. Our finding that most melanoma cells do not have cilia is consistent with this finding, but yet we find a specific subset of cells at the interface that upregulate cilia genes and protein, and these cells appear to be present in human melanoma as well.

How to reconcile these seemingly conflicting pieces of data? Our data would suggest that intact cilia may be most important when they are first encountering new, heterotypic cell types in the neighboring environment. We can envision several different possibilities to why cilia are upregulated specifically at this interface. First, this upregulation of cilia genes and proteins at the interface may be transient, induced by heterotypic cell–cell interactions between tumor and muscle. Primary cilia are critical signaling hubs for the cell, and regulate signaling pathways such as Hedgehog and TGF-β/BMP[41], all of which are important in cancer progression[42] and cell–cell communication. Our NicheNet analysis (Supplementary Fig. 10) suggests that there may be distinct ligand/receptor pairs, including HMG family proteins, that may mediate such signaling. A second possibility is that the primary cilia are acting as mechanotransducers, and play a role in directional migration of the cells as they invade into new tissues. For example, seminal work on primary cilia demonstrated that cilia can orient in the direction of migration in 3T3 cells[43], which has been also seen in the context of wound healing[44,45]. Finally, it is possible that the emergence of cilia at the interface is actually acting as a barrier to systemic metastatic dissemination, and that heterotypic interactions between melanoma and muscle might be restraining progression. It is notable that our zebrafish melanomas metastasize at a low rate, and in fact skeletal muscle (where we most easily see the interface) is a rare site of metastasis in humans, consistent with this possibility. A major endeavor for future studies will be to delineate how cilia act at each step of tumorigenesis, what signaling nodes are most critical, and whether they act as a barrier or enabler of metastasis. Another open and related question is which microenvironment cell types (other than muscle) trigger ciliation of the tumor–microenvironment interface. Our snRNA-seq data (Fig. 4) and analyses of human patient data (Supplementary Fig. 14) suggest that the interface is not solely restricted to tumor and muscle, but that other cell types may also be reprogrammed to adopt this cell state, such as immune cells or liver cells. Recent work has suggested that in melanoma, tumor cells can reprogram microenvironment cells such as liver cells at a distance[8]. It is not clear yet whether direct physical contact between tumor/microenvironment cells is required to induce an interface-like cell state, or whether longer-range signaling mechanisms may also be at play, but determining the nature of these tumor–microenvironment interactions (whether metabolic or epigenetic) is an exciting area for future mechanistic study.

Our results uncovered a role for ETS-family transcription factors in melanoma, as potential transcriptional repressors of cilia genes. Although most ETS TFs can function as transcriptional activators, at least four ETS TFs are known to have repressor activity[46] (Supplementary Table 2). Despite the fact that ETS TFs have a well-characterized role in several types of solid tumors, their role in melanoma has not been studied in depth, although a recent study found that ETS TFs induce a UV damage signature that correlates with increased mutational burden in human melanoma[47]. ETS TFs broadly function in various facets of tumorigenesis, including DNA damage, metabolism, self-renewal, and remodeling of the microenvironment[38]. However, most if not all of these situations have been found to be induced by aberrant upregulation of ETS genes. Conversely, we found a

role for downregulation of ETS TFs specifically where tumors contact surrounding tissues. It is still unclear what triggers this downregulation of ETS genes in such a spatially-restricted region. Despite their role as transcription factors, ETS proteins also participate in a wide range of protein–protein interactions, and their activity is regulated through phosphorylation as a result of signaling cascades[48]. MAPK signaling has been reported to regulate ETS[49], and the MAPK pathway is frequently activated in melanoma[50]. It is unclear if MAPK or other signaling pathways display spatially-restricted patterns of activation within tumors and/or the microenvironment, but the advent of SRT techniques will help to address these questions.

Although it was not a focus of our study, our SRT dataset also uncovered spatially-organized transcriptomic heterogeneity within the tumor itself (Fig. 1f, g). In recent years, the advent of single-cell transcriptomics approaches has identified a substantial degree of transcriptomic heterogeneity in most if not all types of cancer[51]. Tumor heterogeneity often increases as tumors progress, and may be a predictor of poor clinical outcomes as it is believed to be a major contributor to drug resistance[52]. Investigating the underlying cause and complex clonal relationships within different tumor cell subtypes has proven to be challenging for many reasons, one of which being a lack of information regarding the spatial patterning of this heterogeneity. Our dataset acts as a proof-of-principle for the use of spatially resolved transcriptomics in identifying spatially-organized tumor heterogeneity, and lays the groundwork for future studies using our dataset or others to explore the basis of this spatial patterning.

Our study is, to our knowledge, the first spatially-resolved gene expression atlas of the interface between the tumor and its environment. Although we uncovered many genes, pathways and gene modules that are spatially patterned within the tumor and/ or environment, there are likely many more interesting biological phenomena in our dataset that we have yet to identify. Recently, deep learning methods have been applied to histopathology images to uncover spatially-resolved predictions of molecular alterations, mutations, and prognosis[53,54]. A logical next step would be extension of these approaches to integrate deep learning and pattern recognition algorithms with SRT data, to identify interesting spatial patterns of gene expression and also predict transcriptomes based on histopathology. Ultimately, integration of transcriptomics, histopathology, and deep learning techniques will allow us to expand the utility of both SRT and histological datasets and broaden our understanding of cancer cell interactions in vivo.

## Methods

**Zebrafish husbandry.** Zebrafish lines were maintained at 28.5 °C in a dedicated aquatics facility with a 14 h on/10 h off light cycle. *casper*[55] fish were used for all experiments. Fish were anesthetized with Tricaine (MS-222) at a stock concentration of 4 g/L (pH 7.0). All zebrafish experiments and procedures were carried out in compliance with institutional protocols for vertebrate animals, and were approved by the Memorial Sloan Kettering Cancer Center IACUC (protocol #12-05-008).

**Generation of transgenic fish.** Transgenic tumor-bearing fish were generated using the *miniCoopR* system as previously described[56,57]. Briefly, *casper* fish with the genotype *mitfa-BRAF*[V600E]; *p53*[−/−]; *mitfa*[−/−] were incrossed, and the resulting 1-cell stage embryos were injected with plasmids containing *mitfa-MITF* and *mitfa-GFP*. Fish were raised to adulthood (4–6 months) and screened for the presence of pigmented, GFP-positive tumors.

**Generation and validation of ZMEL-cilia cell line.** The mouse ARL13B coding sequence was PCR amplified (removing the stop codon) from the plasmid pENTR-Arl13b2 (Addgene #40871) and subcloned into a middle entry vector containing a C-terminus EGFP tag. Primer sequences can be found in Supplementary Table 3. LR cloning was subsequently performed using 5′ entry ubi promoter, middle entry ARL13B-EGFP, and 3′ entry SV40 fragment. To generate the cell line, 8 million ZMEL1 cells[58] were electroporated with 15 μg of the ubi-ARL13B-EGFP plasmid

using the Neon Transfection System (Thermo Fisher Scientific). Following electroporation, cells were allowed to recover for 72 h, and then GFP+ cells were isolated using a BD FACSAria III Cell Sorter (BD Biosciences). To validate localization of ARL13B-GFP to the cilium and acetylated tubulin staining, cells were grown on chamber slides before fixation with 4% PFA for 15 min at RT. Cells were then washed with PBS before permeabilization with 0.1% Triton X-100 in PBS for 30 min at RT. Cells were blocked for 1 h at RT with 10% goat serum, before incubation overnight at 4 °C with goat anti-GFP (abcam #ab5450, 1:100) and mouse anti-acetylated tubulin (Sigma-Aldrich #6793, 1:100). Secondary antibodies (anti-goat IgG conjugated to Alexa 488 and anti-mouse IgG conjugated to Alexa 555) were used at 1:250 for 2 h at RT. Slides were mounted in Prolong Glass (Thermo Fisher Scientific) and imaged on a Leica SP5 upright line scanning confocal microscope.

### Spatially resolved transcriptomics
*Sample preparation.* Adult tumor-bearing fish were euthanized on ice and washed in 1× PBS. After dissection of the head and tail, the remaining tissue was equilibrated in cold OCT for 2 m, before transfer to a tissue mold filled with fresh OCT for snap-freezing in liquid nitrogen-chilled isopentane. Tissue blocks were stored at −80 °C. For cryosectioning, both the tissue block and the Visium slide were equilibrated inside the cryostat for 15–30 m at 16 °C before sectioning. Transverse sections through the entire fish were cut at a thickness of 10 μm and immediately placed on the Visium array slide (Visium Spatial Gene Expression slides, 10× Genomics). Array slides containing sections were stored at −80 °C for a maximum of 1 week before use.

*Fixation, staining, imaging, and construction of cDNA libraries.* Samples were processed according to the Visium Spatial Gene Expression User Guide (10× Genomics) and all reagents were from the Visium Spatial Gene Expression Kit (10× Genomics). Briefly, sections were fixed in chilled methanol for 30 min at −20 °C, stained with hematoxylin and eosin, and mounted in 85% glycerol for imaging. Imaging was performed on a Leica SCN400 F whole-slide scanner at ×40 magnification. After imaging, sections were permeabilized at 37 °C for 45 m. After permeabilization, the on-slide reverse transcription (RT) reaction was performed at 53 °C for 2 h. Permeabilization time and RT reaction length were determined using the Visium Spatial Tissue Optimization Kit (10× Genomics). Second strand synthesis was subsequently performed on-slide for 15 m at 65 °C. All on-slide reactions were performed in a thermocycler with a metal slide adapter plate. Following second strand synthesis, samples were transferred to tubes for cDNA amplification and cleanup. Library quality was assayed using a Bioanalyzer High Sensitivity chip (Agilent).

*Sequencing.* 10× Genomics Visium libraries were pooled, denatured, and diluted to a loading concentration of 1.8 pM with 1% PhiX control, followed by paired-end sequencing on an Illumina NextSeq 500 to a depth of approximately 110–180 million paired reads per sample. Sequencing parameters: Read1 28 cycles. i7 10 cycles, i5 10, Read2 120 cycles. Sequencing data was processed using the Space Ranger pipeline v.1.0.0 (10× Genomics).

*Dimensionality reduction and clustering.* SRT data was processed using R version 3.6.3, Seurat version 3.1.4[17], Python version 3.6, and MATLAB 2019b. Data was normalized using SCTransform[59]. The three SRT datasets were integrated using the Seurat SCTransform integration workflow, using 3000 integration features and including all common genes between the three datasets. Principal component analysis[60] and UMAP dimensionality reduction[61] were done using default parameters. Initial clustering was done using the FindClusters function implemented in the Seurat R package with the resolution parameter = 0.8. Tissue types of each cluster were inferred and clusters were further refined by plotting clusters onto the associated histology images and identifying marker genes using the Wilcoxon's Rank Sum test. Expression scores for ETS and cilia gene sets were calculated using the Seurat function AddModuleScores with default parameters. A list of cilia genes was obtained from the SYSCILIA gold standard list[33]. A list of ETS genes was obtained from ref. [62].

*Identification of genes enriched in the SRT interface.* To identify genes that were enriched at the interface in the SRT data, we first used the Seurat function FindMarkers and the Wilcoxon rank sum test in order to calculate the average log2 fold change for each gene in our dataset within the interface cluster, relative to all other SRT array spots. We then used the same function to calculate the average log2 fold change of each of these genes within the tumor and muscle clusters. To account for the likely admixture of tumor and muscle cells within the interface region, we defined interface-upregulated genes as: genes with a log2 fold change > 0, log2 fold change in the interface > log2 fold change in the tumor, and log2 fold change in the interface > log2 fold change in the muscle. We defined interface-downregulated genes as: genes with a log2 fold change < 0, log2 fold change in the interface < log2 fold change in the tumor, and log2 fold change in the interface < log2 fold change in the muscle. Finally, we filtered the lists of genes upregulated and downregulated in the interface to only include genes with an adjusted *p*-value of <0.05.

*Non-negative matrix factorization (NMF).* After normalization and integration of SRT data (see "Dimensionality reduction and clustering" section), negative values in the integrated expression matrix were set to zero. NMF was performed with a rank of 11. The optimal number of ranks was estimated using the function nmfEstimateRank[63] based on the first rank for which cophenetic starts decreasing[64] and for which RSS presents an inflection point[65]. Factor scores were first z-scored across factors prior to plotting onto array spots.

*Analysis of gene ontology (GO) terms with spatially coherent expression patterns.* GO term annotations for *Danio rerio* were downloaded from Biomart[66]. For each GO term, the average expression of genes annotated for that GO term was computed. We defined spots that highly express this GO term as spots, whose expression level for these genes is above the mean plus two standard deviations (we required the number of these spots to be at least five to proceed with the analysis). We then computed the Euclidean distance between these spots. Next, we computed the Euclidean distance between the same number of random spots, and repeated this computation 100 times to generate a null distribution of distances. We then compared the GO term spot distances to the null distribution using Wilcoxon's rank sum test to compute a *p*-value.

*Correlation between SRT spots and SRT clusters.* For computing the correlation across SRT clusters, we first computed the average expression of each tissue cluster in the integrated expression matrix of our three datasets. We then used the union of the ~1000 variably expressed genes in each individual dataset to obtain a list of ~2300 total variably expressed genes. We then used these genes to compute the Pearson's correlation and associated *p*-values.

*GSEA and pathway analysis.* Lists of differentially expressed genes for pathway analysis were created using the Seurat function FindMarkers using the Wilcoxon rank sum test. Ribosomal genes and genes with *p*-values above 0.05 were removed. Zebrafish genes were converted to their human orthologs using DIOPT[67], keeping only human orthologs with a DIOPT score >6. In cases where there were multiple zebrafish orthologs for one human gene, the gene with the highest log fold change in expression was used. Pathway analysis and GSEA[68] was done using the fgsea R package[69], using the MSigDB[70] GO[71,72] biological processes and GO cellular component human genesets.

*HOMER motif analysis.* Motif analysis was performed using HOMER[34], using the function findMotifs.pl. Motifs of lengths 8, 10, and 16 were queried within +/− 500 bp of the TSS of differentially expressed genes. Target genes containing the motif of interest were found by filtering the list of differentially expressed genes to contain only those with the desired motif. JASPAR[73] was used to annotate motifs.

*Multimodal intersection analysis (MIA).* To determine cell type enrichment in tissue regions we used MIA[14], which uses the hypergeometric cumulative distribution to determine the statistical significance of the overlap between cell type specific gene sets and tissue region specific gene sets. We used the intersect between all genes in the SRT count matrix and all genes in the snRNA-seq count matrix as the gene background to calculate the *p*-value. In parallel, we tested for cell type depletion by computing $-\log10(1 - p)$.

**Single-cell RNA-seq**

*Sample preparation.* Adult tumor-bearing fish were dissected to obtain only the tumor and surrounding tissues (i.e., head and tail were removed). Tissue was minced with a fresh scalpel and incubated in 0.16 mg/mL liberase (Sigma-Aldrich #5401020001) in 0.9× PBS for 15 m at RT. Tissue was then further dissociated by repeated pipetting with a wide-bore P1000, followed by incubation for an additional 15 m at RT. After adding 500 μL FBS to stop the dissociation reaction, samples were filtered through a 70 μm filter and centrifuged at 500 × g for 5 m at RT. The resulting pellet was resuspended in DMEM supplemented with 2% FBS, and cells were sorted at room temperature to remove debris and doublets, using a BD FACSAria III cell sorter (BD Biosciences). Equal numbers of GFP+ (tumor) and GFP− (microenvironment) cells were collected.

*Cell encapsulation and library preparation.* Equal numbers of sorted GFP+ (tumor) and GFP− (microenvironment) cells were centrifuged at 300×g for 5 m at RT, and resuspended in DMEM + 10% FBS. Droplet-based scRNA-seq was performed using the Chromium Single Cell 3′ Library and Gel Bead Kit v3 (10× Genomics) and Chromium Single Cell 3′ Chip G (10× Genomics). Approximately 10,000 cells from two fish were split encapsulated in a single v3 reaction. GEM generation and library preparation were performed according to manufacturer's instructions.

*Sequencing.* 10× scRNA-Seq libraries were pooled, denatured, and diluted to a concentration of 1.8 pM with 1% PhiX prior to paired-end sequencing on a NextSeq 500. Each library (corresponding to approximately 5000 cells) were sequenced to a depth of 550 M paired-end reads. Sequencing parameters: Read1 28 cycles, index read 8 cycles, Read2 132 cycles. Sequencing data was aligned to our reference zebrafish genome using Cell Ranger v5.0.1 (10× Genomics).

*Analysis.* Data was processed using R version 3.6.3 and Seurat version 3.1.4[17]. Cells with fewer than 200 unique genes or >20% mitochondrial reads were filtered out. Expression data was normalized using SCTransform[59]. Datasets were integrated using the Seurat SCTransform integration workflow, with 3000 integration anchors and including all genes expressed in both datasets (15,154 genes). Principal component analysis[60], UMAP dimensionality reduction[61], HOMER analysis[34], GSEA, and pathway analysis were performed as described above. Cluster annotations were performed using the Seurat function FindAllMarkers, in conjunction with marker genes used in previous analyses[74]. Doublets were detected using the doubletFinder R package[22], using 15 principal components.

**Single-nucleus RNA-seq**

*Sample preparation.* Adult tumor-bearing fish were dissected to obtain only the tumor and surrounding tissues (i.e., head and tail were removed). The tissue was then ground in a Dounce homogenizer on ice in Nuclei EZ Prep Lysis Buffer (Sigma-Aldrich #NUC101). The nuclear suspension was then spun down at 4 °C (500×g, 5 min). After resuspending the pellet in 1 mL wash buffer (250 mM sucrose, 50 mM citric acid, 1% BSA, 20 mM DTT, 0.2U/μL RNAse inhibitor), the sample was again spun at 4 °C for 5 min at 500×g. The pellet was resuspended in 1 mL wash buffer and subsequently sorted at 4 °C to isolate individual nuclei, using a BD FACSAria III cell sorter (BD Biosciences). Approximately equal numbers of GFP+ (tumor) and GFP− (microenvironment) nuclei were collected.

*Cell encapsulation and library preparation.* Equal numbers of sorted GFP+ (tumor) and GFP− (microenvironment) nuclei were centrifuged at 600×g for 5 m at 4 °C. Droplet-based snRNA-seq was performed using the Chromium Single Cell 3′ Library and Gel Bead Kit v3 (10× Genomics) and Chromium Single Cell 3′ Chip G (10× Genomics). Approximately 12,000 nuclei were encapsulated in a single v3 reaction. GEM generation and library preparation were performed according to manufacturer's instructions.

*Sequencing.* 10× snRNA-Seq libraries were pooled, denatured, and diluted to a concentration of 1.8 pM with 1% PhiX prior to paired-end sequencing on a NovaSeq 6000. Sequencing parameters: Read1 26 cycles, Read2 70 cycles, index read 8 cycles. Sequencing depth was approximately 200 million reads per 10,000 nuclei. Sequencing data was aligned to our reference zebrafish genome using CellRanger (10× Genomics).

*Analysis.* Data was processed using R version 3.6.3 and Seurat version 3.1.4[17]. Nuclei with fewer than 200 unique genes, more than 1 million UMIs, predicted doublets[22] and/or >20% mitochondrial reads were filtered out. A putative erythrocyte cluster was also filtered out for quality control reasons, due to the unusual nature of zebrafish erythrocyte nuclei[75]. Expression data was normalized using SCTransform[59]. PCA[60], UMAP[61], and HOMER analysis[34] were performed as described above. Potential doublets were detected with doubletFinder[22] and were filtered out before downstream analyses. Cluster annotations were performed using the Seurat function FindAllMarkers, in conjunction with marker genes used in previous analyses[74]. Modeling of ligand–receptor interactions was performed using NicheNet and the nichenetR R package[23], with the combined interface cluster as the "sender" cell population and all other cells as "receiver", using a cutoff of 0.1 for determining expressed genes and 0.5 for ligand-target scores. For NicheNet analysis, Zebrafish genes were converted to human as described above, using DIOPT[67], keeping only human orthologs with a DIOPT score >6. In cases where there were multiple zebrafish orthologs for one human gene, the gene with the highest log fold change in expression was used.

**Calculation of an interface gene signature.** Genes significantly upregulated in the interface clusters of the SRT, scRNA-seq, and snRNA-seq datasets were calculated using the Seurat function FindMarkers and the Wilcoxon rank sum test. Ribosomal genes (starting with "rps" or "rpl") were filtered out. The three genelists were then merged to only include common genes present on all three lists.

**Immunofluorescence and imaging.** Adult *casper* zebrafish with large pigmented tumors were euthanized on ice and fixed in 4% paraformaldehyde in PBS for 72 h at 4 °C. Fish were then stored in 70% ethanol before embedding in paraffin and sectioning by Histowiz, Inc. FFPE slides were deparaffinized in xylene before several rounds of washing in 100–50% ethanol. Antigen retrieval was performed by heating slides to 95 °C for 20 m in 10 mM sodium citrate pH 6.2 in a pressure cooker. After cooling, slides were blocked in a solution of 5% donkey serum, 1% BSA, and 0.4% Triton-X100 in PBS for 1 h at room temperature, before overnight incubation with primary antibodies in blocking buffer. Primary antibodies used were: goat anti-GFP (abcam #ab5450, 1:200) and mouse anti-acetylated tubulin (Sigma-Aldrich #6793, 1:100). Following overnight incubation with primary antibodies, slides were washed in PBS before incubation with secondary antibodies for 2 h at room temperature. Secondary antibodies used were: donkey anti-goat IgG conjugated to Alexa 488 (Thermo Fisher Scientific #A11055, 1:250) and goat anti-mouse IgG conjugated to Alexa 555 (Cell Signaling Technology #4409S, 1:250). Hoechst 33342 (Thermo Fisher Scientific #H3570) was added to the secondary antibody solution at 1:1000. Slides were mounted in ProLong Glass (Thermo Fisher

Scientific #P36980) and cured overnight at room temperature. Slides were imaged on a Leica SP5 upright line-scanning confocal microscope using a 40× (oil) objective. Twelve-bit Z-stacks were acquired at 0.3–0.5 μm steps using 3× line averaging. Maximum intensity projections were created of the Z-stacks in ImageJ. Noise was removed from each image using the ImageJ "Despeckle" function.

**Re-analysis of Smalley et al. human melanoma scRNA-seq dataset**. The counts matrix was obtained from GEO (GSE174401). All analysis was done using R version 3.6.3 and Seurat version 3.1.4[21], based on the analyses done in the original publication[39]. A Seurat object was created with default parameters, keeping all genes expressed in three or more cells and all cells expressing 200 or more genes, and filtering out any cells with more than 20% expression of mitochondrial genes, resulting in a final object containing 29,247 cells. Dimensionality reduction and clustering were performed using 15 principal components. Expression scores for interface, cilia, and ETS genes were calculated using the Seurat function AddModuleScore with default parameters. Interface marker genes were defined as the human orthologs of all genes with a log fold change >1.5 in our zebrafish scRNA-seq interface cluster.

**Statistical analysis**. Statistical analysis and figure generation were performed in MATLAB (Mathworks, R2019a) and R (R Foundation for Statistical Computing, 3.6.3). Image processing and analysis was performed in MATLAB and ImageJ (NIH). Unless otherwise noted, p-values were calculated using the Wilcoxon rank-sum test, two-sided, with Bonferroni's correction for multiple groups as necessary (R functions wilcox.test and pairwise.wilcox.test). Pearson correlation coefficients and corresponding p-values were calculated using the R function cor.test.

**Reporting summary**. Further information on research design is available in the Nature Research Reporting Summary linked to this article.

## Data availability

The scRNA-seq, snRNA-seq, and SRT data generated in this study have been deposited to the Gene Expression Omnibus (GEO) under accession number "GSE159709". Human scRNA-seq data was obtained from GEO under accession code "GSE174401". All other relevant data supporting the key findings of this study are available within the article and its Supplementary Information files or from the corresponding author upon reasonable request.

## Code availability

All code used for analysis and plotting is available at https://doi.org/10.5281/zenodo.5512629[76].

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

## Acknowledgements

We thank S. Selvaraj and B. Dabovic from the NYU Experimental Pathology Core for technical assistance with imaging for Visium experiments, R. Luther and M. Hogan from the Maurano lab at NYU for assistance with sequencing, R. Chaligne and the MSK Single Cell Research Initiative for assistance with snRNA-seq, and the members of the Yanai and White labs for useful discussions and technical support. M.V.H. was funded by a postdoctoral fellowship from the Canadian Institutes of Health Research. J.M.W. was supported by a NIH Kirschstein-NRSA predoctoral fellowship (F30CA236442), a NIH predoctoral fellowship (T32GM008539) from the Cell and Developmental Biology Program at Weill Cornell Graduate School, and a NIH Medical Scientist Training Program grant (T32GM007739). R.M.W. was funded by grants from the Melanoma Research Alliance, The Debra and Leon Black Family Foundation, NIH Research Program grants R01CA229215 and R01CA238317, NIH Director's New Innovator Award DP2CA186572, The Pershing Square Sohn Foundation, The Mark Foundation, The Alan and Sandra Gerry Metastasis Research Initiative at Memorial Sloan Kettering Cancer Center, The Harry J. Lloyd Foundation, Consano, and the Starr Cancer Consortium.

## Author contributions

M.V.H., R.M., I.Y., and R.M.W. conceived the study. M.V.H. and R.M. performed all experiments and data analysis. J.M.W. generated transgenic fish. M.V.H. and R.M. wrote the manuscript and all authors provided feedback before submission.

## Competing interests

R.M.W. is a paid consultant to N-of-One Therapeutics, a subsidiary of Qiagen. R.M.W. is on the scientific advisory board of Consano, but receives no income for this. R.M.W. receives royalty payments for the use of the *casper* zebrafish line from Carolina Biologicals. M.V.H., R.M., J.M.W., and I.Y. declare no competing interests.

## Additional information

**Peer review information** *Nature Communications* thanks Craig Ceol, Naveed Ishaque and the other anonymous reviewer(s) for their contribution to the peer review this work. Peer reviewer reports are available.

