## [Peer Review File · Nature Communications]

Reviewers' Comments:

Reviewer #1:

Remarks to the Author:

Dear Authors,

I put forward my review of the work presented by Hunter and colleagues whereby they exploit scRNAseq and spatial gene expression profiling to investigate the cellular interaction in a zebrafish melanoma model and identify a novel "interface" transcriptional signal that is enriched for genes involved in cilia related terms. The authors compare this to their own scRNAseq data and find evidence of these interface cells. They believe that the transcriptional signature of the interface is negatively regulated by ETS family transcription factors. Finally the authors examine patient derived cell line models of melanoma and identify the interface signal.

The paper is concise and well written and presents a good use case of exploiting spatially resolved gene expression profiling although I find the limited number of samples worrying (3 for Visium, and 2 for scRNAseq).

I find the results of the study to be interesting, however I would recommend several major amendments before I would consider it to be suitable for publishing in Nature Communications. I hope that with the following points that I can explain my reasoning.

Major points

1) In my opinion, the use of the term "microenvironment" is incorrect. I believe that the classical definition of TME relates to cellular heterogeneity within the tumor mass and adjacent cells. This study does not investigate transcriptional nor spatial heterogeneity within the tumor mass, nor immune cells, or vascular structures. The extent to which the surrounding stromal cells constitute the TME are debateable, so the idea of the interface region being part of the TME are indeed reasonable. However the authors push the idea of every cell type identified in the Visium experiment be the TME (lines 76-77 postulate that every cell type is part of the TME), which I find to be an unreasonable definition. This misclassification of all cells that are outside of the tumor to be "microenvironment" are reflected in the title. Because of this I would recommend being more conservative about what is being studied (which to me is how the tumor remodels the transcriptional landscape of adjacent stromal cells or the stromal component of the TME) and refrain from describing all identified cell types as constituent parts of the TME. While this is a major point, it actually does very little to change the results or conclusions, but rather only improve the wording.

2) There needs to be better integration and analysis of the scRNAseq. The authors omit the obvious correlation analysis of the scRNAseq cluster and Visium cluster transcriptional signatures – this would be a much better indication of which scRNAseq cluster/cell types are associated with the Visium spot signatures. This would need to be complemented with more sophisticated analysis given that Visium spots are likely to contain signals for multiple cell types, justifying the use algorithms such as Steroscope by Andersson et al (<https://doi.org/10.1038/s42003-020-01247-y>) or Spotlight by Elosua et al (<https://doi.org/10.1101/2020.06.03.131334>). I would suggest using one of these algorithms to identify the cell mixtures that contribute to the interface (and other cell types) in the Visium data. For example, this might explain why the macrophage cluster also exhibits an intermediate ST interface gene expression score (fig 3c). Furthermore, the authors should perform receptor ligand-interaction analysis using one of the available tools (e.g. cellphoneDB, nichenetR, cellchat, icellnet), and pay particular attention to the relative number of potential signalling interactions of the interface cluster with other clusters in the scRNAseq data.

3) I do not believe in the usefulness of the human derived cell line scRNAseq analysis (section "Interface cells are conserved in human melanomas"), and if anything it makes the conclusions more questionable given that the models would be lacking an interface component (as there is no interface in the cell line models in Wouters et al 2020, and their analysis does not implicate major transcriptional). Despite questionable suitability of using the scRNAseq data of cell line models that are not interfacing with other cells and lack a microenvironment, the signature is apparent in also

all cells in, e.g., samples MM047, MM087 and even A375 (the established cell line model for melanoma). The conclusion of this section implies that the melanoma cell line models would also include muscle cells, which is not proven or shown by the authors (and impossible to prove for A375 as this is an established melanoma cell line, despite it exhibiting a high interface marker gene score). This analysis should be redone on scRNAseq dataset of resected melanoma samples, and the identification of the interface marker gene signature should be revised as to only be enriched in a reasonable subset of cells that would represent the cell type proportions of interface cells compared to the proportion of tumor cells.

4) Limited number of replicates. I appreciate that for the Visium there are 3 replicates, but there should be more for the scRNAseq. If the authors can provide at least 3 scRNAseq samples, and show that their signature of interface cells is present in all 3 with a relative abundance (compared to the tumor cell proportion) similar to the Visium results, then that should be sufficient.

Intermediate

5) The authors should show the scRNAseq UMAP in fig 3b for each of the samples to show that the interface cells are not just from 1 of the 2 samples.

6) There is limited explanation of the differences and similarities between the samples. Based on the NMF in fig 2d and S4, I believe that NMF factors 8 and 15 are more prominent in the interfaces for sample A and B, and factor 10 to be more prominent in C. Additionally, fig S1 shows that samples A and B exhibit bi-modality of expression of genes and UMI counts per spots, but this is less prominent in C.

7) The authors state that there is a depletion of muscle cell types in the scRNAseq data due to difficulties of profiling multinucleated cells, yet the interface cells were sequencable. Does this mean that the interface may consist of mono-nucleated muscle-like cells? If so, the authors should state this explicitly.

8) Can the authors hypothesize why the interface is limited to only muscle cells?

9) The analysis of genes with an ETS motif within 500 kb of the TSS is very likely going to identify ETS binding sites in enhancers, which do not always regulate the closest gene. The authors should limit the region to e.g. 5 kb to capture ETS binding sites that are more likely to be associated to the gene, or to resolve the "distal" ETS binding sites via known enhancer promoter interactions in Zebrafish (similar to GeneHancer in humans, although I am not familiar if such a resource exists for zebrafish).

10) The ETS family of TF is known to be mainly involved in transcriptional activation, however cases of negative regulation have been reported in the case of particular transcription factors. The authors should investigate and describe whether the enriched ETS TF motifs identified are indeed shown to be involved in repression to add weight to their belief that these motifs are indeed for repressors. E.g. From figure 5a ETV6 is described as a repressor in Uniprot, but ELF5 has been described to be an activator.

11) Line 105: what does "transcriptionally unique" mean? In the UMAP the interface cells are cluster next to the muscle cells, so this would implicate alternative cell states rather than distinct cell types. Based on this, I think the wording is too suggestive of something that is more different than it really is. It also bugs me that despite the interface cells being adjacent to the muscle cells in the UMAP, but that the correlation analysis in figure 2 contradicts the UMAP embedding. The authors should explain why this is possible, or to further investigate other UMAP analysis to be more consistent with the correlation analysis.

12) General use of "cell type" vs "cell state". This is still not resolved in the single cell field, and without strong evidence that you are dealing with novel cell types, I would encourage the use of the term "cell state", e.g. muscle vs brain would be cell type difference, but the subclustering of the interface cells in the scRNAseq data in fig 3d would be cell state

13) What's the rationale for 15 factors? Many of the factors seem redundant. Please check the usual metrics for optimal number of factors using a tool such as <https://nmf.r-forge.r-project.org/nmfEstimateRank.html>. In the methods text, it states that the rank is set to 10, but 15 factors are presented – this should be corrected.

14) The interpretation of results should be toned down. In many cases the authors claim that their observations "prove" their conclusions, in many cases without validation. These should be toned down.

15) The authors should compile their code used for analysis and producing figures and upload

them to a code repository (github, gitlab, etc).

Minor

16) I dislike the use of the term ST when the specific ST array is not being used. This becomes confusing when the authors refer to "ST spots" when instead the Visium assay is used and not the ST assay. I would much prefer that the authors rather introduce "Spatial resolved gene expression profiling" to describe the field (including a more generalised description to include single molecule and LCM methods along side plate/array based methods), and replace other instances of ST with Visium as appropriate.

17) In figure 1, the authors show that the tumor component to have more transcriptional activity. The authors should describe this in the manuscript, and explain whether this arises from tumor cells expression more transcripts, or form the tumor region to be more densely populated with cells.

18) Line 27, 31: Abbreviation "TME" used on line 31 but not described (line 27)

19) Line 37: missing comma after "by their nature"?

20) Line 39: this is not a study of "tumor invasion". The authors should use a more accurate term of what is being shown.

21) Line 49, 60: "in vivo" should be "in situ"

22) Line 143 – the enrichment analysis should be presented in a more systematic way such as a supplementary table with the enrichment score per cluster.

23) Line 146 – The use of "confirm" is not justified. The result points to the existence of 2 transcriptional expression patterns that have to still be confirmed. It's unlikely that these are cell types, so please describe them as cell states unless you have evidence.

24) Line 156 – the authors should choose more appropriate phrases to describe what is being shown – here the authors have not established this as a model for tumor invasion

Figures

25) 2C – the names of the genes are unreadable due to overlapping. Please use something like ggrepel to prevent overlap.

26) 2e – NMF factor 15 doesn't seem to correlate with the spots that were annotated as interface, but rather other regions. Can the authors explain what is represented in factor 15?

27) 3d – please indicate the interface cell sub clustering (using a small split or vertical dashed line)

28) 4d – I find this figure incomplete without the muscle and tumor cell types.

29) S1 – It would be nice to have the histology slide adjacent to the spot plots. Whats the explanation for bimodality of transcripts per spot? More transcripts per cell (in which case please show similar evidence in scRNAseq with UMI per cell), or more cells per spot (in which case please show DAPI stained slides and approximate the density of cells)

30) S2 – what are the grey spots? Why are these inconsistent with figure 1? I could not find a corresponding section to the un-integrated analysis.

31) S4 – please rotate and order images to match the previous rotation and presentation of samples A, B and C

Reviewer #2:

Remarks to the Author:

This manuscript utilizes spatial and single cell transcriptomics to uncover a distinct population of 'interface' cells where the tumor and microenvironment meet. The transcriptomes of these cells are analyzed to suggest cilia genes are specifically upregulated in interface cells and ETS transcription factors, which are downregulated in interface cells, normally repress cilia genes in these cells. The existence of an interface population with a unique transcriptome and functional characteristics would be a very interesting advance in melanoma and tumor biology in general. There are several issues in the current manuscript that should be addressed to validate the existence of the interface population and its unique transcriptome.

- Is there a distinct population of interface cells?

As the authors note, and Figure 2a confirms, the “interface” ST spots contain an admixture of tumor and microenvironment cells. As such, the tumor-like muscle spots and the muscle-like tumor spots seen in Fig1e could simply reflect the ratios of muscle and tumor cells found at these ST spots – the spots that cluster closer to muscle have a higher muscle:tumor cell ratio, and the ones that cluster closer to tumor have a lower muscle:tumor ratio. To address this possibility, the authors utilize scRNAseq to further interrogate the identity and possible heterogeneity of this interface population. In the scRNAseq they find two subpopulations of cells – one subpopulation that looks more like tumor and another that looks more like muscle. Should these be treated as a single interface population or separately as muscle and tumor populations in all subsequent analyses? The authors should strongly consider the latter. To do so there are two factors that need to be addressed:

a. The lack of a normal muscle cell population in the scRNAseq dataset impairs assessment of the signature found in the “interface (muscle)” population. Without this comparison it is difficult to assess if the ciliated signature seen in Fig3d is distinctive or also found in normal muscle – it is notable that many of the cilia markers are enriched in the interface (muscle) subpopulation compared to the interface (tumor) subpopulation. These assertions would be easily visualized in a new Fig3f containing muscle, interface (muscle), interface (tumor), and tumor clusters comparing expression of marker genes for each cluster. If the proposed “interface (muscle)” and “interface (tumor)” populations are transcriptionally unique, genes from the “upregulated in interface” signature should be commonly expressed in the scRNAseq interface, but not in the bulk muscle or tumor groups. A survey of the literature suggests that while difficult, it is technically feasible to isolate muscle cells via scRNAseq.

b. As noted above the interpretation of this “interface” signature rests in part on the identification of interface (muscle) cells. The two current scRNAseq datasets contain only 4 and 17 of these cells. To properly power the analysis of these groups, the authors should consider another scRNAseq run. Isolation of more “interface (muscle)” cells will enable properly powered downstream analysis of the two distinct interface signatures instead of pooling interface (muscle) and interface (tumor) cells together as in Fig4a-d.

- Are cilia genes upregulated in interface cells and are cilia present on these cells?

This question is related to the one above. The question in essence is whether the cilia signature identified in interface cells is a product of the interface (muscle) subpopulation and is this subpopulation different from normal muscle?

a. The identification of a ciliated interface population is predicated on the identification of an interface population expressing cilia gene markers. Interrogation of the GEO scRNAseq reveals that the ciliated gene signature is driven by only the 21 interface (muscle) cells captured. It appears that these cells, which appear transcriptionally distinct from the interface (tumor) cells, are driving the ciliated signature. As mentioned above, capture of a non-interface muscle population, would provide an adequate comparison to reinforce that these ciliated muscle cells only arise at the tumor interface.

b. Many of the stainings presented in Fig4 and SFig5 are confusing or inconclusive. In Fig4g it appears that AcTub is staining axonemes at the edges of some muscle fibers but isn't staining anything in the muscle fibers that are surrounded by tumor cells. Fig4h and Fig4i are completely different – there is now no axonemal staining but instead diffuse staining in the muscle fibers that are surrounded by tumor cells. In Fig4i it appears that Arl13b is staining cilia, and some appear to arise from tumor cells – but these are seen with the AcTub staining in the same panel. The gamma tubulin staining in Fig4h and Fig4i is very difficult to interpret – are the puncta supposed to be basal body staining in single cells? In SFig5a the Arl13b staining is strange and does not overlap with AcTub. The AcTub staining is somewhat consistent with the presence of primary cilia – a single puncta per cell, but without a high magnification image it is difficult to know. To sort this out the authors should confirm that the antibodies are appropriate for use in zebrafish with control samples and provide high quality high magnification images.

- How do ETS transcription factors regulate gene expression in interface cells?

The analysis of ETS regulation of interface cell genes raises the question of whether ETS factors are identified through their regulation of genes in interface (tumor) cells and whether the upregulation of cilia genes is linked to low expression of ETS factors.

a. As effectors of MAPK pathway signaling, ETS factors are expected to be active in tumor cells. In the analyses presented in Fig5a, each of the interface groups in which the ETS motif is enriched

has a higher proportion of tumor cells. To what extent is the identification of ETS motifs in Fig5a driven by upregulation of ETS motif genes in interface (tumor) cells compared to the downregulation of ETS motif genes in interface (muscle) cells? It would be helpful to see motif enrichment analysis performed by treating the interface (tumor) and interface (muscle) subpopulations separately.

b. A model that emerges is that lower expression of ETS factors in interface cells causes derepression and upregulation of cilia genes in these interface cells. The rationale for the model seems to stem from the lower expression of ETS factors in tumor cells. However, the upregulation of cilia genes is predominantly observed in interface (muscle) cells. This seems inconsistent – how would lowered expression of ETS factors in tumor cells cause derepression in interface (muscle) cells?

Other issues:

1. The authors should consider the possibility of doublets contributing to the “interface” cell population. As the authors note, spatial transcriptomics lack the resolution required to rule out multiple cells contributing to a spot signature. The authors should provide supplemental plots from their scRNAseq data showing “nCounts_RNA” and “nFeatures_RNA” to reinforce that these interface transcriptomes are not artifacts from muscle/tumor doublets.
2. Figure 2D looks less convincing than NMF 8 and 10 found in the supplement, the authors should consider picking a different NMF for display in the main figure. Additionally, highlighting an NMF showing an enrichment of cilia genes in the interface region vs muscle would contribute to the argument that cilia genes are specific to the interface.
3. The authors should consider showing feature plots of SYSCILIA genes (or a composite score) to show that this signature is enriched in just the interface population and not other cell types. Furthermore, a SYSCILIA feature plot would reveal if the ciliated signature is broadly expressed over both interface tumor and interface muscle cells, or just one population.
4. In Figure 3b/c the point overlap obscures the red/blue expression score, the authors should adjust the point size for each cell for easier visualization of the ST interface score.
5. The point made in Figure 3e could be more cohesively made with a zoomed in feature plot of Figure 3b where the muscle cells appear to cluster as a small independent island next to the “interface (tumor)” cells.
6. The fold change between “interface (tumor)” and “tumor” does not appear significant in the current figure. The authors should state in the figure legend what the p-value represents.
7. The authors should reconcile their motif analysis methods and text for clarity. In text the authors state, “We queried the zebrafish genome for genes with an ETS motif within 500kb of the transcription state site...” whereas in the methods the authors state “...were queried within +/- 500bp of the TSS of differentially expressed genes.”
8. The authors should consider scoring their ETS and Ciliary signatures in the Tirosh 2016 scRNAseq dataset. The Tirosh data set is more analogous to the model system used by the authors as it contains melanoma and microenvironmental cells isolated from primary tumors.
9. The authors should fix the missing words in the sentence on line 413 so that it doesn’t hang onto reference 43 in the superscript.
10. The NMFs found in the supplement are hard to interpret. The orientation appears different from the slices found in the main figures. Furthermore, the resolution of the top GO terms in some NMF’s is low and hard to read (NMF 10/12). Correction of these issues would greatly facilitate interpreting the NMF results.

Reviewer #3:

Remarks to the Author:

This manuscript by Hunter and Moncada et al. describes a combined spatial transcriptomics and single cell RNA-seq analysis of a melanoma zebrafish model. The analysis is focused on the spatial interface between tumor and muscle regions, which is shown to express high levels of cilia genes. This cilia expression profile is suggested to be regulated by ETS transcription factors and potentially to promote invasion of melanoma to surrounding tissue.

The use of cutting edge methods - a combination of scRNA-seq and spatial transcriptomics - is a

potential strength of this work, although apart from the cilia story there are very limited results that could be considered as demonstrating the utility of this technology, thereby decreasing the significance of this point. The increased cilia expression in the tumor-muscle interface is an interesting and potentially important result. However, it suffers from two substantial limitations. First, I have serious concerns about the validity of the analysis and conclusions. More specifically, the extent to which the current analysis distinguishes between cilia expression by the tumor and muscle cells is unclear, and accordingly, I am currently not convinced that there is a high degree of interface-specific Cilia expression among the tumor cells, as described further below. Second, the significance of such cilia expression profile relates to the possibility that it facilitates melanoma invasion, but there are no follow up studies to support this point and to describe the exact influence of cilia genes on invasion.

Specific comments:

1. As the authors acknowledge, in the spatial transcriptomics data, the interface region contains a combination of tumor and muscle cells. It is therefore somewhat trivial that these regions might be identified as a cluster that is distinct from tumor-only and muscle-only regions, even if the transcriptome of individual cells in the interface region is identical to the corresponding cell types in other (non-interface) regions. Thus, an alternative to the author's interpretation of this cluster as describing a unique profile of cells in the tumor-muscle interface is that it is merely the combination of two distinct profiles. Fig. 2C is described as comparing the interface program to that of tumor AND muscle cells, but it is not clear to me what exactly that means (i.e. were equal proportions of muscle and tumor cells taken?). Instead, the authors should use a more rigorous approach to analyze the interface expression profile and identify genes that are more highly or lowly expressed in the interface region compared to the expected expression as defined by a combination of tumor and muscle cell, as defined by the corresponding "pure" regions. A quantitative approach that considers the expected expression of a combination of tumor/muscle cells (in various proportions) and possibly of other relevant cell types that might exist in the interface, would be appropriate; but this is perhaps computationally subtle and challenging. Alternatively, a very simple conservative approach would be to compare the interface expression to the maximum of each gene between tumor and muscle areas, thereby ensuring that any gene that comes up as upregulated cannot simply be explained by cell type mixture; similarly, downregulated genes may be defined as those lower in the interface than in the minimum of the two types of regions (tumor/muscle).

2. To aid in the interpretation of the spatial transcriptomics data, the single cell RNA-seq data was generated with the hope of defining all of the individual cell types. However, this data lacks muscle cells and therefore is less useful for the combined analysis than it was envisaged. The authors mention the size of multinucleated muscle cells as a potential reason for not obtaining them. Accordingly, it might be useful to attempt a single nuclei (as opposed to single cell) approach to isolate and profile those cells.

3. In the single cell data, the interface cluster is shown to contain two subclusters reflecting tumor and muscle cells. Yet, much of the analysis is done with this entire cluster as a single entity (i.e. ignoring the division into two cell types). This approach introduces a considerable confounder that makes it difficult to interpret the results. If the cluster contains two cell types then the clustering should be modified to separate them and all analyses should be done for the re-defined clusters that will have a more accurate annotation. The current approach of keeping the combined cluster would be expected to increase the consistency with the spatial data (in which the interface region is a combination of tumor and muscle cells) but such consistency reflects a suboptimal use of the single cell RNA-seq data in which the resolution of the analysis is in fact reduced to be lower than it should be.

4. Related to the above point about the single cell RNA-seq analysis: there is no distinct cluster of muscle cells; but the "interface" cluster, which clusters together with the tumor cells, has a subset which is distinguished by high expression of muscle genes. This pattern strongly suggests that at least some of the "interface" cluster may reflect doublets of tumor and muscle cells. This could explain why muscle cells are not captured on their own but are captured in the context of the interface cluster; it could also again explain why the interface scRNA-seq cluster has some

consistency with the spatial transcriptomics analysis. The possibility of doublets is a well known limitation in single cell RNA-seq analysis, and multiple methods have been developed to try to deal with it (albeit they still have limitations). I am surprised to see that the authors completely ignore this possibility, and do not mention doublets anywhere in the manuscript. The implication, along the same line as noted above for the spatial transcriptomics, is that it is not clear what parts of the results indeed reflect a unique profile of interface cells, and what part reflects the combined signal of tumor and muscle cells.

5. Fig. 4E demonstrates very clearly that cilia expression is dramatically higher in the muscle interface cells than in all other cells. This effect is extremely strong. In contrast, the same figure shows that the difference in cilia expression among tumor cells in the interface and non-interface is much smaller; it is still statistically significant, but seems biologically somewhat negligible compared to the former effect, and it is important to note that statistical significance is of limited value when the number of profiled cells is so large (i.e. any gene would be statistically significant when comparing two cell types with enough cells profiled from each cell type) and hence the focus in such analysis should be on effect-size rather than on statistical significance. Thus, cilia expression is very high in muscle cells, compared to tumor cells both in the interface and outside of the interface. It is then easy to imagine that all results related to cilia expression in the interface reflect the residual signal of muscle cells in the interface clusters of both spatial transcriptomics and single cell RNA-seq analysis, as described extensively in the previous comments. Overall, it seems problematic to focus this manuscript on an effect that is (A) barely noticeable (compare "interface(tumor)" to "tumor" in Fig. 4E), and (B) can be confounded by the presence of an effect that is an order-of-magnitude higher (compare "interface(muscle)" to all others in Fig. 4E).

6. The authors include analysis of existing single cell RNA-seq data from human cell lines to show consistency of the results in human melanoma. However, it would be much more relevant to examine data directly from human patients, especially given the manuscript focus on cell-cell interactions and interface regions that would not exist in cell lines; such patient data is available through multiple studies including those already referenced in the manuscript.

7. If the main conclusion of the work is the possibility that cilia expression has a role in invasion, then follow up experiments to support that would be extremely useful.

Reviewer #1:

1) In my opinion, the use of the term “microenvironment” is incorrect. I believe that the classical definition of TME relates to cellular heterogeneity within the tumor mass and adjacent cells. This study does not investigate transcriptional nor spatial heterogeneity within the tumor mass, nor immune cells, or vascular structures. The extent to which the surrounding stromal cells constitute the TME are debateable, so the idea of the interface region being part of the TME are indeed reasonable. However the authors push the idea of every cell type identified in the Visium experiment be the TME (lines 76-77 postulate that every cell type is part of the TME), which I find to be an unreasonable definition. This misclassification of all cells that are outside of the tumor to be “microenvironment” are reflected in the title. Because of this I would recommend being more conservative about what is being studied (which to me is how the tumor remodels the transcriptional landscape of adjacent stromal cells or the stromal component of the TME) and refrain from describing all identified cell types as constituent parts of the TME. While this is a major point, it actually does very little to change the results or conclusions, but rather only improve the wording.

We agree this needs better clarity. Conceptually, what we were trying to describe is the notion that tumor cells can interact with a significant number of other non-tumor cell types. In primary melanomas, these can include not only immune or vascular cells, but also other cells such as adipocytes or keratinocytes. During metastasis, they can then interact with entirely new cells such as glial cells or hepatocytes. Further compounding this complexity is the fact that cell-cell interactions are not always direct. For example, a recent study showed that melanoma cells can metabolically interact with liver cells “at a distance” even when they are not in direct contact (Naser, *Cell Metabolism* 2021 May 11;S1550-4131). Thus perhaps a better way of describing these interactions is that they can be “microenvironmental” (i.e. a direct interaction between a tumor cell and its immediate surrounding cell types) versus “macroenvironmental” (i.e. an indirect interaction between a tumor cell and other cells not in the immediate vicinity). The purpose of our study was to dissect the interaction of the melanoma cells with their immediate surrounding cells, which is why we used the term “microenvironment”. And while we see your point that using the phrase “stromal” cell instead of “microenvironment” cell might be useful, we are not entirely sure this is correct either. Stromal cells are generally defined as cells that give rise to the connective and supportive tissue of an organ, i.e. fibroblasts. In our study, the cell at the interface is a skeletal muscle cell, and we think it would be very confusing to call this a stromal cell. For these reasons, we think a better way of describing our study is to more precisely describe which it is we are studying, which is the nature of the cell-cell interactions between the tumor and its immediately surrounding microenvironment. By this logic, we think the title appropriately describes what we focused on in the paper, but felt this needed a more thorough explanation in the text. We have discussed each of these points in the introduction, and explained our rationale for the use of the word microenvironment in this setting to make it very clear what we are studying in the paper.

2) There needs to be better integration and analysis of the scRNAseq. The authors omit the obvious correlation analysis of the scRNAseq cluster and Visium cluster transcriptional

signatures – this would be a much better indication of which scRNAseq cluster/cell types are associated with the Visium spot signatures. This would need to be complemented with more sophisticated analysis given that Visium spots are likely to contain signals for multiple cell types, justifying the use algorithms such as Steroscope by Andersson et al (<https://doi.org/10.1038/s42003-020-01247-y>) or Spotlight by Elosua et al (<https://doi.org/10.1101/2020.06.03.131334>). I would suggest using one of these algorithms to identify the cell mixtures that contribute to the interface (and other cell types) in the Visium data. For example, this might explain why the macrophage cluster also exhibits an intermediate ST interface gene expression score (fig 3c).

As suggested, we used multimodal intersection analysis (MIA), a tool described in our recent publication (Moncada et al., *Nature Biotechnology* 2020) to better integrate our transcriptomics datasets and identify the potential cell types present within the interface region. For this analysis, we used the cell types present in our new snRNA-seq dataset to deconvolve the SRT interface region. We used our new snRNA-seq dataset for these analyses because this dataset included significantly more cells/nuclei and cell types than our scRNA-seq dataset (see **new Fig. 4** for an overview of our snRNA-seq data). Our MIA analysis suggested that the interface regions in our SRT dataset are enriched in cell types including muscle, macrophages, and tumor. The cell type that was most significantly enriched in the interface region was the muscle-like interface cell state, in accordance with the histology of our SRT samples that showed that the interface region closely resembles the surrounding muscle (**Fig. 2a**). We have added our MIA results to **new Fig. S7**.

Furthermore, the authors should perform receptor ligand-interaction analysis using one of the available tools (e.g. cellphoneDB, nichenetR, cellchat, icellnet), and pay particular attention to the relative number of potential signalling interactions of the interface cluster with other clusters in the scRNAseq data.

As this reviewer suggested, we performed receptor-ligand interaction analysis using nichenetR, modelling interactions between interface cells and all other cells in our snRNA-seq dataset. We used our snRNA-seq dataset for this analysis as opposed to our scRNA-seq dataset as our snRNA-seq dataset included many more nuclei than cells in our scRNA-seq dataset, as well as a greater breadth of cell/nuclei types. As the NicheNet model is currently designed to work with human genes, we performed this analysis with the human orthologs of the zebrafish genes in our dataset. We have added our NicheNet results and associated gene expression data as **new Fig. S8**.

The top ligand predicted to be active in interface nuclei was *HMGB2*, of which there are two zebrafish orthologs: *hmg2a* and *hmg2b*. These genes were highly expressed in the interface clusters across our snRNA-seq, scRNA-seq and SRT datasets. Interestingly, HMGB2 expression has been reported to be correlated with tumor aggressiveness (Kwon et al., 2010; Fu et al., 2018). The predicted receptors for HMGB2 were *AR*, *ITPR1* and *CDH1* (fish orthologs: *ar*, *itpr1a*, *itpr1b*, *cdh1*). Of these potential receptors, *cdh1* was the most highly expressed in

general across the 3 datasets. *cdh1* was expressed in various microenvironment cell types, including intestinal cells, keratinocytes, and also in some interface cell states.

cdh1 (E-cadherin) is a core component of adherens junctions along with α -catenin and β -catenin. Interestingly, HMGB2 and β -catenin have been reported to cooperate to promote melanoma progression (Mo et al., 2019). We feel that the relationship between chromatin modifier genes such as HMGB2 and cell surface proteins such as E-cadherin is an important area for future functional studies, particularly in the context of how cell-cell interactions modulate changes in gene expression at the tumor-microenvironment interface.

3) I do not believe in the usefulness of the human derived cell line scRNAseq analysis (section “Interface cells are conserved in human melanomas”), and if anything it makes the conclusions more questionable given that the models would be lacking an interface component (as there is no interface in the cell line models in Wouters et al 2020, and their analysis does not implicate major transcriptional). Despite questionable suitability of using the scRNAseq data of cell line models that are not interfacing with other cells and lack a microenvironment, the signature is apparent in also all cells in, e.g., samples MM047, MM087 and even A375 (the established cell line model for melanoma). The conclusion of this section implies that the melanoma cell line models would also include muscle cells, which is not proven or shown by the authors (and impossible to prove for A375 as this is an established melanoma cell line, despite it exhibiting a high interface marker gene score). This analysis should be redone on scRNAseq dataset of resected melanoma samples, and the identification of the interface marker gene signature should be revised as to only be enriched in a reasonable subset of cells that would represent the cell type proportions of interface cells compared to the proportion of tumor cells.

We have removed all analyses using the Wouters et al. dataset from the manuscript, and replaced them with an analysis of the Tirosh et al., 2016 scRNA-seq dataset which is comprised of tumor and microenvironment cells from human melanoma patients. These data are in **new Fig. S10**. Similar to our scRNA-seq and snRNA-seq datasets, we found evidence of cells in an interface-like state within both the tumor and microenvironment cell types in this dataset (**Fig. S10a-c**). Interface cells were detected by scoring each cell for expression of the human orthologs of genes upregulated by more than 1.5-fold in our scRNA-seq interface cluster. Interface cells were defined as cells with marker gene enrichment scores above 0.5, which we have clarified in **Fig. S10b**. We found that both the tumor-like and CAF-like interface cell clusters upregulated cilia genes and downregulated ETS genes, and that the two were significantly negatively correlated (further suggesting that ETS transcription factors may act as transcriptional repressors of cilia genes) ($R = -0.1462$; $p = 1.31 \times 10^{-23}$; **Fig. S10d-e**).

4) Limited number of replicates. I appreciate that for the Visium there are 3 replicates, but there should more for the scRNAseq. If they authors can provide at least 3 scRNAseq samples, and show that their signature of interface cells is present in all 3 with a relative abundance (compared to the tumor cell proportion) similar to the Visium results, then that should be sufficient.

For the original scRNA-seq experiment in the paper, we completed 2 half-reactions, and sorted cells from a total of 3 fish split across the two half-reactions. However, we acknowledge that we obtained a relatively low number of cells from our scRNA-seq experiment in general. Instead of doing additional scRNA-seq, to better address reviewer comments regarding sequencing of muscle cells, we instead performed single-nucleus RNA-seq (snRNA-seq). This would allow us to increase the number of cells/nuclei profiled, and also to capture nuclei from cell types that are difficult to encapsulate for scRNA-seq such as muscle. In a pilot experiment, we confirmed that tumor nuclei in our transgenic melanoma model are GFP+ and can be sorted via FACS. We then isolated nuclei from 3 different fish, sorted GFP+ (tumor) and GFP- (non-tumor) nuclei for each fish, and mixed relatively equal numbers of tumor and non-tumor cells for snRNA-seq. We attempted to label which cells came from which fish by hashing, using barcoded antibodies against the nuclear pore complex to multiplex the tumor and non-tumor nuclei isolated from each fish, for a total of 6 groups. This hashing strategy has not been validated for zebrafish nuclei; however, we reasoned that the nuclear pore complex is relatively well-conserved across species so there was some chance of success. Unfortunately, we obtained a high number of nuclei labelled with multiple hashing labels and thus had to remove a significant number of nuclei from our analyses of the hashing data. Although we were able to assign some nuclei to a specific hashing group using the Seurat function HTODemux, the hashing results were not in agreement with our clustering or gene expression data, as we would expect nuclei expressing GFP transcripts to be the same nuclei hashed as the GFP+ sorted group but did not observe this.

Thus, while the hashing did not allow us to definitively show the presence of an interface cluster in all 3 fish used for our snRNA-seq experiment, but we have shown the presence of an interface cluster in all 3 of our SRT samples (**Fig. S2**) and both of our scRNA-seq reactions (**new Fig. S5c,f**), we feel that it is likely that the interface cluster is found in all of our snRNA-seq samples as well.

5) The authors should show the scRNAseq UMAP in fig 3b for each of the samples to shown that the interface cells are not just from 1 of the 2 samples.

We have added the UMAP plots for both scRNA-seq reactions (separately) to **new Fig. S5c** and **S5f**.

6) There is limited explanation of the differences and similarities between the samples. Based on the NMF in fig 2d and S4, I believe that NMF factors 8 and 15 are more prominent in the interfaces for sample A and B, and factor 10 to be more prominent in C. Additionally, fig S1 shows that samples A and B exhibit bi-modality of expression of genes and UMI counts per spots, but this is less prominent in C.

In regards to the bimodality of genes and UMIs per spot in samples A and B, we feel this is likely due to the presence of more cells/spot in the tumor region relative to the other tissue types present in the sample (see response to this reviewer's point 29). We have added a **new Fig. S1f-g** with data about the number of UMIs per spot across the different tissue types in our SRT data, and images approximating the number of nuclei per Visium array spot. We generally captured fewer UMIs and genes per spot across the entire tissue for sample C (see **Fig. S1a-b**), likely due to technical reasons, so we feel this is why the bimodality of genes/UMIs per spot is dampened in this sample. We have added more explanation of the differences/similarities between samples to the text (lines 101-102).

7) The authors state that there is a depletion of muscle cell types in the scRNAseq data due to difficulties of profiling multinucleated cells, yet the interface cells were sequencable. Does this mean that the interface may consist of mono-nucleated muscle-like cells? If so, the authors should state this explicitly.

We have added a possible explanation for the presence of muscle-like interface cells to the text (lines 209-211).

8) Can the authors hypothesise why the interface limited to only muscle cells?

Our initial scRNA-seq and SRT experiments did indicate that the interface was mainly limited to muscle-like cells. Zebrafish melanomas frequently invade into muscle, which likely explains why we frequently see interface cells in a muscle-like state. However, in our new snRNA-seq data, we see that there are actually multiple interface-like cells in addition to muscle (i.e. liver-like, etc, **new Fig. 4**). Our MIA results also suggest that the interface region is composed of multiple cell types, including muscle, tumor, and immune cells (**new Fig. S7**). This is especially interesting considering recent work (Naser, *Cell Metabolism* 2021 May 11;S1550-4131) showing that melanomas can lead to reprogramming of liver cells "at a distance" even when not directly in contact. Thus while muscle-like interface cells are probably the most common based on the biological behavior of our tumor model, it is possible that other cells get transcriptionally rewired by the melanoma cells, which will be an area for future explanation.

9) The analysis of genes with an ETS motif within 500 kb of the TSS is very likely going to identify ETS binding sites in enhancers, which do not always regulate the closest gene. The authors should limit the region to e.g. 5 kb to capture ETS binding sites that are more likely to be associated to the gene, or to resolve the "distal" ETS binding sites via known enhancer

promoter interactions in Zebrafish (similar to GeneHancer in humans, although I am not familiar if such a resource exists for zebrafish).

This was a typo in the text and should have said 500 bp. We have fixed the typo in the text.

10) The ETS family of TF is known to be mainly involved in transcriptional activation, however cases of negative regulation have been reported in the case of particular transcription factors. The authors should investigate and describe whether the enriched ETS TF motifs identified are indeed shown to be involved in repression to add weight to their belief that these motifs are indeed for repressors. E.g. From figure 5a ETV6 is described as a repressor in Uniprot, but ELF5 has been described to be an activator.

We have added a supplemental table (**new Table S1**) clarifying the activator/repressor status for each zebrafish ETS-family transcription factor.

11) Line 105: what does “transcriptionally unique” mean? In the UMAP the interface cells are cluster next to the muscle cells, so this would implicate alternative cell states rather than distinct cell types. Based on this, I think the wording is too suggestive of something that is more different than it really is. It also bugs me that despite the interface cells being adjacent to the muscle cells in the UMAP, but that the correlation analysis in figure 2 contradicts the UMAP embedding. The authors should explain why this is possible, or to further investigate other UMAP analysis to be more consistent with the correlation analysis.

We apologize for not clarifying our definition of “transcriptionally unique”. In this context, we define this term to mean that cells in the interface region upregulate genes that are not upregulated in any other cell type present in the SRT/scRNA-seq/snRNA-seq datasets. This is also shown in the heatmaps in **Figs. 3e** (scRNA-seq data) and **new Fig. 4f** (snRNA-seq data). However, as this term may be confusing, we have changed all references to a “transcriptionally unique” cell state to “transcriptionally distinct”. In the UMAP in **Fig. 1e**, we note that interface spots are adjacent to both tumor and muscle spots (see the bottom left corner of the tumor cluster), which is in agreement with our scRNA-seq and snRNA-seq datasets showing that the interface consists of cells in tumor- or muscle-like states.

12) General use of “cell type” vs “cell state”. This is still not resolved in the single cell field, and without strong evidence that you are dealing with novel cell types, I would encourage the use of the term “cell state”, e.g. muscle vs brain would be cell type difference, but the subclustering of the interface cells in the scRNAseq data in fig 3d would be cell state

We agree that the use of “cell type” in this context may be misleading. We have changed “cell type” to “cell state” across the manuscript where appropriate.

13) What’s the rationale for 15 factors? Many of the factors seem redundant. Please check the usual metrics for optimal number of factors using a tool such as

<https://nmf.r-forge.r-project.org/nmfEstimateRank.html>. In the methods text, it states that the rank is set to 10, but 15 factors are presented – this should be corrected.

Thank you for this suggestion - we have now used `nmfEstimateRank` to determine the optimal number of factors. We have selected the number of factors based on the rank for which cophenetic starts decreasing (Hutchins et al., 2008) and for which RSS presents an inflection point (Frigyesi and Höglund, 2008).

Based on these results, we have selected 11 factors for our analyses because cophenetic began decreasing at factor 10, and the RSS inflection point occurred around factor 12. We have added a description of how we selected the number of factors to the Methods and corrected the stated rank to match the data.

14) The interpretation of results should be toned down. In many cases the authors claim that their observations “prove” their conclusions, in many cases without validation. These should be toned down.

We have removed any references to “proving” conclusions from the manuscript.

15) The authors should compile their code used for analysis and producing figures and upload them to a code repository (github, gitlab, etc).

We have compiled all code in the Github repository github.com/mvhunter1/Hunter_Moncada_2021.

16) I dislike the use of the term ST when the specific ST array is not being used. This becomes confusing when the authors refer to “ST spots” when instead the Visium assay is used and not the ST assay. I would much prefer that the authors rather introduce “Spatial resolved gene expression profiling” to describe the field (including a more generalised description to include

single molecule and LCM methods along side plate/array based methods), and replace other instances of ST with Visium as appropriate.

We agree that the use of ST may not be appropriate in this context. We have changed all references to the “ST array” to “Visium array”. We have also removed all references to “spatial transcriptomics” or “ST” in the manuscript and changed them to “spatially resolved transcriptomics” or “SRT” to be in accordance with current practices in the field (i.e Marx, *Nature Methods* 2020). We have also changed the manuscript title to “Spatially resolved transcriptomics reveals the architecture of the tumor/microenvironment interface”. In addition, as suggested we have added a description of the different methods to profile spatial gene expression to the Introduction (lines 62-69).

17) In figure 1, the authors show that the tumor component to have more transcriptional activity. The authors should describe this in the manuscript, and explain whether this arises from tumor cells expression more transcripts, or form the tumor region to be more densely populated with cells.

We have added an approximation of the number of cells per Visium array spot to **new Fig. S1g** and feel the increased number of transcripts detected in the tumor region is likely due to the tumor region containing more cells. Also see response to your point #29, below.

18) Line 27, 31: Abbreviation “TME” used on line 31 but not described (line 27)

We have removed all references to the TME from the manuscript.

19) Line 37: missing comma after “by their nature”?

We have rewritten this part of the Introduction.

20) Line 39: this is not a study of “tumor invasion”. The authors should use a more accurate term of what is being shown.

We agree that we are not studying tumor invasion per se. It could be argued that tumor cells moving from their primary site of origin (the dermal/epidermal junction in the case of melanoma) into new tissues (the muscle in our case) is a form of invasion, but we agree that the paper was not explicitly looking at how this occurs, and instead focuses on how the tumor cells interact with their environment. Based on this, we have changed the wording of this sentence.

21) Line 49, 60: “in vivo” should be “in situ”

We changed “in vivo” to “in situ” (line 57) and removed the other noted reference.

22) Line 143 – the enrichment analysis should be presented in a more systematic way such as a supplementary table with the enrichment score per cluster.

We agree that the UMAP projection is not the clearest way to show the enrichment of the SRT interface score for each cluster. To show the enrichment in a more quantitative way, we replaced the UMAP with a violin plot (**new Fig. 3c**).

23) Line 146 – The use of “confirm” is not justified. The result points to the existence of 2 transcriptional expression patterns that have to still be confirmed. It’s unlikely that these are cell types, so please describe them as cell states unless you have evidence.

We have changed “cell type” to “cell state” in this context and changed “confirm” to “suggests” (lines 194-196).

24) Line 156 – the authors should choose more appropriate phrases to describe what is being shown – here the authors have not established this as a model for tumor invasion

As discussed in point #20 above, we agree that our study is not a model of tumor invasion per se. We have removed all references to tumor invasion from our manuscript.

25) 2C – the names of the genes are unreadable due to overlapping. Please use something like ggrepel to prevent overlap.

We have made the suggested changes to **Fig. 2c**.

26) 2e – NMF factor 15 doesn’t seem to correlate with the spots that were annotated as interface, but rather other regions. Can the authors explain what is represented in factor 15?

As noted in point #13 above, we have redone our NMF analyses to only include 11 factors after estimating the optimal number of factors using nmfEstimateRank.

27) 3d – please indicate the interface cell sub clustering (using a small split or vertical dashed line)

We have added this as a **new Fig. 3d** with the tumor-like and muscle-like interface cell states labelled.

28) 4d – I find this figure incomplete without the muscle and tumor cell types.

We agree that this figure is lacking a quantification of enrichment scores across the two interface cell states and have added this as **new Fig. 5b**.

29) S1 – It would be nice to have the histology slide adjacent to the spot plots. Whats the explanation for bimodality of transcripts per spot? More transcripts per cell (in which case please show similar evidence in scRNAseq with UMI per cell), or more cells per spot (in which case please show DAPI stained slides and approximate the density of cells)

We have added the histology images to **new Fig. S1c**. We believe the bimodality of transcripts per spot in our SRT data is likely due to both the number of cells per spot as well as the number of transcripts per spot, as our scRNA-seq and snRNA-seq datasets show that tumor and interface cells/nuclei contain somewhat more transcripts than other cell types present in the dataset. We have approximated cell density in our SRT data from Hoescht-stained sections (tumor and muscle regions) and added it to Fig. S1 (**new Fig. S1g**). We also added a plot (**new Fig. S1f**) showing the number of UMIs/spot in each of the cell types in our SRT dataset. We have also added a **new Fig. S5** with plots showing the number of UMIs per cell for the different cell types in our scRNA-seq dataset and a **new Fig. S6** with plots showing the number of UMIs per nucleus in our snRNA-seq dataset.

30) S2 – what are the grey spots? Why are these inconsistent with figure 1? I could not find a corresponding section to the un-integrated analysis.

The inconsistency between Fig. 1 and the previous version of Fig. S2 stems from slightly different cluster assignments between the individual SRT samples and the integrated analysis of all 3 samples together. We have adjusted the cluster assignments in **Fig. S2** to match that of **Fig. 1**.

31) S4 – please rotate and order images to match the previous rotation and presentation of samples A, B and C

We have rotated and reordered all images in **Fig. S4**.

Reviewer #2:

- Is there a distinct population of interface cells?

As the authors note, and Figure 2a confirms, the “interface” ST spots contain an admixture of tumor and microenvironment cells. As such, the tumor-like muscle spots and the muscle-like tumor spots seen in Fig1e could simply reflect the ratios of muscle and tumor cells found at these ST spots – the spots that cluster closer to muscle have a higher muscle:tumor cell ratio, and the ones that cluster closer to tumor have a lower muscle:tumor ratio. To address this possibility, the authors utilize scRNAseq to further interrogate the identity and possible heterogeneity of this interface population. In the scRNAseq they find two subpopulations of cells – one subpopulation that looks more like tumor and another that looks more like muscle. Should these be treated as a single interface population or separately as muscle and tumor populations in all subsequent analyses? The authors should strongly consider the latter. To do so there are two factors that need to be addressed:

a. The lack of a normal muscle cell population in the scRNAseq dataset impairs assessment of the signature found in the “interface (muscle)” population. Without this comparison it is difficult to assess if the ciliated signature seen in Fig3d is distinctive or also found in normal muscle – it is notable that many of the cilia markers are enriched in the interface (muscle) subpopulation compared to the interface (tumor) subpopulation. These assertions would be easily visualized in a new Fig3f containing muscle, interface (muscle), interface (tumor), and tumor clusters comparing expression of markers genes for each cluster. If the proposed “interface (muscle)” and “interface (tumor)” populations are transcriptionally unique, genes from the “upregulated in interface” signature should be commonly expressed in the scRNAseq interface, but not in the bulk muscle or tumor groups. A survey of the literature suggests that while difficult, it is technically feasible to isolate muscle cells via scRNAseq.

We agree that it is difficult to draw conclusions about the muscle-like interface cell state without the presence of a muscle cluster in our scRNA-seq dataset. To address this, we performed single-nucleus RNA-seq of 10,748 nuclei from 3 fish with large transgenic melanomas. We have added **new Figures 4** and **S6** with our snRNA-seq results. Our snRNA-seq dataset contains tumor and muscle clusters as well as various other cell types (**new Fig. 4a**), in addition to a cluster of cells upregulating many of the genes upregulated in our scRNA-seq interface cluster (**new Fig. 4b**). As this group of cells clustered with the interface cluster from our scRNA-seq dataset when we integrated our scRNA-seq and snRNA-seq results (**new Fig. 4c**), we concluded that this cluster represents an “interface” cell state in our snRNA-seq dataset. Within the interface cluster, we identified multiple subclusters representing tumor-like and muscle-like cell states (**new Fig. 4d**), confirming our initial scRNA-seq results. Using this larger dataset, we also found interface-like cells representing other cell states other than muscle (i.e. liver). This is especially interesting considering recent work (Naser, *Cell Metabolism* 2021 May 11;S1550-4131) showing that melanomas can lead to reprogramming of liver cells “at a distance” even when not directly in contact. Thus, while muscle-like interface cells are probably the most common based on the biological behavior of our tumor model, it is possible that other cells get transcriptionally rewired by the melanoma cells, which will be an area for future explanation.

We confirmed that the interface cluster in our snRNA-seq dataset is transcriptionally distinct from the other cell types in our dataset by, as the reviewer suggested, plotting expression of interface marker genes across all the cell types in our snRNA-seq dataset (**new Fig. 4f**). We also made a similar heatmap from our scRNA-seq data, showing, as this reviewer suggested, expression of interface marker genes across the two interface cell states as well as the other cells in the dataset (**new Fig. 3e**).

We again found an upregulation of cilia genes in the tumor-like and muscle-like cell states within our snRNA-seq interface clusters (**new Fig. 5d-f**). However, the enrichment of cilia genes in interface nuclei in our snRNA-seq dataset is not nearly as dramatic as in our scRNA-seq dataset, which is likely because the level of absolute expression of these genes was considerably lower in our snRNA-seq data vs. our scRNA-seq data (**new Fig. S6c-d**). This is likely due to the fact that the nucleus contains only a fraction of the transcripts of the entire cell. That being said, several cilia genes were upregulated specifically in the tumor-like and muscle-like interface cell states in our snRNA-seq dataset, including *stmn1a* (**new Fig. 4b**) and *ran*, *tubb4b*, *tuba4l*, and *gmnn* (**new Fig. 5f**). Thus, while we were still able to quantify upregulation of cilia genes within the interface clusters in our snRNA-seq data, these technical limitations gave us less ability to see a large dynamic range of cilia genes in this particular dataset.

b. As noted above the interpretation of this “interface” signature rests in part on the identification of interface (muscle) cells. The two current scRNAseq datasets contain only 4 and 17 of these cells. To properly power the analysis of these groups, the authors should consider another scRNAseq run. Isolation of more “interface (muscle)” cells will enable properly powered downstream analysis of the two distinct interface signatures instead of pooling interface (muscle) and interface (tumor) cells together as in Fig4a-d.

We agree that our scRNA-seq dataset contains relatively few cells. As noted above, instead of doing more scRNA-seq, we chose to do snRNA-seq since that allowed us to capture muscle cells, which was not possible in our scRNA-seq approach. With regards to cell numbers, our new snRNA-seq dataset contains 10,748 nuclei, of which 891 are within the “interface” cluster. Within this cluster, the tumor-like subcluster contains 320 nuclei and the muscle-like subcluster contains 244 nuclei. We have also separated the interface subclusters in both our scRNA-seq and snRNA-seq datasets where appropriate (i.e **new Figs. 3e, 4d-f, 5b, 5d, 5e, 6b, 6c, and 6e**), and only pooled them when making direct comparisons to the SRT interface cluster, such as in **new Figs. 5c, 6a and 6f**.

• Are cilia genes upregulated in interface cells and are cilia present on these cells?

This question is related to the one above. The question in essence is whether the cilia signature identified in interface cells a product of the interface (muscle) subpopulation and is this subpopulation different from normal muscle?

a. The identification of a ciliated interface population is predicated on the identification of an interface population expressing cilia gene markers. Interrogation of the GEO scRNAseq reveals

that the ciliated gene signature is driven by only the 21 interface (muscle) cells captured. It appears that these cells, which appears transcriptionally distinct from the interface (tumor) cells, are driving the ciliated signature. As mentioned above, capture of a non-interface muscle population, would provide an adequate comparison to reinforce that these ciliated muscle cells only arise at the tumor interface.

As described above, our new snRNA-seq dataset shows significant enrichment of cilia genes in the interface cell states, similar to what we found in our SRT and scRNA-seq data. However, due to relatively low absolute expression of some of these genes in our snRNA-seq dataset (**new Fig. S6c-d**), the dynamic range was not very high (**new Figs. 5e and S6d**). But when considering our SRT, scRNA-seq and snRNA-seq datasets as a group, each dataset suggested that a ciliation gene signature was elevated at the mRNA level in the interface region. To better address whether cilia proteins are also upregulated at the interface, as would be predicted from our transcriptomics datasets, we performed new higher resolution staining of the cilia marker acetylated tubulin in tissue sections of the tumor-microenvironment interface, as discussed in more detail below. This new staining shows a much more striking enrichment of acetylated tubulin at the tumor-muscle interface that is not present in distant muscle or tumor (**new Figs. 5g-h and S9**). We would also highlight that in our scRNA-seq dataset, although the upregulation of cilia genes is much higher in the muscle-like interface state, we still see a significant upregulation of cilia genes in the tumor-like interface state relative to the rest of the tumor (**new Fig. 5d**). We feel that the high number of points plotted on top of the violins in our previous version of Fig. 5d may have obscured the differences between the groups; hence, we have removed the points from these plots to better show the overall trends across groups.

b. Many of the stainings presented in Fig4 and SFig5 are confusing or inconclusive. In Fig4g it appears that AcTub is staining axonemes at the edges of some muscle fibers but isn't staining anything in the muscle fibers that are surrounded by tumor cells. Fig4h and Fig4i are completely different – there is now no axonemal staining but instead diffuse staining in the muscle fibers that are surrounded by tumor cells. In Fig4i it appears that Arl13b is staining cilia, and some appear to arise from tumor cells – but these are seen with the AcTub staining in the same panel. The gamma tubulin staining in Fig4h and Fig4i is very difficult to interpret – are the puncta supposed to be basal body staining in single cells? In SFig5a the Arl13b staining is strange and does not overlap with AcTub. The AcTub staining is somewhat consistent with the presence of primary cilia – a single puncta per cell, but without a high magnification image it is difficult to know. To sort this out the authors should confirm that the antibodies are appropriate for use in zebrafish with control samples and provide high quality high magnification images.

We completely agree with this reviewer that our staining protocol and selected antibodies required optimization. We have since optimized our staining protocol by changing our antigen retrieval method. With these improvements, we agree, as this reviewer suspected, that our ARL13B antibody is not sufficiently specific enough to reliably label cilia in zebrafish FFPE tissue sections. For this reason, we have removed all stainings with this ARL13B antibody from the manuscript. Unfortunately, there is no commercially available antibody for zebrafish ARL13B (Sco). However, we were able to obtain an antibody against Sco from the lab of Zhaoxia Sun,

who generated their own antibody in-house (Duldulao et al., *Development* 2009). Unfortunately, this antibody clearly did not work for staining our tissue sections, where we observed broadly non-specific staining across all conditions we tried.

From these studies, we concluded that there is not currently an adequate way to stain for Sco/ARL13B in our sections. However, as part of the staining optimization process, we generated dramatically improved images of acetylated tubulin-positive cell projections at the tumor-microenvironment interface (**new Figs. 5g-h and S9**). In comparing our immunofluorescence images with images of acetylated tubulin-positive cilia from the literature, the morphology of these acetylated tubulin-positive projections is strongly consistent with that of cilia. The higher resolution imaging of the tumor, the muscle, and the interface makes it clear that these acetylated tubulin-positive cell projections are much more commonly found in the interface than elsewhere in the tissue section (**new Figs. 5g-h and S9**), entirely consistent with our transcriptomics data. However, even with the higher resolution imaging, this does still leave open the question of whether the cilia originate from the muscle-like cells in the interface, the tumor-like cells in the interface, or both. To address this, we are currently in the process of building an *in vivo* transgenic cilia reporter into our melanoma models. This is based on prior work from the Ciruna lab showing that an ARL13B-GFP fusion protein, when expressed in zebrafish, is a highly sensitive reporter of cilia (Borovina et al., *Nature Cell Biology* 2010). We will express this fluorescent cilia reporter under tumor- or muscle-specific promoters to determine from which cells the interface cilia originate, together with genetic manipulations described below, but we estimate that these types of functional experiments will take 2-3 years to complete. They will be a major focus of our next manuscript.

- How do ETS transcription factors regulate gene expression in interface cells?

The analysis of ETS regulation of interface cell genes raises the question of whether ETS factors are identified through their regulation of genes in interface (tumor) cells and whether the upregulation of cilia genes is linked to low expression of ETS factors.

a. As effectors of MAPK pathway signaling, ETS factors are expected to be active in tumor cells. In the analyses presented in Fig5a, each of the interface groups in which the ETS motif is enriched has a higher proportion of tumor cells. To what extent is the identification of ETS motifs in Fig5a driven by upregulation of ETS motif genes in interface (tumor) cells compared to the downregulation of ETS motif genes in interface (muscle) cells? It would be helpful to see motif

enrichment analysis performed by treating the interface (tumor) and interface (muscle) subpopulations separately.

We have added a **new Fig. 6b** showing the results from separate motif analyses of the interface (tumor-like) and interface (muscle-like) cell states, showing the enrichment of ETS motifs in both subclusters.

b. A model that emerges is that lower expression of ETS factors in interface cells causes derepression and upregulation of cilia genes in these interface cells. The rationale for the model seems to stem from the lower expression of ETS factors in tumor cells. However, the upregulation of cilia genes is predominantly observed in interface (muscle) cells. This seems inconsistent – how would lowered expression of ETS factors in tumor cells cause derepression in interface (muscle) cells?

We agree that the relationship between the different interface cell states is still unclear. However, we note that in both our scRNA-seq and snRNA-seq datasets, we observed a significant upregulation of cilia genes in the tumor-like interface states relative to the rest of the tumor cells/nuclei (**Fig. 5d-e**). We also quantified a significant downregulation of ETS TFs in tumor-like interface cells relative to the rest of the tumor in our scRNA-seq and snRNA-seq datasets (**Fig. 6c,e**), and a significant downregulation of ETS genes in muscle-like interface nuclei relative to the rest of the muscle nuclei in our snRNA-seq dataset (**Fig. 6e**). Thus, we see a downregulation of ETS genes and an upregulation of cilia genes in both the tumor-like and muscle-cell states in our scRNA-seq and snRNA-seq datasets. It is unclear if downregulation of ETS genes in tumor-like interface cells then triggers downregulation of ETS in muscle-like interface cells, or vice versa. The role of the cilia themselves in this process is also currently unclear. We are actively performing functional experiments in the lab to try and understand these relationships. As noted above, we are using a new cilia reporter line to image cilia specifically in tumor or microenvironment cells. In addition, we are also performing CRISPR/Cas9 experiments for all of the ETS genes we identified, focusing on their role in muscle-like versus tumor-like cells at the interface. These functional experiments will be critical in better understanding the relationship between ETS and cilia genes at the tumor-microenvironment interface. However, we feel that these experiments will likely take 2-3 years to uncover the precise molecular mechanism linking ETS and cilia in this context, and intend this to be the major focus of our next manuscript.

Other issues:

1. The authors should consider the possibility of doublets contributing to the “interface” cell population. As the authors note, spatial transcriptomics lack the resolution required to rule out multiple cells contributing to a spot signature. The authors should provide supplemental plots from their scRNAseq data showing “nCounts_RNA” and “nFeatures_RNA” to reinforce that these interface transcriptomes are not artifacts from muscle/tumor doublets.

We agree that there is a possibility that doublets may be present in the interface population, however we would note that the size of muscle fibers makes it almost impossible for them to be

encapsulated for scRNA-seq. To rule this out, we used the doubletFinder R package to detect potential doublets in our scRNA-seq dataset. The results indicate that there are very few doublets present in the interface. We have added a **new Fig. S5** with QC metrics from our scRNA-seq dataset, including the suggested “nCount_RNA” and “nFeature_RNA” plots (**new Fig. S5a,d**), as well as the doubletFinder results (**Fig. S5g**).

2. Figure 2D looks less convincing than NMF 8 and 10 found in the supplement, the authors should consider picking a different NMF for display in the main figure. Additionally, highlighting an NMF showing an enrichment of cilia genes in the interface region vs muscle would contribute to the argument that cilia genes are specific to the interface.

We have redone our NMF analyses after using the function `nmfEstimateRank` from the `nmf` R package to determine the optimal number of factors, by calculating the factor(s) at which cophenetic starts decreasing (Hutchins et al., 2008) and for which RSS presents an inflection point (Frigyesi and Höglund, 2008).

Based on these results, we have selected 11 factors for our analyses because cophenetic began decreasing at factor 10, and the RSS inflection point occurred around factor 12. Our new NMF results can be found in **new Fig. S4**. We have added new interface-enriched NMF factor 7 to **Fig. 2d**, in which we feel there is a convincing enrichment at the interface in samples A and B, and a somewhat lower enrichment in sample C. We have also highlighted the enrichment of cilia-related GO terms (such as membrane-enclosed lumen and organelle lumen) in **Fig. 2e** and in the text (lines 150-152 and 283-284).

3. The authors should consider showing feature plots of SYSCILIA genes (or a composite score) to show that this signature is enriched in just the interface population and not other cell types. Furthermore, a SYSCILIA feature plot would reveal if the ciliated signature is broadly expressed over both interface tumor and interface muscle cells, or just one population.

We have added **new Figs. 4d** (scRNA-seq data) and **4e-f** (snRNA-seq data) showing upregulation of cilia genes across multiple cell states in the interface cluster, relative to the other cell types present in the datasets.

4. In Figure 3b/c the point overlap obscures the red/blue expression score, the authors should adjust the point size for each cell for easier visualization of the ST interface score.

We have changed the previous UMAP projection in Fig. 3b to a violin plot (**new Fig. 3c**) to better visualize the enrichment of the SRT interface score in the scRNA-seq interface cluster.

5. The point made in Figure 3e could be more cohesively made with a zoomed in feature plot of Figure 3b where the muscle cells appear to cluster as a small independent island next to the “interface (tumor)” cells.

We have added this as new **Fig. 3d**.

6. The fold change between “interface (tumor)” and “tumor” does not appear significant in the current figure. The authors should state in the figure legend what the p-value represents.

We are not sure which plot the reviewer is referring to, but have clarified the p-value labels in the legends of all figures and in the section “Statistical analysis” in the Methods.

7. The authors should reconcile their motif analysis methods and text for clarity. In text the authors state, “We queried the zebrafish genome for genes with an ETS motif within 500kb of the transcription start site...” whereas in the methods the authors state “...were queried within +/- 500bp of the TSS of differentially expressed genes.”

This was a typo in the text and should have said 500 bp. We have fixed the typo in the text.

8. The authors should consider scoring their ETS and Ciliary signatures in the Tirosh 2016 scRNAseq dataset. The Tirosh data set is more analogous to the model system used by the authors as it contains melanoma and microenvironmental cells isolated from primary tumors.

We agree that the Tirosh et al., 2016 melanoma scRNA-seq is more analogous to our zebrafish system. As a result, we have removed all analyses using the Wouters et al. dataset from the manuscript, and replaced them with an analysis of the Tirosh dataset which is comprised of tumor and microenvironment cells from human melanoma patients. These data are in new **Fig. S10**. Similar to our scRNA-seq and snRNA-seq datasets, we found evidence of cells in an interface-like state within both the tumor and microenvironment cell types in this dataset (**Fig. S10a-c**). Interface cells were detected by scoring each cell for expression of the human orthologs of genes upregulated by more than 1.5-fold in our scRNA-seq interface cluster. Interface cells were defined as cells with marker gene enrichment scores above 0.5, which we have clarified in **Fig. S10b**. We found that both the tumor-like and CAF-like interface cell clusters upregulated cilia genes and downregulated ETS genes, and that the two were

significantly negatively correlated (further suggesting that ETS transcription factors may act as transcriptional repressors of cilia genes) ($R = -0.1462$; $p = 1.31 \times 10^{-23}$; **Fig. S10d-e**).

9. The authors should fix the missing words in the sentence on line 413 so that it doesn't hang onto reference 43 in the superscript.

We have reorganized the manuscript to remove this formatting error.

10. The NMFs found in the supplement are hard to interpret. The orientation appears different from the slices found in the main figures. Furthermore, the resolution of the top GO terms in some NMF's is low and hard to read (NMF 10/12). Correction of these issues would greatly facilitate interpreting the NMF results.

As described in point #2 above, we have rotated and rearranged the images in **Fig. S4** and also redone our NMF analysis after selecting an optimal number of ranks using `nmfEstimateRank`.

Reviewer #3:

1. As the authors acknowledge, in the spatial transcriptomics data, the interface region contains a combination of tumor and muscle cells. It is therefore somewhat trivial that these regions might be identified as a cluster that is distinct from tumor-only and muscle-only regions, even if the transcriptome of individual cells in the interface region is identical to the corresponding cell types in other (non-interface) regions. Thus, an alternative to the author's interpretation of this cluster as describing a unique profile of cells in the tumor-muscle interface is that it is merely the combination of two distinct profiles. Fig. 2C is described as comparing the interface program to that of tumor AND muscle cells, but it is not clear to me what exactly that means (i.e. were equal proportions of muscle and tumor cells taken?). Instead, the authors should use a more rigorous approach to analyze the interface expression profile and identify genes that are more highly or lowly expressed in the interface region compared to the expected expression as defined by a combination of tumor and muscle cell, as defined by the corresponding "pure" regions. A quantitative approach that considers the expected expression of a combination of tumor/muscle cells (in various proportions) and possibly of other relevant cell types that might exist in the interface, would be appropriate; but this is perhaps computationally subtle and challenging. Alternatively, a very simple conservative approach would be to compare the interface expression to the maximum of each gene between tumor and muscle areas, thereby ensuring that any gene that comes up as upregulated cannot simply be explained by cell type mixture; similarly, downregulated genes may be defined as those lower in the interface than in the minimum of the two types of regions (tumor/muscle).

We thank the reviewers for this excellent point and for the suggestions for analysis approaches to address this issue. We agree that the interface likely contains a mixture of both tumor and muscle cells, and so simply computing the fold-change in expression in the tumor versus both tumor AND muscle (as we presented previously) to identify interface-enriched genes is insufficient to address the issue of cell type mixtures in the interface. As per the reviewer's suggestions, we instead computed the fold-change in expression/*p*-values for each gene by comparing a given gene's expression in the interface to the maximum expression in the tumor OR the muscle (whichever the gene is more highly expressed in). We have added this plot as **new Fig. 2c**. We clarified the details of the analysis in the Methods section entitled "Identification of genes enriched in the SRT interface". We would also highlight the fact that we identified an interface cluster in each of our scRNA-seq and snRNA-seq datasets, indicating that the interface cell state is not simply a mix of two cell types.

2. To aid in the interpretation of the spatial transcriptomics data, the single cell RNA-seq data was generated with the hope of defining all of the individual cell types. However, this data lacks muscle cells and therefore is less useful for the combined analysis than it was envisaged. The authors mention the size of multinucleated muscle cells as a potential reason for not obtaining them. Accordingly, it might be useful to attempt a single nuclei (as opposed to single cell) approach to isolate and profile those cells.

We agree that it is difficult to draw conclusions about the muscle-like interface cell state without the presence of a muscle cluster in our scRNA-seq dataset. To address this, we performed single-nucleus RNA-seq of 10,748 nuclei from 3 fish with large transgenic melanomas. We have added **new Figures 4 and S6** with our snRNA-seq results. Our snRNA-seq dataset contains tumor and muscle clusters as well as various other cell types (**new Fig. 4a**), in addition to a cluster of cells upregulating many of the genes upregulated in our scRNA-seq interface cluster (**new Fig. 4b**). As this group of cells clustered with the interface cluster from our scRNA-seq dataset when we integrated our scRNA-seq and snRNA-seq results (**new Fig. 4c**), we concluded that this cluster represents an “interface” cell state in our snRNA-seq dataset. Within the interface cluster, we identified multiple subclusters representing tumor-like and muscle-like cell states (**new Fig. 4d**), confirming our initial scRNA-seq results. Using this larger dataset, we also found interface-like cells representing other cell states other than muscle (i.e. liver). This is especially interesting considering recent work (Naser, *Cell Metabolism* 2021 May 11;S1550-4131) showing that melanomas can lead to reprogramming of liver cells “at a distance” even when not directly in contact. This will be an area of future study for our lab.

3. In the single cell data, the interface cluster is shown to contain two subclusters reflecting tumor and muscle cells. Yet, much of the analysis is done with this entire cluster as a single entity (i.e. ignoring the division into two cell types). This approach introduces a considerable confounder that makes it difficult to interpret the results. If the cluster contains two cell types then the clustering should be modified to separate them and all analyses should be done for the re-defined clusters that will have a more accurate annotation. The current approach of keeping the combined cluster would be expected to increase the consistency with the spatial data (in which the interface region is a combination of tumor and muscle cells) but such consistency reflects a suboptimal use of the single cell RNA-seq data in which the resolution of the analysis is in fact reduced to be lower than it should be.

We have now separated the interface subclusters in both our scRNA-seq and snRNA-seq datasets where appropriate (i.e **new Figs. 3e, 4d-f, 5b, 5d, 5e, 6b, 6c, and 6e**), and only pooled them when making direct comparisons to the SRT interface cluster, such as in **new Figs. 5c, 6a and 6f**.

4. Related to the above point about the single cell RNA-seq analysis: there is no distinct cluster of muscle cells; but the “interface” cluster, which clusters together with the tumor cells, has a subset which is distinguished by high expression of muscle genes. This pattern strongly suggests that at least some of the “interface” cluster may reflect doublets of tumor and muscle cells. This could explain why muscle cells are not captured on their own but are captured in the context of the interface cluster; it could also again explain why the interface scRNA-seq cluster has some consistency with the spatial transcriptomics analysis. The possibility of doublets is a well known limitation in single cell RNA-seq analysis, and multiple methods have been developed to try to deal with it (albeit they still have limitations). I am surprised to see that the authors completely ignore this possibility, and do not mention doublets anywhere in the manuscript. The implication, along the same line as noted above for the spatial transcriptomics,

is that it is not clear what parts of the results indeed reflect a unique profile of interface cells, and what part reflects the combined signal of tumor and muscle cells.

We agree that there is a possibility that doublets may be present in the “interface” population, although we note that muscle fibers are almost impossible to encapsulate for scRNA-seq due to their size, indicating the unlikeliness of tumor-muscle doublets present in our scRNA-seq data. However, to rule out the presence of doublets in the interface, we used the doubletFinder R package to detect potential doublets in our scRNA-seq dataset. The results indicate that there are very few doublets present in the interface. We have added a **new Fig. S5** with QC metrics from our scRNA-seq dataset, including the doubletFinder results (**Fig. S5g**). We also filtered all predicted doublets out of our new snRNA-seq dataset before any downstream analyses, as described in the Methods.

5. Fig. 4E demonstrates very clearly that cilia expression is dramatically higher in the muscle interface cells than in all other cells. This effect is extremely strong. In contrast, the same figure shows that the difference in cilia expression among tumor cells in the interface and non-interface is much smaller; it is still statistically significant, but seems biologically somewhat negligible compared to the former effect, and it is important to note that statistical significance is of limited value when the number of profiled cells is so large (i.e. any gene would be statistically significant when comparing two cell types with enough cells profiled from each cell type) and hence the focus in such analysis should be on effect-size rather than on statistical significance. Thus, cilia expression is very high in muscle cells, compared to tumor cells both in the interface and outside of the interface. It is then easy to imagine that all results related to cilia expression in the interface reflect the residual signal of muscle cells in the interface clusters of both spatial transcriptomics and single cell RNA-seq analysis, as described extensively in the previous comments. Overall, it seems problematic to focus this manuscript on an effect that is (A) barely noticeable (compare "interface(tumor)" to "tumor" in Fig. 4E), and (B) can be confounded by the presence of an effect that is an order-of-magnitude higher (compare "interface(muscle)" to all others in Fig. 4E).

We agree that cilia gene expression is much higher in the muscle-like interface cluster in our scRNA-seq dataset relative to all the other cell types. However, we did quantify a significant, albeit smaller, upregulation of cilia genes in the tumor-like interface cells as well (**new Fig. 5d-e**). The small-looking difference between interface (tumor-like) and tumor in **Fig. 5d** is likely due to the extremely high upregulation of cilia genes in the interface (muscle-like) cluster. With different Y-axis limits this difference may look more substantial.

But you raise a very important question: exactly what function, and in what cells, are the cilia most important? While our collective SRT, scRNA-seq and snRNA-seq datasets all support an increase in ciliation gene programs at the mRNA level, we felt that this required better analysis at the protein level. To better address whether cilia proteins are also upregulated at the interface, as would be predicted from our transcriptomics datasets, we performed new higher resolution staining of the cilia marker acetylated tubulin in tissue sections of the tumor-microenvironment interface. This new staining shows a much more striking enrichment of

acetylated tubulin at the tumor-muscle interface that is not present in distant muscle or tumor (new Figs. 5g-h and S9), entirely consistent with our transcriptomics analyses. However, even with the higher resolution imaging, this does still leave open the question of whether the cilia originate from the muscle-like cells in the interface, the tumor-like cells in the interface, or both, in addition to the major open question of the functional role of cilia in this context. To address this, we are currently in the process of building an *in vivo* transgenic cilia reporter into our melanoma models. This is based on prior work from the Ciruna lab showing that an ARL13B-GFP fusion protein, when expressed in zebrafish, is a highly sensitive reporter of cilia (Borovina et al., *Nature Cell Biology* 2010). We will express this fluorescent cilia reporter under tumor- or muscle-specific promoters to determine from which cells the interface cilia originate. We are also performing CRISPR/Cas-mediated inactivation of individual zebrafish ETS genes specifically in tumor or microenvironment cells to determine in which cells ETS transcription factors function to promote melanoma growth and ciliation. In parallel, we are also inactivating cilia genes in tumor or microenvironment cells to determine the functional role of cilia at the interface. However, we estimate that these types of functional experiments will take 2-3 years to complete. They will be a major focus of our next manuscript.

6. The authors include analysis of existing single cell RNA-seq data from human cell lines to show consistency of the results in human melanoma. However, it would be much more relevant to examine data directly from human patients, especially given the manuscript focus on cell-cell interactions and interface regions that would not exist in cell lines; such patient data is available through multiple studies including those already referenced in the manuscript.

As suggested, we have removed all analyses using the Wouters et al. dataset from the manuscript, and replaced them with an analysis of the Tirosh et al., 2016 scRNA-seq dataset which is comprised of tumor and microenvironment cells from human melanoma patients. These data are in new Fig. S10. Similar to our scRNA-seq and snRNA-seq datasets, we found evidence of cells in an interface-like state within both the tumor and microenvironment cell types in this dataset (Fig. S10a-c). Interface cells were detected by scoring each cell for expression of the human orthologs of genes upregulated by more than 1.5-fold in our scRNA-seq interface cluster. Interface cells were defined as cells with marker gene enrichment scores above 0.5, which we have clarified in Fig. S10b. We found that both the tumor-like and CAF-like interface cell clusters upregulated cilia genes and downregulated ETS genes, and that the two were significantly negatively correlated (further suggesting that ETS transcription factors may act as transcriptional repressors of cilia genes) ($R = -0.1462$; $p = 1.31 \times 10^{-23}$; Fig. S10d-e).

7. If the main conclusion of the work is the possibility that cilia expression has a role in invasion, then follow up experiments to support that would be extremely useful.

We agree that the role of cilia in tumor invasion is a major open and exciting question that stems from our current work. As noted above, we are actively working now to determine how cilia function at the tumor-microenvironment interface using both a transgenic cilia reporter as well as knockout of the ETS transcription factors. We are excited to perform these functional

experiments, but because the focus of the current manuscript is computational, we would prefer to address these important functional experiments in a subsequent manuscript.

Reviewers' Comments:

Reviewer #1:

Remarks to the Author:

Dear Authors,

The revised manuscript by Hunter et al presents their work in identifying and characterising a group of "interface cells" located in between the tumor mass and normal cells in a zebrafish melanoma model system. The revision is substantially improved over their original submission: they provide new snRNAseq data that enables the characterisation of muscles cells, they expand and improve data analysis and provide a comparison to human melanoma samples. I am also very happy to see that the presence of a code availability section and a link to GitHub.

However, there are a number of non-major issues which should be addressed before I deem the work to be suitable for publication.

[Intermediate]

1) It is nice to see that human primary melanoma samples. However, I feel that with the data and analysis presented it is too bold to say that the interface cell state is "conserved" in human melanoma" (line 376). To make the claim that they are indeed conserved, the authors should show specificity to melanoma by analysing negative control scRNAseq datasets where they do not expect to see the interface signature (e.g. peripheral blood from healthy individuals, and human CNS) to determine that the effect is specific, and quantify effect size and relating this to potential strength of phenotype. Instead, it would be more appropriate to state there are indications that this interface cell state "signature" identified in this study, is observed in human primary melanoma, and this warrants further study. This would affect the title and conclusion of the section in lines 364-382.

2) Differential analysis of SYSCILIA and ETS enrichment scores should be performed as paired analysis: e.g. interface-tumor-like VS tumor; interface-CAF-like VS CAF; interface-Tcell-like VS Tcells (i.e. split up the interface-immune-like into the specific immune cell populations as the normal immune cell populations seems to have different ETS and SYSCILIA enrichment scores).

3) The NMF analysis (e.g. figure S4) is not well described in the text. The analysis has been performed, with some description of which factors are enriched at the interface in different samples without any meaningful interpretation of what these factors represent. My opinion is echoed in the authors conclusion of this analysis (lines 167-169) which doesn't actually draw a conclusion from the NMF analysis. Perhaps this would be an opportunity to describe a role for organelle (a term enriched in factor 7 and 8) mediate signalling at the interface.

[minor]

4) Line 68. In addition to the 55um spot size, the 100 um spot to spot distance should be mentioned (or instead a 45um gap between spots).

5) Line 69-71. This sentence could be formulated better, e.g. "However, to overcome the limited spatial resolution of SRT arrays, a number of computational methods to infer single cell resolved gene expression profiles have recently been developed". It would be appropriate to also cite other tools to perform such tasks.

6) Line 90. The SRT assay doesn't really profile "cell interaction" more than it does "cell spatial organisation". The cell-cell interaction analysis is investigated using sequencing data at a later section.

7) Line 211. The citation [16] only reports tumor-immune cell fusions. The authors should be specific about this when referring to the paper, as this citation does not provide support for tumor-muscle cell fusions (which is what is eluded to in the current text).

8) Line 241. No need to specifically mention UMAP

9) Line 258. Suppl fig 7b isn't required. You can simply highlight the either the name in 7a, or place a bold or red box around the column in the heatmap to which you want to draw attention.

10) Line 259. Is "enriched" the correct word? Unless MIA performed enrichment test, it would be perhaps more appropriate to say that the interface region "consisted" of ...

11) Line 265. The last part of the conclusion seems a little far fetched. The authors do not provide a clear rationale of how their results implicate the interface gene expression program to contribute to tumor growth into new tissues.

12) Line 357. The origin of table S1 is unclear. If this is compiled from literature, the sources should be stated, otherwise the authors should describe how this was computed in the methods.

13) Line 380. No cell-cell interaction (i.e. NicheNetR) analysis is presented. Perhaps the authors mean cellular spatial organisation?

14) Upon acceptance, the github repo should be made citable:
<https://guides.github.com/activities/citable-code/>

Reviewer #2:

Remarks to the Author:

The addition of snRNAseq data and nuclei from muscle cells is a welcome addition to the manuscript and adequately addresses our concern about the distinct nature of interface cells. The authors have addressed all of our other concerns adequately with the exception of staining of primary cilia in zebrafish sections. In these sections in the revised manuscript, acetylated tubulin is used as a marker of primary cilia. There are two issues with the staining presented: a) the morphologies of the Ac tubulin staining are not typical of primary cilia – the staining in many instances is non-contiguous and in many instances it varies in width (as in Fig 5h), and b) the length of the staining would be very atypical of primary cilia, which in most cell types are ~5-10uM in length. It's not clear what the Ac tubulin staining is showing. It is recommended that the authors perform a convincing positive control on zebrafish cells with the Ac tubulin antibody that is being used.

Reviewer #3:

Remarks to the Author:

The revised version is improved and includes significant new data. However, it still suffers from several limitations brought up previously which were only partially addressed, as described below.

Major comments:

1. In response to my comment about the difficulty in distinguishing a true "interface" expression profile from a mixture profile (of tumor and muscle cells) the authors added Fig. 2C. However, this figure seems to be ill defined; it is a volcano plot but does not follow the expected V-shape; instead, the most significant genes (by p-value) tend to have very low fold-changes, and even the genes around a fold-change of zero appear significant; this would suggest an error in defining this plot. Moreover, the genes that come up as interface-specific in this analysis are not consistent with those that come up in other analyses and hence the figure does not support the authors' claim nor does it eliminate the concern brought up in my previous review.

2. The authors add new snRNA-seq data in which a cluster of muscle cells is present and where a cluster of interface cells (similar to that seen in scRNA-seq) is also detected. This provides an important support. However, I am still concerned that a subset of the cells in such interface clusters may reflect doublets (or perhaps some other form of technical confounders in which reads from one cell make their way into another cell). I am not suggesting that all interface cells are

doublets but that a fraction of them appears likely to be doublets and that additional analysis is warranted to better distinguish between the true interface cells and such doublets. My concern is driven by (1) the location of the "interface" cluster in Fig. 4a (in the center between multiple other clusters, exactly as would be expected from doublets among those clusters); (2) the observation that interface cells tend to have higher num. of detected genes compared to all other clusters (e.g. Fig. S6 as well as Fig. S5); (3) the observation that interface cells may be further sub-clustered into those with immune, liver, intestine and muscle patterns, consistent with what we would expect from a collection of doublets (or other forms of "leakiness" in assignment of reads to cells). Since doublets are a well known issue in scRNA-seq, are particularly relevant in the context of comparison of scRNA-seq to spatial data (which is inherently measuring doublets/triplets etc.), and could account for some of the main observations in this work (e.g. Cilia expression in interface cells, which may be explained by doublets with muscle cells) this issue should be considered more rigorously.

I appreciate the author's attempt to exclude this possibility by running doubletFinder; but this would seem insufficient to me, as running such a program "off-the-shelf" might not be effective enough, it might require troubleshooting and suffers from several limitations in detecting all doublets. The output that is shown for doubletFinder only shows the assignment of cells as doublets or not, but does not provide any direct data to support this classification. The interface cells may be suspected to reflect particular types of doublets and the data to exclude this possibility should be shown clearly. Thus, I would ask the authors to more clearly test and demonstrate that the interface cells are inconsistent with expected doublet profiles and possibly to divide the "initial" interface cells to those that may, and those that may not, be accounted by doublets; for example, any interface cell that is assigned to the immune-like subcluster can be further interrogated by the expression of all genes that are unique to each immune cell type (one cell type at a time); if it is a doublet, then a large fraction of all genes unique to a certain immune cell type should be expressed by that cell, while if it is truly a state of interface cell that expresses certain immune-related genes, then the fraction of such immune-specific genes should be quite limited. Thus, a plot that would directly demonstrate that interface cells are incompatible with the corresponding doublets (given their sub-cluster assignments) would be very helpful in excluding this possibility.

4. I previously commented on the limited degree of cilia upregulation in interface (tumor-like) cells compared to non-interface tumor cells, and that the main cilia effect in fact comes not from interface cells but rather from muscle cells; this issue is further exacerbated by the possibility that some of the interface cells may reflect doublets. The authors responded that they are working on complex experiments that would allow them to dig deeper into this phenomena and that these experiments are beyond the scope of the current work. I appreciate the complexity of these experiments and do not expect them to be included in this work, but I still feel that the results should be modified or supplemented to better decouple the interface cilia effect from the muscle contribution and to provide more convincing data regarding the cilia expression in interface (tumor-like) cells.

5. As requested, the authors added an analysis of the Tirosh et al. melanoma patient data, and they claim that this analysis supports the presence of interface-like cells in melanoma patient samples with the same expression profile as seen in the zebrafish model. This analysis does not seem convincing to me. Of course, one could score cells for any program and with a lenient-enough definition could argue that a subset of cells express that signature; but more convincing evidence is needed in order to support the claim of a coherent upregulation of interface genes that is biologically and statistically significant; such evidence appears to be lacking in this analysis. The threshold of 0.5 is quite lenient and a subset of cells from each of the cell types passes it, as opposed to the expectation from the zebrafish analysis that this phenomena should be rather specific to the tumor cells; the co-expression of interface-related genes in those cells is not shown in any way (only the average score for the signature) and there is no basis for claiming statistical significance for this result. Currently, I do not see any evidence that the cell scores for interface-like signature go beyond what is expected by the overall variability of gene expression in this data and that point to the existence of interface-like cells.

Reviewer #1

The revised manuscript by Hunter et al presents their work in identifying and characterising a group of “interface cells” located in between the tumor mass and normal cells in a zebrafish melanoma model system. The revision is substantially improved over their original submission: they provide new snRNAseq data that enables the characterisation of muscles cells, they expand and improve data analysis and provide a comparison to human melanoma samples. I am also very happy to see that the presence of a code availability section and a link to GitHub.

However, there are a number of non-major issues which should be addressed before I deem the work to be suitable for publication.

[Intermediate]

1) It is nice to see that human primary melanoma samples. However, I feel that with the data and analysis presented it is too bold to say that the interface cell state is “conserved” in human melanoma” (line 376). To make the claim that they are indeed conserved, the authors should show specificity to melanoma by analysing negative control scRNAseq datasets where they do not expect to see the interface signature (e.g. peripheral blood from healthy individuals, and human CNS) to determine that the effect is specific, and quantify effect size and relating this to potential strength of phenotype. Instead, it would be more appropriate to state there are indications that this interface cell state “signature” identified in this study, is observed in human primary melanoma, and this warrants further study. This would affect the title and conclusion of the section in lines 364-382.

We have changed our description of the interface cell state to an “interface signature” as suggested, and removed any mentions to the interface state being “conserved” in human melanoma. As suggested, we have changed the title of that section to “An interface signature may be present in human melanoma”.

We have also redone this analysis with a newer, much larger human metastatic melanoma scRNA-seq dataset, containing 29,247 cells from 43 patient metastatic melanoma samples (Smalley et al., *Clin Cancer Res* 2021). This dataset contained more than 6-fold more cells than the Tirosh et al. dataset we previously analyzed (29,247 cells vs. 4,645 cells), providing significantly improved statistical power. We clustered the cells into 8 groups, using known marker genes for melanoma, as well as non-melanoma marker genes from the original publication (**new Fig. S11a**). As we did previously, we scored each cell in the dataset for expression of the human orthologs of the genes upregulated by more than 1.5-fold in the interface cluster in our zebrafish scRNA-seq dataset (**new Fig. S11b**). With this much larger dataset, we were able to identify a distinct subset of both tumor and microenvironment cells that upregulated the same genes upregulated in the interface in our zebrafish datasets (**new Fig. S11b-d**). These cells formed a distinct subcluster within both the tumor, T/NK cell, and myeloid cell clusters (**new Fig. S11b-c**), suggesting they represent a distinct cell state. We also plotted the expression of selected genes upregulated in the interface in our zebrafish datasets, such as

PLK1 (Figs. 3e, 4b, and new S11e), *HMGB2* (Figs. S9 and new S11e), *TUBB4B* (Figs. 5f and new S11e) and *TPX2* (Figs. 3e and new S11e). In all cases, we saw that these interface marker genes were highly upregulated within cells that we classified as “interface-like”. Although we feel that comparing the fish to human data will need even more validation in the future, this new analysis of the Smalley et al. human data provides compelling evidence that this ciliated interface cell population also exists in human melanoma.

2) Differential analysis of SYSCILIA and ETS enrichment scores should be performed as paired analysis: e.g. interface-tumor-like VS tumor; interface-CAF-like VS CAF; interface-Tcell-like VS Tcells (i.e. split up the interface-immune-like into the specific immune cell populations as the normal immune cell populations seems to have different ETS and SYSCILIA enrichment scores).

One of the limitations of using the Tirosh dataset to investigate whether the interface cell state may be conserved in human melanoma is that the number of cells in this dataset is relatively small, which was why we chose to instead take advantage of the newer and much larger dataset from Smalley et al. In this new analysis, within the 3 major groups of interface cells (tumor-like, T/NK cell-like, and myeloid-like), we scored each interface cell and their corresponding tumor/TME cells for expression of the human SYSCILIA genes, and found that SYSCILIA genes were significantly upregulated in the 3 major interface subclusters relative to the corresponding tumor/TME cell types (new Fig. S11f). We acknowledge the reviewer’s concerns about differences in SYSCILIA scores across different immune cell subpopulations, but in this case we chose to classify the immune and myeloid cells into larger groups (as done in the original publication for this dataset) in order to provide greater statistical power to allow us to make pairwise comparisons, which was the major advantage of this dataset compared to the Tirosh dataset. Although the original publication for the Smalley et al. dataset does also eventually separate the clusters into very specific subpopulations of different types of, for example, T cells, doing so in our analyses would have resulted in much lower numbers of cells for the different immune/myeloid cell types, which would have again significantly reduced our statistical power. However, we have added a sentence in the text acknowledging that different immune cell types within the interface may have different levels of expression of interface genes and potentially different functions within the interface (lines 385-386).

3) The NMF analysis (e.g. figure S4) is not well described in the text. The analysis has been performed, with some description of which factors are enriched at the interface in different samples without any meaningful interpretation of what these factors represent. My opinion is echoed in the authors conclusion of this analysis (lines 167-169) which doesn’t actually draw a conclusion from the NMF analysis. Perhaps this would be an opportunity to describe a role for organelle (a term enriched in factor 7 and 8) mediate signalling at the interface.

We have added an interpretation of the NMF results to the text: “This result suggests a high degree of biological activity within the interface region, with a potential role for membrane-bound organelles in signaling within this region.” (lines 160-163). Although we think it is likely that other membrane-bound organelles such as mitochondria may be important within the interface region,

we have chosen to focus on cilia in our manuscript as we have a significant amount of data implicating cilia as an important organelle in melanoma.

[minor]

4) Line 68. In addition to the 55um spot size, the 100 um spot to spot distance should be mentioned (or instead a 45um gap between spots).

We have added a description of the distance between spots as suggested (lines 63-64).

5) Line 69-71. This sentence could be formulated better, e.g. “However, to overcome the limited spatial resolution of SRT arrays, a number of computational methods to infer single cell resolved gene expression profiles have recently been developed”. It would be appropriate to also cite other tools to perform such tasks.

We have added the suggested sentence, citing two additional tools (SPOTlight and Stereoscope) (lines 64-66).

6) Line 90. The SRT assay doesn't really profile “cell interaction” more than it does “cell spatial organisation”. The cell-cell interaction analysis is investigated using sequencing data at a later section.

We have modified this sentence to read: “To investigate the transcriptional landscape of tumors and neighboring tissues *in situ* with spatial resolution...” (lines 85-86).

7) Line 211. The citation [16] only reports tumor-immune cell fusions. The authors should be specific about this when referring to the paper, as this citation does not provide support for tumor-muscle cell fusions (which is what is eluded to in the current text).

We have reworded this sentence to clarify that we are referring to tumor-immune cell fusion in the noted citation, and to explicitly state that tumor-muscle cell fusion has not yet been reported (lines 205-207).

8) Line 241. No need to specifically mention UMAP

We removed the mention of UMAP (line 236).

9) Line 258. Suppl fig 7b isn't required. You can simply highlight the either the name in 7a, or place a bold or red box around the column in the heatmap to which you want to draw attention.

We have removed **Fig. S7b** (now Fig. S8) as requested.

10) Line 259. Is “enriched” the correct word? Unless MIA performed enrichment test, it would be perhaps more appropriate to say that the interface region “consisted” of ...

MIA is in fact performing an enrichment test (hypergeometric test), as described in more detail in our paper Moncada et al., *Nature Biotechnology* (2020).

11) Line 265. The last part of the conclusion seems a little far fetched. The authors do not provide a clear rationale of how their results implicate the interface gene expression program to contribute to tumor growth into new tissues.

We have toned down these statements (“...composed of tumor and microenvironment cells which upregulate a common gene program that may contribute to tumor-microenvironment cell interactions at the tumor boundary”, lines 270-272).

12) Line 357. The origin of table S1 is unclear. If this is compiled from literature, the sources should be stated, otherwise the authors should describe how this was computed in the methods.

We have stated in the legend of **Table S1** (now **Table S2**) that the data presented in this table is from ZFIN and Uniprot. We also added this information to the table itself.

13) Line 380. No cell-cell interaction (i.e. NicheNetR) analysis is presented. Perhaps the authors mean cellular spatial organisation?

We did complete NicheNet analyses which the reviewer may have missed - these analyses are shown in **Fig. S9**.

14) Upon acceptance, the github repo should be made citable:
<https://guides.github.com/activities/citable-code/>

Thank you for this suggestion - we will include a citable version of the Github repository in the final article.

Reviewer #2

The addition of snRNAseq data and nuclei from muscle cells is a welcome addition to the manuscript and adequately addresses our concern about the distinct nature of interface cells. The authors have addressed all of our other concerns adequately with the exception of staining of primary cilia in zebrafish sections. In these sections in the revised manuscript, acetylated tubulin is used as a marker of primary cilia. There are two issues with the staining presented: a) the morphologies of the Ac tubulin staining are not typical of primary cilia – the staining in many instances is non-contiguous and in many instances it varies in width (as in Fig 5h), and b) the length of the staining would be very atypical of primary cilia, which in most cell types are ~5-10 μ M in length. It's not clear what the Ac tubulin staining is showing. It is recommended that the authors perform a convincing positive control on zebrafish cells with the Ac tubulin antibody that is being used.

We agree that the morphology of the acetylated tubulin-positive projections we observe at the tumor boundary is not completely consistent with the size and structure of typical primary cilia. We think this is an exciting area for future study, and are actively working now to characterize the internal structure of these acetylated tubulin-positive projections we observe at the interface, and their function at the tumor boundary. As suggested, we did validate our acetylated tubulin antibody using a zebrafish melanoma cell line (ZMEL) expressing a fluorescent cilia reporter transgene. We generated this cell line by nucleofecting a plasmid containing the cilia reporter construct ARL13B-GFP under a ubiquitous promoter into ZMEL cells. For this construct, we used a mouse ARL13B gene that had been previously used to generate a transgenic zebrafish line expressing a fluorescent cilia reporter (Borovina et al., 2010). After nucleofecting this construct into ZMELs, we were able to visualize the transgene in a subset of melanoma cells, as expected. We then stained these for acetylated tubulin (using the same antibody as in the stainings in the manuscript) by immunofluorescence and found a perfect overlap between the ARL13B-GFP and tubulin staining. We would also note that these cilia are almost exactly the length noted by the reviewer (5-10 μ m). Thus, these results indicate that this antibody does in fact detect cilia in zebrafish cells.

We also note that the acetylated tubulin antibody we used for these studies (Sigma #T6793) has been used in numerous publications, with 1028 citations at the present time (https://www.sigmaaldrich.com/US/en/search/t6793?focus=papers&page=1&perPage=30&sort=relevance&term=T6793&type=citation_search).

Reviewer #3

The revised version is improved and includes significant new data. However, it still suffers from several limitations brought up previously which were only partially addressed, as described below.

Major comments:

1. In response to my comment about the difficulty in distinguishing a true "interface" expression profile from a mixture profile (of tumor and muscle cells) the authors added Fig. 2C. However, this figure seems to be ill defined; it is a volcano plot but does not follow the expected V-shape; instead, the most significant genes (by p-value) tend to have very low fold-changes, and even the genes around a fold-change of zero appear significant; this would suggest an error in defining this plot. Moreover, the genes that come up as interface-specific in this analysis are not consistent with those that come up in other analyses and hence the figure does not support the authors' claim nor does it eliminate the concern brought up in my previous review.

We agree that the previous versions of Fig. 2c may have been difficult to interpret. We thus returned to the suggestion by this reviewer in their previous set of comments:

"...a very simple conservative approach would be to compare the interface expression to the maximum of each gene between tumor and muscle areas, thereby ensuring that any gene that comes up as upregulated cannot simply be explained by cell type mixture; similarly, downregulated genes may be defined as those lower in the interface than in the minimum of the two types of regions (tumor/muscle)."

We have thus redone our analysis based on this suggestion. First, we calculated the average log₂ fold change of each gene in the interface relative to the rest of the SRT array spots in the dataset, using the Wilcoxon rank sum test within the Seurat function FindMarkers. This function calculates an average log₂ fold change for each gene, as well as a *p* value. Next, we calculated the average log₂ fold change for each of these genes within the tumor and muscle clusters (separately). Finally, we filtered our list of interface genes to include only genes for which their average log₂ fold change was higher in the interface than both of the tumor and muscle clusters, and did a final filtering step to only include genes with an adjusted *p* value of less than 0.05. This resulted in a list of 264 genes: 252 upregulated and 12 downregulated. We have included a table with the average log₂ fold change for each of these 264 genes in the interface, tumor, and muscle clusters, and adjusted *p* value, as **new Table S1**. We also plotted expression of each gene in the interface on a volcano plot in **new Fig. 2c**, with the 264 up- and downregulated genes relative to the tumor and muscle labelled. We have also added all of the code used for these calculations and plotting to our Github repository (see **Code availability** section).

2. The authors add new snRNA-seq data in which a cluster of muscle cells is present and where a cluster of interface cells (similar to that seen in scRNA-seq) is also detected. This provides an important support. However, I am still concerned that a subset of the cells in such interface clusters may reflect doublets (or perhaps some other form of technical confounders in which

reads from one cell make their way into another cell). I am not suggesting that all interface cells are doublets but that a fraction of them appears likely to be doublets and that additional analysis is warranted to better distinguish between the true interface cells and such doublets.

My concern is driven by (1) the location of the "interface" cluster in fig. 4a (in the center between multiple other clusters, exactly as would be expected from doublets among those clusters; (2) the observation that interface cells tend to have higher num. of detected genes compared to all other clusters (e.g. Fig. S6 as well as Fig. S5); (3) the observation that interface cells may be further sub-clustered into those with immune, liver, intestine and muscle patterns, consistent with what we would expect from a collection of doublets (or other forms of "leakiness" in assignment of reads to cells). Since doublets are a well known issue in scRNA-seq, are particularly relevant in the context of comparison of scRNA-seq to spatial data (which is inherently measuring doublets/triplets etc.), and could account for some of the main observations in this work (e.g. Cilia expression in interface cells, which may be explained by doublets with muscle cells) this issue should be considered more rigorously. I appreciate the author's attempt to exclude this possibility by running doubletFinder; but this would seem insufficient to me, as running such a program "off-the-shelf" might not be effective enough, it might require troubleshooting and suffers from several limitations in detecting all doublets. The output that is shown for doubletFinder only shows the assignment of cells as doublets or not, but does not provide any direct data to support this classification. The interface cells may be suspected to reflect particular types of doublets and the data to exclude this possibility should be shown clearly. Thus, I would ask the authors to more clearly test and demonstrate that the interface cells are inconsistent with expected doublet profiles and possibly to divide the "initial" interface cells to those that may, and those that may not, be accounted by doublets; for example, any interface cell that is assigned to the immune-like subcluster can be further interrogated by the expression of all genes that are unique to each immune cell type (one cell type at a time); if it is a doublet, then a large fraction of all genes unique to a certain immune cell type should be expressed by that cell, while if it is truly a state of interface cell that expresses certain immune-related genes, then the fraction of such immune-specific genes should be quite limited. Thus, a plot that would directly demonstrate that interface cells are incompatible with the corresponding doublets (given their sub-cluster assignments) would be very helpful in excluding this possibility.

We agree the potential presence of cell doublets is an important factor that must be considered in all scRNA-seq experiments, and that it needed further consideration in our paper. While it is not possible to exclude all possible instances of doublets, we have now more rigorously investigated this possibility to delineate the likelihood of doublets contributing to the biology we have uncovered. We have now done five separate analyses and filtering steps (as detailed below) to examine and rule out the possibility of doublets. We would note that the first 3 of these analyses are exactly what is suggested by 10X Genomics to exclude doublets, but we also did an additional 2 analyses to further investigate this issue (<https://kb.10xgenomics.com/hc/en-us/articles/360001074271-Does-Cell-Ranger-automatically-exclude-doublets->).

- (1) We quantified the number of UMIs and genes across all clusters in our snRNA-seq and scRNA-seq datasets, as shown in **Fig. S5** (scRNA-seq) and **Fig. S6** (snRNA-seq). We would note that in all cases, the interface and tumor clusters contain similar numbers of both UMIs and genes per cell, so we are not sure how this suggests the presence of doublets in the interface clusters.
- (2) We used the R package DoubletFinder to identify potential doublets (the package specifically suggested by 10X Genomics in the article above). DoubletFinder did identify some potential doublets in our scRNA-seq dataset, which are shown in **Fig. S5g**; however, these potential doublets were scattered across all clusters, with very few found within the interface. We also used DoubletFinder to score the nuclei in our new snRNA-seq dataset before proceeding with any downstream analyses. As we stated in the Methods section, we filtered all predicted doublets out of our snRNA-seq dataset before continuing with analysis. In regards to the reviewer's concerns about DoubletFinder, we would note that in a recent paper that benchmarked 9 computational doublet-detection methods against a scRNA-seq dataset containing known doublets, the authors found that DoubletFinder was by far the most accurate of the 9 methods (Xi and Li, *Cell Systems* 2021).
- (3) We have now added a **new Fig. S7** with additional analyses of our snRNA-seq data, as suggested by this reviewer. As suggested, we calculated all of the genes expressed by the various tumor/microenvironment cell types, using the Seurat function FindMarkers with the Wilcoxon rank sum test. We then calculated the fraction of those tumor/microenvironment genes expressed by the corresponding interface cell state (i.e., the fraction of tumor-specific genes expressed by each tumor-like interface nucleus). We defined expression of a given gene as normalized expression > 0. The results are shown in the middle panel of **new Fig. S7**. We would like to highlight the muscle-like interface cell state, as this reviewer has repeatedly questioned whether this cell state can be attributed to the presence of doublets. As shown in **new Fig. S7**, the muscle-like interface nuclei expressed very few of the genes expressed by nuclei in the muscle cluster ($p = 4.0 \times 10^{-111}$). In addition, as noted in this reviewer's comment #4 below, we also do not see a high level expression of cilia genes in muscle in our SRT and snRNA-seq datasets, making it unlikely that doublets between tumor and muscle cells would explain our result.
- (4) As an additional analysis, we also calculated all genes upregulated by the various interface cell states and their related tumor or microenvironment clusters. We then calculated the overlap between genes upregulated in the interface cell state and the related tumor/microenvironment cluster. These results are shown in the Venn diagrams in **new Fig. S7** (left panels). These diagrams show that there is not a high degree of overlap between the genes upregulated in the interface cell states and the genes upregulated in the corresponding tumor/microenvironment cell type. At most (for example, in the liver and liver-like interface cell states) only about a third of genes were shared.
- (5) We also added plots to **new Fig. S7** (right panels) showing the number of UMIs per nucleus in the interface cell states relative to their corresponding tumor/microenvironment cell types.

As we described in the text (lines 248-255), in most cases there was not a significant degree of overlap between all genes upregulated between both cell states, indicating that these interface cell states are not caused by doublets. We did observe some overlap between genes expressed by NK cells and macrophages relative to the immune-like interface cells, suggesting that some doublets could be present within the immune-like interface cluster. We have added a sentence in the text explicitly stating that the immune-like interface cluster may contain doublets, and suggesting that determining whether these possible doublets result from technical or biological reasons would be an important area of future study (lines 255-260).

Importantly, we would also note that we cannot rule out the possibility of tumor-microenvironment cell fusion at the interface, or other biological effects that may contribute to the effects mentioned by the reviewer. In fact, we had already mentioned in the text that tumor-microenvironment cell fusion at the interface is a distinct possibility (lines 205-207). Notably, tumor-immune cell fusion has been reported in melanoma (Gast et al., *Science Advances* 2018). The fact that we recovered muscle-like interface cells from our scRNA-seq experiment, even though muscle fibres are difficult if not impossible to encapsulate for scRNA-seq, suggests that the muscle-like cells that we did capture may be in a hybrid state. As we have now done 5 different orthogonal analyses to examine the presence of doublets in the interface cluster(s), we do not feel that further analyses of our current data will be able to definitively determine whether the effects mentioned by the reviewer are due to technical or biological reasons.

4. I previously commented on the limited degree of cilia upregulation in interface (tumor-like) cells compared to non-interface tumor cells, and that the main cilia effect in fact comes not from interface cells but rather from muscle cells; this issue is further exacerbated by the possibility that some of the interface cells may reflect doublets. The authors responded that they are working on complex experiments that would allow them to dig deeper into this phenomena and that these experiments are beyond the scope of the current work. I appreciate the complexity of these experiments and do not expect them to be included in this work, but I still feel that the results should be modified or supplemented to better decouple the interface cilia effect from the muscle contribution and to provide more convincing data regarding the cilia expression in interface (tumor-like) cells.

We would note that both our new snRNA-seq dataset and our SRT dataset contain clusters of “normal” muscle transcriptomes, and we do not observe an upregulation of cilia genes in either muscle cluster (**Fig. 5e** for snRNA-seq data, and SRT data below).

Thus, we feel that it is unlikely that any enrichment of cilia genes we see in muscle-like interface cells can be attributed purely to doublets with muscle cells, as our SRT and snRNA-seq data show that muscle cells do not express cilia genes at high levels.

Additionally, see response to point #3 from this reviewer, where we explain in detail the 5 orthogonal analyses we have completed to rule out the presence of doublets in the interface. As we described above, our analyses strongly suggest that nuclei in the muscle-like interface cell state cannot be attributed to doublets with muscle nuclei, as the fraction of muscle-specific genes expressed by nuclei in the muscle-like interface state was much lower than in the muscle cluster ($p = 4 \times 10^{-111}$). Furthermore, it is unlikely that we would observe doublets containing muscle cells in our scRNA-seq dataset, since as mentioned above, it is technically almost impossible to encapsulate muscle fibres for scRNA-seq. Thus, we feel that it is extremely unlikely that any enrichment of cilia genes observed in interface cells can be attributed to possible doublets containing muscle cells. Additionally, our IF data (see **Figs. 5g-h** and **S10**) shows clearly low levels of acetylated tubulin (cilia marker) staining in muscle relative to the interface cells. Since protein expression is perhaps the ultimate validation of RNA-seq results, based on our multiple transcriptomics analyses showing low levels of cilia gene expression in muscle, and our imaging data showing low levels of cilia protein expression in muscle, we do not see how our results could suggest the upregulation of cilia genes we see in interface cells is purely from doublets containing interface cells with muscle cells. We agree that we cannot currently determine whether the interface cilia initiate from tumor-like or muscle-like interface cells; however, determining the source of the cilia requires complicated genetics experiments that will take at least a year to complete, and we feel are beyond the scope of the current paper.

5. As requested, the authors added an analysis of the Tirosh et al. melanoma patient data, and they claim that this analysis supports the presence of interface-like cells in melanoma patient samples with the same expression profile as seen in the zebrafish model. This analysis does not seem convincing to me. Of course, one could score cells for any program and with a lenient-enough definition could argue that a subset of cells express that signature; but more convincing evidence is needed in order to support the claim of a coherent upregulation of

interface genes that is biologically and statistically significant; such evidence appears to be lacking in this analysis. The threshold of 0.5 is quite lenient and a subset of cells from each of the cell types passes it, as opposed to the expectation from the zebrafish analysis that this phenomena should be rather specific to the tumor cells; the co-expression of interface-related genes in those cells is not shown in any way (only the average score for the signature) and there is no basis for claiming statistical significance for this result. Currently, I do not see any evidence that the cell scores for interface-like signature go beyond what is expected by the overall variability of gene expression in this data and that point to the existence of interface-like cells.

One of the limitations in comparing our zebrafish data to the Tirosh et al. human data is the relatively low number of cells in that dataset (4,645 cells). Therefore, to further investigate the relationship between our zebrafish data and human melanoma, we took advantage of a very recently published scRNA-seq dataset of 29,247 cells from 43 patient metastatic melanoma samples (not cell lines) (Smalley et al., *Clin Cancer Res* 2021). This dataset contained more than 6-fold more cells than the Tirosh et al. dataset we previously analyzed (29,247 cells vs. 4,645 cells), providing significantly improved statistical power. We clustered the cells into 8 groups, using known marker genes for melanoma as well as non-melanoma marker genes from the original publication (**new Fig. S11a**). As we did previously, we scored each cell in the dataset for expression of the human orthologs of the genes upregulated by more than 1.5-fold in the interface cluster in our zebrafish scRNA-seq dataset (**new Fig. S11b**). Interestingly, with this much larger dataset, we were able to identify clearly distinct subsets of both tumor and microenvironment cells that upregulated the same genes upregulated in the interface in our zebrafish datasets (**new Fig. S11b-e**). These cells formed a distinct subcluster within the tumor, T/NK cell, and myeloid cell clusters (**new Fig. S11b-c**), suggesting they represent a distinct cell state and are not purely the effect of stochastic variability in gene expression. We also performed statistical analysis on the interface gene expression scores as requested, and found that the upregulation of interface genes within the 3 major groups of interface cells was highly statistically significant relative to the related cell types ($p = 7.3 \times 10^{-163}$ for tumor-like cells; $p = 5.4 \times 10^{-147}$ for T/NK-like cells; $p = 5.8 \times 10^{-16}$ for myeloid-like cells, Wilcoxon rank sum test with Bonferroni's correction; **new Fig. S11d**). In regards to the reviewer's concerns about the cutoff of 0.5 for classifying cells as interface-like, we would note that any score above 0 in the Seurat pipeline is considered an upregulation of the genes of interest (<https://github.com/satijalab/seurat/issues/522>), so a cutoff of 0.5 is actually relatively conservative.

As requested by the reviewer, we also plotted the expression of genes upregulated in the interface in our zebrafish datasets, such as *PLK1* (**Fig. 3e, 4b, and new S11e**), *HMGB2* (**Fig. S9 and new S11e**), *TUBB4B* (**Fig. 5f and new S11e**) and *TPX2* (**Fig. 3e and new S11e**). In all cases, we saw that these interface marker genes were highly upregulated within cells that we classified as "interface-like". Within the 3 major groups of interface cells (tumor-like, T/NK cell-like, and myeloid-like), we scored each interface cell and their corresponding tumor/TME cells for expression of the human SYSCILIA genes, and in all cases found that cilia genes were significantly upregulated in the interface cell state relative to the corresponding tumor/TME cell

type (**new Fig. S11f**; p -values are indicated). Thus, similar to our zebrafish analyses, our re-analysis of this new human melanoma scRNA-seq dataset also identifies a distinct “interface” cell state composed of both tumor and TME cells, in which cilia genes are upregulated.

In regards to the reviewer’s comment “*the expectation from the zebrafish analysis [is] that this phenomena should be rather specific to the tumor cells*”, we would note that we show in **Figs. 3, 4, 5, and 6** (among others) that the interface cell state is not restricted to tumor cells, and that we have found evidence for muscle, liver, intestinal, and immune cells sharing a common interface cell state with the tumor. In fact, the discovery of a shared interface cell state between tumor and TME cells is one of the major findings of this manuscript.

Reviewers' Comments:

Reviewer #2:

Remarks to the Author:

The staining of primary cilia with anti-Ac tubulin antibody was the only remaining issue we had with the revised manuscript. In the response to this criticism, we commend the authors for providing a control staining that shows more normal-looking cilia. To provide necessary context to readers we ask the authors to make two minor additions: a) include the control staining in supplementary figure 10, and b) include a statement in the main text to let readers know that the authors are aware of the unusual nature of the Ac-tubulin structures in the tumor-muscle interface ... that their length and discontinuity is unlike normal primary cilia. We do not wish for the authors to perform additional experiments, but suggest these additions would let readers know that the authors (and reviewers) are not overlooking something obvious.

Reviewer #3:

Remarks to the Author:

This is the second revision of the work by Hunter et al. about interface cells in a zebrafish model that has been studied by spatial transcriptomics and single cell RNA-seq. The authors tried to address my earlier comments, but unfortunately, these attempts are only partially successful as they are hindered by technical caveats and improper implementations. I therefore have follow up comments to all of my previous points, as described below.

1. Differential expression (comment #1 in previous round): The authors revised their analysis of differential expression between interface cells and those of tumor or muscle origin in order to exclude the option of doublet-related signals. They note that this was done as proposed in my previous comment, including a separate comparison to muscle and to tumor cells, but this was not done properly and has several major caveats. First, and most importantly, a gene was considered as significantly differentially expressed in interface compared to both other cell types, even if in the individual comparisons it had opposite signs, namely it was significantly higher in interface than in muscle but significantly lower in interface than in tumor cells (i.e. rows in Table S1 in which the last two columns have opposite signs). This is clearly not the way in which this analysis should be done as it will retain those genes that are consistent with the possibility of interface cells/spots representing a combination of muscle and tumor cells. Second, it is important to ensure that "significance" is not defined only by a statistical p-value but also by a fold-change threshold. With the availability of hundreds of spots or cells in each class it is very easy to obtain statistical significance that is biologically meaningless and hence I believe that it is essential to also add an effect-size threshold. Indeed, many of the genes that passed the current analysis have a fold-change of only 1.1 or 1.2, which is far from what would typically be considered as biologically meaningful. Third, this analysis should specifically highlight and relate back to the same genes that are noted in the rest of the paper as an interface signature and particularly to the cilia-related genes. Such consistency across analyses is currently lacking and would be needed in order to validate the authors claims.

2. Excluding doublets (comment #2 in previous round): The authors note several approaches to exclude the possibility of doublets. However, there are multiple caveats with this response and unfortunately I still remain unconvinced that much of the signal for interface cells may be related to doublets of distinct cell types. First, the authors argue based on Fig. S5B and S6B that interface cells do not have higher numbers of genes/UMIs compared to other cell types. However, at least visually, I do see in these figures an apparent higher numbers of genes in the interface cells; the significance of the difference is difficult to assess visually, but the authors' dismissal of this pattern without any specific analysis or p-value is unclear to me. Second, in the new Fig. S7 the authors compare the signal between pure cell types and the corresponding subsets of interface cells, demonstrating that only a limited fraction of the cell type-upregulated genes are shared with the corresponding subset of interface cells. However, this is expected and does not exclude the option of doublets, because if interface cells are doublets then only a part of their signal comes from the corresponding cell type and hence they would have a weaker expression of the genes reflecting that cell type, compared to the pure cell type. The analysis should instead be performed

in a more careful way, evaluating and comparing the expression of all groups of marker genes across the various subsets of interface and non-interface cells; if muscle interface cells have highest expression (compared to other interface cells) of the muscle genes, then they would be consistent with doublets of muscle cells; if instead, they would have their own unique signature that is not as high in muscle cells nor in other interface cells then it would support the authors claim of a unique cellular state. A heatmap combining all cell subsets and all sets of marker genes should help to resolve these questions but that is currently not included. The venn diagrams that are instead presented remain difficult to interpret because they compare each type of interface subset to all of the other cells; it is difficult to predict which genes would exactly come up in a doublet scenario and these would certainly not be exactly the genes that come up in one of the pure cell types that constitute the doublets but rather those cells that are highest in the specific doublet combination.

3. Inconsistencies in cilia expression (comment #4 in previous round): In response to my comment the authors argue that muscle cells have low expression of cilia genes, pointing to Fig. 5e and to the SRT data they include in the response. Both of these sources indeed support the authors argument; but at the same time, they point to major inconsistencies in the data. First, Fig. 5e shows a minimal difference in cilia genes between interface (tumor) and interface (muscle), in contrast to Fig. 5d which shows a very large difference. This suggests that despite using the same labels, the subsets of cells from scRNA-seq and snRNA-seq are quite distinct; this should be explained/clarified. Second, in the SRT figure pasted in the rebuttal there does not seem to be any upregulation of the cilia genes in the interface cells, in contrast to a central claim in the paper. These inconsistencies in the pattern of interface cells across modalities and figures make it difficult to reach any coherent view of the meaning of those cells and their expression of cilia and other genes. What is lacking is a unifying figure similar to Fig. 3e but that shows the signature of interface cells across modalities/datasets, such that individual genes can be assessed for their upregulation in individual modalities (and with the appropriate comparisons to reference cells). This signature should also be included as a supplementary table, as it is currently difficult to understand which genes are included in this list beyond a few labeled genes and the cilia enrichment.

4. Similar interface signature across cell types (related to previous comment #5): The authors revise the analysis of melanoma patients data, which now shows an interface expression state in multiple cell types. This is also somewhat consistent with the snRNA-seq analysis. Accordingly, the authors note that "the discovery of a shared interface cell state between tumor and TME cells is one of the major findings of this manuscript". I agree that if this is correct then it is an important finding. I would note that the initial manuscript highlighted an interface cancer/muscle state - which sounds very reasonable - while the revised paper's notion of a common interface state is much more surprising and unexpected. I would naively expect that interface programs would differ between cell types as distinct as tumor cells, macrophages and lymphocytes. Given this unexpected observation I would urge the authors to be extra careful about this claim and to ensure its strength and lack of confounding effects. Accordingly, I would suggest the addition of a heatmap with the common interface signature genes, as well as cell type markers, shown across multiple cell types, each of which separated into interface cells and non-interface cells. This would provide a more concrete and "raw-data" view of this finding in a way that would reduce any doubts.

Reviewer #4:

Remarks to the Author:

All concerns raised by reviewer #1 have been satisfactorily addressed in the revised manuscript.

Reviewer #2:

The staining of primary cilia with anti-Ac tubulin antibody was the only remaining issue we had with the revised manuscript. In the response to this criticism, we commend the authors for providing a control staining that shows more normal-looking cilia. To provide necessary context to readers we ask the authors to make two minor additions: a) include the control staining in supplementary figure 10, and b) include a statement in the main text to let readers know that the authors are aware of the unusual nature of the Ac-tubulin structures in the tumor-muscle interface ... that their length and discontinuity is unlike normal primary cilia. We do not wish for the authors to perform additional experiments, but suggest these additions would let readers know that the authors (and reviewers) are not overlooking something obvious.

As suggested, we have added the control staining as **new Fig. S13** and a description of the differences between these structures and typical cilia to the text (lines 342-345).

Reviewer #3:

This is the second revision of the work by Hunter et al. about interface cells in a zebrafish model that has been studied by spatial transcriptomics and single cell RNA-seq. The authors tried to address my earlier comments, but unfortunately, these attempts are only partially successful as they are hindered by technical caveats and improper implementations. I therefore have follow up comments to all of my previous points, as described below.

1. Differential expression (comment #1 in previous round): The authors revised their analysis of differential expression between interface cells and those of tumor or muscle origin in order to exclude the option of doublet-related signals. They note that this was done as proposed in my previous comment, including a separate comparison to muscle and to tumor cells, but this was not done properly and has several major caveats.

First, and most importantly, a gene was considered as significantly differentially expressed in interface compared to both other cell types, even if in the individual comparisons it had opposite signs, namely it was significantly higher in interface than in muscle but significantly lower in interface than in tumor cells (i.e. rows in Table S1 in which the last two columns have opposite signs). This is clearly not the way in which this analysis should be done as it will retain those genes that are consistent with the possibility of interface cells/spots representing a combination of muscle and tumor cells.

We agree that this is an important point, but we think this reviewer may be misinterpreting this analysis and **Table S1**. The analysis was done following this suggestion from this reviewer in a previous round of comments:

“...a very simple conservative approach would be to compare the interface expression to the maximum of each gene between tumor and muscle areas, thereby ensuring that any gene that comes up as upregulated cannot simply be explained by cell type mixture; similarly, downregulated genes may be defined as those lower in the interface than in the minimum of the two types of regions (tumor/muscle).”

This is precisely the analysis we are presenting in **Table S1** and **Fig. 2c**. For all genes we quantified as upregulated in the interface (column 1, log2FC_in_interface), we also calculated the expression of each of those genes within the tumor (column 3, log2FC_in_tumor) and muscle clusters (column 4, log2FC_in_muscle). We then filtered this list to only include genes that had higher log2FC in the interface (column 1) compared to both tumor (column 3) and muscle (column 4). The reviewer mentions a hypothetical example where a given gene has higher log2FC in the interface than muscle, but a lower log2FC in the interface than tumor. Any genes like this would have been filtered out of our genelist, because we only consider a gene to be upregulated in the interface if it has a higher log2FC in the interface relative to the log2FC in tumor and the log2FC in muscle, as you can see in **Table S1**. The example the reviewer indicated would require the value in column 1 to be lower than at least one of the values in column 3 or 4 (for all upregulated genes), which is not the case for any of the genes in **Table S1**. We agree that this hypothetical example would be the incorrect way to do this analysis, but we would invite the reviewer to indicate which genes in Table S1 we have misclassified. The

reviewer mentions cases where columns 3 and 4 (expression in tumor and muscle) have opposite signs - this just means that the given gene is upregulated in either tumor or muscle and downregulated in the other (a common and expected occurrence). This is not related to interface expression of the given gene, which is shown in the first column.

Second, it is important to ensure that "significance" is not defined only by a statistical p-value but also by a fold-change threshold. With the availability of hundreds of spots or cells in each class it is very easy to obtain statistical significance that is biologically meaningless and hence I believe that it is essential to also add an effect-size threshold. Indeed, many of the genes that passed the current analysis have a fold-change of only 1.1 or 1.2, which is far from what would typically be considered as biologically meaningful.

We agree that many of the genes we have classified as upregulated in the interface have a somewhat low log₂FC. The relatively low expression of these genes is likely related to the lower amount of transcripts captured by SRT compared to scRNA-seq. This is evident in comparing the number of UMIs captured per SRT spot (**Fig. S1f**; approximately 1,000 - 15,000 UMIs/spot, containing multiple cells) and per cell (**Fig. S5b,e**; approximately 10,000 - 75,000 UMIs/per cell). This is a known technical limitation of SRT experiments, which is why we paired our SRT results with scRNA-seq. The challenge in setting a fold-change cutoff is that many genes with relatively low log₂FC can still have important biological effects, and many highly expressed genes may not be important. For these reasons, it would be hard to know where to arbitrarily implement a log₂FC threshold, and could be detrimental in excluding genes that may be biologically important.

We would note that the purpose of this analysis is simply to calculate which genes are up- or downregulated in the interface relative to tumor and muscle. We are not attempting to draw biological meaning purely from this list of genes, as we later do many additional analyses to investigate the biology of the interface, such as NMF (**Fig. S4**), GSEA (**Fig. 5c**), and motif analysis (**Fig. 6a**), as well as using scRNA-seq and snRNA-seq to improve the resolution of our SRT data. Furthermore, analyses such as GSEA and NMF take into account the relative expression of each gene, so the somewhat low expression of these interface genes would be included in these analyses. Finally, as one of the points suggested by our SRT and other datasets is the identification of cilia genes at the interface, this is exactly why we performed protein immunofluorescence (as discussed in point #2 below) as a separate and orthogonal validation.

Third, this analysis should specifically highlight and relate back to the same genes that are noted in the rest of the paper as an interface signature and particularly to the cilia-related genes. Such consistency across analyses is currently lacking and would be needed in order to validate the authors claims.

We have added a sentence to the text highlighting certain microtubule/cilia-related genes from Table S1 (*tuba1a*, *tuba1c*; lines 146-147). Additionally, see response to point #4 below. We have added a **new Fig. S11** and **Table S2** defining an interface signature across the different modalities and again highlighting certain cilia genes.

2. Excluding doublets(comment #2 in previous round): The authors note several approaches to exclude the possibility of doublets. However, there are multiple caveats with this response and unfortunately I still remain unconvinced that much of the signal for interface cells may be related to doublets of distinct cell types. First, the authors argue based on Fig. S5B and S6B that interface cells do not have higher numbers of genes/UMIs compared to other cell types. However, at least visually, I do see in these figures an apparent higher numbers of genes in the interface cells; the significance of the difference is difficult to assess visually, but the authors' dismissal of this pattern without any specific analysis or p-value is unclear to me.

We have done the suggested statistical tests of the UMI/gene data in Figs. S5b, S5e, and S6b and added the p-values to the plots. If the interface clusters were composed of a large fraction of doublets, we might expect to see that the interface cluster expressed significantly more UMIs/genes than every other cluster in the dataset (although we cannot rule out that the interface region may also just be more transcriptionally active, as Table S1 suggests - see lines 146-147). Instead, we did not consistently see such a pattern. In some cases there was no difference between the interface clusters and the other clusters in the dataset, particularly in our scRNA-seq reaction 2 (both genes and UMIs). In other cases there were statistically significant differences between the interface clusters and certain other clusters in the datasets, however in many cases the p-values were relatively high and not even consistent between the two scRNA-seq reactions. In addition, in some cases where there were significant differences the interface actually expressed fewer UMIs/genes than other clusters, such as the intestinal cluster in our snRNA-seq dataset. We have added a description of these results to the text, clearly stating that the results were inconclusive: in some cases the interface did not express significantly more genes/UMIs than other clusters, whereas in other cases it did (lines 249-257). Thus, while UMI calculations can be useful to help exclude doublets, it is not the most robust way to do this, which is why we performed further analyses (i.e. DoubletFinder, as recommended by 10X Genomics as well as other analyses) which are described below and in the text (lines 257-268).

Second, in the new Fig. S7 the authors compare the signal between pure cell types and the corresponding subsets of interface cells, demonstrating that only a limited fraction of the cell type-upregulated genes are shared with the corresponding subset of interface cells. However, this is expected and does not exclude the option of doublets, because if interface cells are doublets then only a part of their signal comes from the corresponding cell type and hence they would have a weaker expression of the genes reflecting that cell type, compared to the pure cell type. The analysis should instead be performed in a more careful way, evaluating and comparing the expression of all groups of marker genes across the various subsets of interface and non-interface cells; if muscle interface cells have highest expression (compared to other interface cells) of the muscle genes, then they would be consistent with doublets of muscle cells; if instead, they would have their own unique signature that is not as high in muscle cells nor in other interface cells then it would support the authors claim of a unique cellular state. A heatmap combining all cell subsets and all sets of marker genes should help to resolve these

questions but that is currently not included. the venn diagrams that are instead presented remain difficult to interpret because they compare each type of interface subset to all of the other cells; it is difficult to predict which genes would exactly come up in a doublet scenario and these would certainly not be exactly the genes that come up in one of the pure cell types that constitute the doublets but rather those cells that are highest in the specific doublet combination.

In the previous review, the Reviewer stated “*if it is a doublet, then a large fraction of all genes unique to a certain immune cell type should be expressed by that cell, while if it is truly a state of interface cell that expresses certain immune-related genes, then the fraction of such immune-specific genes should be quite limited.*” In this new review, they then state “*...the authors compare the signal between pure cell types and the corresponding subsets of interface cells, demonstrating that only a limited fraction of the cell type-upregulated genes are shared with the corresponding subset of interface cells. However, this is expected and does not exclude the option of doublets*”. It is difficult to reconcile these two statements, since in the prior review the reviewer suggested this exact analysis to exclude doublets, yet now the reviewer states this analysis supports the presence of doublets.

As requested, we have created a heatmap to better examine this question on a gene level (**new Fig. S8**). In this analysis, we would expect, for example, that a “muscle-like interface cell” would (by definition) express genes also expressed by muscle cells, as well as interface-specific genes shared by all interface cells. In this heatmap, we find exactly that. Some genes (i.e. *atp2a1l*) are expressed in both the muscle and the muscle-interface cell; other genes (i.e. *zp3.2*) are more highly expressed in the muscle-like interface than in the muscle. It is unclear how this result could be caused by doublets. We previously showed this exact effect in Fig. 3g, in which several genes (*BRAF^{V600E}*, *mitfa*, *pmela*) are actually more highly expressed in, for example, the tumor-like interface cells compared to the tumor, which would be inconsistent with doublets, since (as the reviewer mentioned) a doublet between a tumor cell and another cell type would result in lower expression of tumor-specific genes within the doublet, not higher expression. Furthermore, Figs. 3e and 4f show many genes highly upregulated in interface cells and not expressed in any other cluster, which is also inconsistent with multipliers.

We acknowledge the reviewer’s concerns about doublets, but would note that the issue of doublets and multipliers is a general issue for the single-cell field, and a caveat in all single-cell transcriptomics datasets. It does not seem possible that we can entirely solve a problem in the field in this one paper, but we have gone to significant lengths to exclude doublets and explicitly addressed this possibility in the manuscript. We have now completed at least 5 different filtering steps and analyses related to the issue of doublets (quantification of UMIs/genes, DoubletFinder scoring and filtering, calculation of overlap of cell-type specific genes, heatmap showing expression of interface- and cell type-specific genes). We also already explicitly stated in the text that there may be doublets present in the interface due to biological or technical reasons, and that in fact tumor-immune cell doublets have been reported in melanoma (lines 264-267). Thus, we feel that we have gone far beyond the scope of what is typically expected in the single-cell transcriptomics field to investigate the presence of doublets, and have also openly

stated in the text that some doublets may be present. We are not sure what else we could do to address this topic, as with the data at hand we cannot determine whether any possible doublets could be due to biological or technical reasons.

3. Inconsistencies in cilia expression (comment #4 in previous round): In response to my comment the authors argue that muscle cells have low expression of cilia genes, pointing to Fig. 5e and to the SRT data they include in the response. Both of these sources indeed support the authors argument; but at the same time, they point to major inconsistencies in the data. First, Fig. 5e shows a minimal difference in cilia genes between interface (tumor) and interface (muscle), in contrast to Fig. 5d which shows a very large difference. This suggests that despite using the same labels, the subsets of cells from scRNA-seq and snRNA-seq are quite distinct; this should be explained/clarified.

We agree that we observed a strong upregulation of cilia genes in the muscle-like interface cluster in our scRNA-seq data but observed a dampened result in our snRNA-seq data. We think a very important reason for this is the much lower dynamic range of snRNA-seq compared to scRNA-seq. We had already mentioned this in our manuscript, in which we observed relatively low expression of most genes, including cilia genes, in our snRNA-seq data relative to our scRNA-seq data, due to the well known technical limitations of snRNA-seq (lines 321-326 and **Fig. S6c-d**). Thus, our ability to see large fold-changes in our snRNA-seq data is limited, and likely explains why we saw somewhat dampened expression of cilia genes in the muscle-like interface cluster in our snRNA-seq dataset relative to our scRNA-seq dataset. Despite this limitation, we still showed in **Fig. 5f** that muscle-like interface cells expressed the given cilia genes most highly compared to the other interface cell states for all 4 given genes. Thus, this pattern is entirely consistent with our scRNA-seq results. We have added a sentence to the text describing the strong upregulation of cilia genes in the muscle-like interface cluster in our scRNA-seq data (lines 320-321), and the corresponding upregulation of the cilia genes in **Fig. 5f** in the muscle-like interface cluster (lines 328-329).

An important point here is that while our RNA-seq analyses (SRT, scRNA-seq, snRNA-seq) suggested that cilia are upregulated at the interface (albeit to different extents), it is ultimately evidence at the protein level with immunofluorescence that is the key orthogonal proof. This is why we feel that the RNA evidence is strongly bolstered by the IF staining for cilia in **Figs. 5g** and **S12**. These results clearly show a highly specific accumulation of acetylated tubulin, a gold standard cilia marker, at the interface. As showing protein localization/enrichment is perhaps the ultimate validation of transcriptomics data, we believe our IF result clearly validates our transcriptomics results suggesting an enrichment of cilia genes at the interface, as we clearly see a very specific accumulation of cilia proteins in the interface region. We do not see where these cilia proteins could be coming from if they are not being produced by interface cells upregulating cilia genes. Our scRNA-seq results also show a clear and statistically significant upregulation of cilia genes in interface cells. Thus, based on our scRNA-seq, snRNA-seq, and IF results we do not feel that there can be any question that cilia genes/proteins are upregulated at the interface.

Second, in the SRT figure pasted in the rebuttal there does not seem to be any upregulation of the cilia genes in the interface cells, in contrast to a central claim in the paper. These inconsistencies in the pattern of interface cells across modalities and figures make it difficult to reach any coherent view of the meaning of those cells and their expression of cilia and other genes. What is lacking is a unifying figure similar to Fig. 3e but that shows the signature of interface cells across modalities/datasets, such that individual genes can be assessed for their upregulation in individual modalities (and with the appropriate comparisons to reference cells). This signature should also be included as a supplementary table, as it is currently difficult to understand which genes are included in this list beyond a few labeled genes and the cilia enrichment.

We agree that generating a list of interface-specific genes across all 3 modalities would help to interpret this data. However, we also recognize the limitations of this approach, since these 3 modalities (SRT, scRNA-seq, snRNA-seq) have very different detection limits, dynamic ranges, and resolution, as we discuss in the manuscript. Thus, it is likely that at the individual gene level, there could be substantial differences in which ones are most altered in the interface from datasets generated using different modalities. With this caveat in mind, we have performed the suggested analysis by calculating all genes significantly upregulated within the interface clusters relative to all other cells/spots in each of our SRT, scRNA-seq, and snRNA-seq datasets, using the Wilcoxon rank sum test. We then combined those 3 lists into one unified list, and then filtered that list to only include genes that were upregulated in all 3 of the SRT, scRNA-seq and snRNA-seq interface clusters. For this analysis of the SRT data specifically, we felt that excluding genes with a higher log₂FC in tumor/muscle than the interface, as we did in Fig. 2c and Table S1, was not necessary. The reason we did this filtering step for Fig. 2c was to correct for SRT spots that may also contain “normal” tumor and muscle cells. In this analysis, we are now able to take advantage of our scRNA-seq and snRNA-seq datasets to more effectively delineate interface-specific genes from tumor- or microenvironment-specific genes. Since our scRNA-seq/snRNA-seq datasets are single-cell resolution, any genes upregulated in interface cells in those datasets cannot be attributed to contaminating tumor or microenvironment cells, and must be true interface-specific genes. Therefore, we felt it was most appropriate here to filter the list of SRT interface genes by the scRNA-seq and snRNA-seq interface genes, rather than by tumor/muscle log₂FC. As such, this list may vary slightly from Table S1, where we used a very conservative approach of excluding genes with higher log₂FC in tumor/muscle. This does not mean that Table S1 is incorrect - rather, that our conservative filtering approach may have resulted in genes being excluded from this list that our scRNA-seq and snRNA-seq datasets later revealed to be important. This new analysis resulted in a list of 55 genes upregulated in the interface across our scRNA-seq, snRNA-seq and SRT datasets. We would note again the significant technical differences between our SRT, scRNA-seq, and snRNA-seq datasets and the differences in the number of genes and UMIs detected across these datasets, as quantified in **Figs. S1, S5a-f, and S6a-b**. Some genes highlighted within the manuscript may not be present on this list, due to lower expression of these genes within at least one of the datasets. However, we did identify a number of cilia genes on this list, including *stmn1a*, *ran*, *tuba8l4*, and *tubb4b*, which again is consistent with an enrichment of cilia in the interface (that we have most importantly validated by immunofluorescence in **Figs. 5g and S12**). We then created heatmaps

showing expression of these common interface marker genes within each of the SRT, scRNA-seq and snRNA-seq datasets. We note the clear upregulation of these interface-specific genes in each of the scRNA-seq and snRNA-seq datasets. The lower expression of these genes within the interface cluster of the SRT dataset likely relates to the detection/resolution issues with SRT that we already mentioned above. We have added these heatmaps as **new Fig. S11**, and the list of common interface genes as **Table S2**.

4. Similar interface signature across cell types (related to previous comment #5): The authors revise the analysis of melanoma patients data, which now shows an interface expression state in multiple cell types. This is also somewhat consistent with the snRNA-seq analysis. Accordingly, the authors note that "the discovery of a shared interface cell state between tumor and TME cells is one of the major findings of this manuscript". I agree that if this is correct then it is an important finding. I would note that the initial manuscript highlighted an interface cancer/muscle state - which sounds very reasonable - while the revised paper's notion of a common interface state is much more surprising and unexpected. I would naively expect that interface programs would differ between cell types as distinct as tumor cells, macrophages and lymphocytes. Given this unexpected observation I would urge the authors to be extra careful about this claim and to ensure its strength and lack of confounding effects. Accordingly, I would suggest the addition of a heatmap with the common interface signature genes, as well as cell type markers, shown across multiple cell types, each of which separated into interface cells and non-interface cells. This would provide a more concrete and "raw-data" view of this finding in a way that would reduce any doubts.

While the initial paper was indeed focused on the cancer/muscle interface, the more recent availability of the new scRNA-seq data from human melanoma patients, as well as our new snRNA-seq dataset, gave us a better opportunity to probe whether this interface state occurred more broadly. To better visualize this, we have added the suggested heatmap to **Fig. S14g**, where we plotted the human orthologs of the interface gene signature in Table S2, as well as markers for the tumor, T/NK-cell, and myeloid clusters. We do not yet know the mechanism for why this interface signature is present in those other cell types such as macrophages and lymphocytes, but this is a key area for future exploration. To ensure that we are appropriately cautious about overinterpreting this result without this mechanism, we have added a discussion of this as a future important mechanistic direction (lines 458-467).

Reviewers' Comments:

Reviewer #3:

Remarks to the Author:

The authors have addressed most of my comments, but I still have some final concerns/suggestions, mostly about defining interface-upregulated genes:

1. The authors retain in Table S1 all the genes that are higher in interface than the maximum of tumor and muscle, without any requirement for a fold-change (and also without a proper p-value as noted below). Many of the included genes, including those cited in the paper and those relating to cilia have extremely low fold-changes such that effectively it would seem more accurate to say that they are comparable in the interface to one of the cell types (tumor or muscle) rather than higher. For example, *tuba1a* and *tuba1c*, which the authors specifically mention in their rebuttal, have a log₂FC difference of only 0.04 and 0.06 compared to tumor, indicating a fold-change of only 1.03 and 1.04. I don't think that this can be considered as reliable evidence for upregulation. I do understand the authors point about technical difficulties making it difficult to detect upregulation, but this still does not imply that it is acceptable to define such genes as being upregulated in interface cells compared to tumor cells based on such minimal and unreliable differences.

2. Importantly, these minimal differences are not evaluated directly by the p-value, as the p-value is calculated based on comparison to both tumor and muscle cells combined, which leads to cases of astronomical p-values but minimal fold-changes below 1.05. Therefore, I would suggest re-defining an adjusted p-value for interface vs. max(tumor,muscle), instead of interface vs. (tumor and muscle), then retaining in the table/figure all genes that come up as significant, but while marking genes below a certain fold-change threshold (say 1.5) as "potential" upregulation and acknowledging this caveat. If I understand correctly, after such correction the cilia genes would no longer be defined as upregulated which would not support the author's claim.

3. In the list of common upregulated genes (Table S2), the most common function appears to be of ribosomal protein genes, but this is not mentioned and should be discussed. Do the authors believe that this is a technical or biological effect? if the list is reported as it is now and future studies would use it then the "interface signature" would effectively reflect mostly the ribosomal protein (RP) genes, and hence if the authors do not think that this is a biological effect it would be good to try to define a clean version that reflects a true interface signature. Along these lines, did the authors use this RP-rich signature for any other analysis and if so, wouldn't that affect the analysis?

4. Figure 4F is helpful, and it would be good to label more of the genes in it. However, it also demonstrates that there is high expression of many of the interface genes in erythrocytes, which is not mentioned in the manuscript. Can the authors explain this effect? could it reflect ambient RNA that is either released from erythrocytes or alternatively released by other cells and taken up by erythrocytes? If so, the genes appearing specifically high in erythrocytes might need to be removed from analysis of interface-specific genes.

5. Figure S8 is an excellent addition to the paper. But the interface cells which are the main focus here are very dense and it is difficult to understand their patterns. I would therefore suggest keeping the content of the heatmap but splitting it into two - one for just interface cells and one for other cell types - and then show both at equal sizes such that there would be more space devoted to the interface cells and it would be easier to see their patterns.

Reviewer #3 (Remarks to the Author):

The authors have addressed most of my comments, but I still have some final concerns/suggestions, mostly about defining interface-upregulated genes:

1. The authors retain in Table S1 all the genes that are higher in interface than the maximum of tumor and muscle, without any requirement for a fold-change (and also without a proper p-value as noted below). Many of the included genes, including those cited in the paper and those relating to cilia have extremely low fold-changes such that effectively it would seem more accurate to say that they are comparable in the interface to one of the cell types (tumor or muscle) rather than higher. For example, *tuba1a* and *tuba1c*, which the authors specifically mention in their rebuttal, have a log2FC difference of only 0.04 and 0.06 compared to tumor, indicating a fold-change of only 1.03 and 1.04. I don't think that this can be considered as reliable evidence for upregulation. I do understand the authors point about technical difficulties making it difficult to detect upregulation, but this still does not imply that it is acceptable to define such genes as being upregulated in interface cells compared to tumor cells based on such minimal and unreliable differences.

We completely agree that many of these genes, which are very highly upregulated in our scRNA-seq and/or snRNA-seq datasets, display somewhat lower log2FCs in our SRT dataset. We too felt cautious about classifying these genes as "upregulated" in our SRT dataset due to their relatively low log2FC. The issue in filtering these out entirely, though, is that the increased resolution of scRNA-seq/snRNA-seq revealed that many of these very low fold-change genes in SRT are actually very strongly upregulated in the interface at the single-cell level (i.e., log2FC > 2). So too strict a filtering of interface-specific genes from our SRT dataset by log2FC would have caused us to miss genes that are actually highly upregulated at the single-cell level, including some cilia genes that our scRNA-seq/snRNA-seq and IF data show to be clearly upregulated at the interface. Thus, removing those genes could also be confusing.

Perhaps the most transparent way to deal with this is to present all genes that we have calculated to be upregulated (based on statistical significance and log2FC, see point #2 below) in the interface, even if the upregulation is subtle, with the corresponding p-values explicitly defined (as described below) along with log2FC in Table S1. We have converted Table S1 to an Excel spreadsheet such that the reader can then "filter" the list themselves as they see fit. To make this clear what we have done and our rationale, we have also added a sentence to the text acknowledging the somewhat lower upregulation of these interface-specific genes in our SRT dataset (lines 148-151) but that many of these were found to have higher FC in the scRNA-seq/snRNA-seq.

2. Importantly, these minimal differences are not evaluated directly by the p-value, as the p-value is calculated based on comparison to both tumor and muscle cells combined, which leads to cases of astronomical p-values but minimal fold-changes below 1.05. Therefore, I would suggest re-defining an adjusted p-value for interface vs. max(tumor,muscle), instead of interface vs. (tumor and muscle), then retaining in the table/figure all genes that come up as

significant, but while marking genes below a certain fold-change threshold (say 1.5) as "potential" upregulation and acknowledging this caveat. If I understand correctly, after such correction the cilia genes would no longer be defined as upregulated which would not support the author's claim.

We note that as we had stated in the figure legend, the p-value displayed in Table S1 (pval_in_interface) is actually not the p-value for the given gene compared to the tumor and muscle clusters, but the p-value for the given gene compared to all other SRT array spots (i.e. all other tissues present on the entire array). However, we agree that a better way to do this analysis is to adjust the p-value to be relative to the tumor and/or muscle clusters. In the interest of transparency and to allow the reader to interpret Table S1 as they see fit, we have added three new columns to Table S1: (1) the p-value for the given gene in the interface relative to tumor and muscle clusters; (2) the p-value for the given gene in the interface relative to the tumor cluster only; (3) the p-value for the given gene in the interface relative to the muscle cluster only. As we stated above, we feel that in the interest of transparency and for technical reasons related to limitations of SRT described above and in previous rebuttals, we prefer not to filter Table S1 by p-value or log2FC. We prefer to provide the reader with all relevant information in Table S1 (log2FCs and p-values for the interface, tumor and muscle clusters) and allow them to filter/interpret the table themselves as they see fit.

3. In the list of common upregulated genes (Table S2), the most common function appears to be of ribosomal protein genes, but this is not mentioned and should be discussed. Do the authors believe that this is a technical or biological effect? If the list is reported as it is now and future studies would use it then the "interface signature" would effectively reflect mostly the ribosomal protein (RP) genes, and hence if the authors do not think that this is a biological effect it would be good to try to define a clean version that reflects a true interface signature. Along these lines, did the authors use this RP-rich signature for any other analysis and if so, wouldn't that affect the analysis?

You raise a good point, since it is often difficult to determine whether an enrichment of ribosomal genes suggests a technical or biological effect. While it is certainly possible that many of the ribosomal genes could have biological function, given this uncertainty we had indeed filtered out ribosomal genes from the gene lists used for GSEA and other analyses (this was indicated in the Methods of the prior version). As such, we feel that it is best to also filter ribosomal genes out from Table S2 and Fig. S8. We have removed these genes using the same method we use for filtering of our GSEA gene lists (removing any gene beginning with "rps" or "rpl") and have added a description of this to the Methods.

4. Figure 4F is helpful, and it would be good to label more of the genes in it. However, it also demonstrates that there is high expression of many of the interface genes in erythrocytes, which is not mentioned in the manuscript. Can the authors explain this effect? Could it reflect ambient RNA that is either released from erythrocytes or alternatively released by other cells and taken up by erythrocytes? If so, the genes appearing specifically high in erythrocytes might need to be removed from analysis of interface-specific genes.

We agree that some genes upregulated in the interface are also upregulated in the erythrocyte cluster in our snRNA-seq dataset (although none of the top enriched genes seem to be upregulated in erythrocytes). We are not entirely sure what to make of this as we do not see the same effect in our scRNA-seq dataset (see Fig. 3e). However, we are hesitant to draw biological conclusions about the erythrocyte cluster in our snRNA-seq dataset due to the unusual nature of zebrafish erythrocyte nuclei. Unlike mammalian erythrocytes, zebrafish erythrocytes are nucleated. However, the nature of those nuclei, and their transcriptional activity, is not clear. In one study (<https://www.ncbi.nlm.nih.gov/pmc/articles/PMC5977992/>) they found considerable chromatin heterogeneity in these condensed nuclei, which could easily reflect the short-lived nature of these cells in circulation, and the mechanical damage inflicted on these cells as they travel through the vasculature. This makes it difficult in the snRNA-seq data to know if this is a technical artifact rather than true biological effect. We feel comfortable including the erythrocyte cluster in our scRNA-seq dataset as we used cell viability dyes and FACS sorting to isolate viable (i.e. non-dying) single cells for the experiment, giving us confidence that the erythrocytes present in our scRNA-seq dataset are in a healthy state. However, we are not able to have the same confidence in the erythrocyte cluster in our snRNA-seq dataset, as we cannot determine whether these nuclei were isolated from healthy erythrocytes or not, and due to erythrocyte biology and lifespan, it is likely that a large fraction of these nuclei may come from damaged or dying cells. We would also note that we have not been able to find any other published snRNA-seq datasets of zebrafish erythrocyte nuclei, making us unable to compare the erythrocytes in our snRNA-seq dataset to a reference snRNA-seq dataset. Thus, due to the discrepancies between the erythrocyte clusters in our scRNA-seq and snRNA-seq datasets, we feel that it is best to remove the erythrocyte cluster from our snRNA-seq dataset to ensure that we have the highest confidence in the dataset that we are presenting in our paper and making available to the scientific community. We have thus removed the erythrocyte cluster from the data presented in Fig. 4 and related supplemental figures, and added a sentence to the Methods stating this. We have also labelled more genes in Fig. 4f as suggested.

5. Figure S8 is an excellent addition to the paper. But the interface cells which are the main focus here are very dense and it is difficult to understand their patterns. I would therefore suggest keeping the content of the heatmap but splitting it into two - one for just interface cells and one for other cell types - and then show both at equal sizes such that there would be more space devoted to the interface cells and it would be easier to see their patterns.

We have made the suggested changes to Fig. S8.

Reviewers' Comments:

Reviewer #3:

Remarks to the Author:

The authors have successfully addressed all of my comments